# Assessment of human leukocyte antigen-based neoantigen presentation to determine pan-cancer response to immunotherapy

Jiefei Han [1,2,8], Yiting Dong [1,8], Xiuli Zhu [3,4,5,8], Alexandre Reuben [6,8], Jianjun Zhang [6], Jiachen Xu [1], Hua Bai[1], Jianchun Duan [1], Rui Wan [1], Jie Zhao [1], Jing Bai[3], Xuefeng Xia[3], Xin Yi[3], Chao Cheng [7] ✉, Jie Wang [1] ✉ & Zhijie Wang [1] ✉

Despite the central role of human leukocyte antigen class I (HLA-I) in tumor neoantigen presentation, quantitative determination of presentation capacity remains elusive. Based on a pooled pan-cancer genomic dataset of 885 patients treated with immune checkpoint inhibitors (ICIs), we developed a score integrating the binding affinity of neoantigens to HLA-I, as well as HLA-I allele divergence, termed the HLA tumor-Antigen Presentation Score (HAPS). Patients with a high HAPS were more likely to experience survival benefit following ICI treatment. Analysis of the tumor microenvironment indicated that the antigen presentation pathway was enriched in patients with a high HAPS. Finally, we built a neural network incorporating factors associated with neoantigen production, presentation, and recognition, which exhibited potential for differentiating cancer patients likely to benefit from ICIs. Our findings highlight the clinical utility of evaluating HLA-I tumor antigen presentation capacity and describe how ICI response may depend on HLA-mediated immunity.

Immune checkpoint inhibitors (ICIs) targeting programmed death-1 (PD-1), programmed death-ligand 1 (PD-L1), and cytotoxic T lymphocyte-associated antigen 4 (CTLA-4) are used for the treatment of various malignancies, markedly improving survival[1–4]. However, only a subset of patients benefits from ICIs. Therefore, researchers are exploring predictive biomarkers of ICI response, e.g., PD-L1 expression, tumor mutational burden (TMB)[5–7], tumor immune phenotype[8], somatic genomic features[9], and the gut microbiome[10]. Effective tumor neoantigen production is essential for anti-tumor immunity. It is

partially reflected by the TMB due to the randomness of tumor mutations. Generally, ICI response-predictive biomarkers are helpful yet insufficient, with patient stratification for immunotherapy remaining an active research area.

The major histocompatibility complex (MHC), or human leukocyte antigen (HLA) in humans, plays a central role in neoantigen peptide presentation[11]. The HLA-I genotype includes a pair of alleles at each classic class I gene (*HLA-A, -B, -C*). The divergence observed in HLA alleles plays a pivotal role in

[1]State Key Laboratory of Molecular Oncology, CAMS Key Laboratory of Translational Research on Lung Cancer, Department of Medical Oncology, National Cancer Center/National Clinical Research Center for Cancer/Cancer Hospital, Chinese Academy of Medical Sciences and Peking Union Medical College, Beijing 100021, China. [2]Department of Neuro-oncology, Neurosurgery Center, Beijing Tiantan Hospital, Capital Medical University, Beijing, China. [3]Geneplus-Beijing Institute, Beijing, China. [4]University of Chinese Academy of Sciences, Beijing 100049, China. [5]CAS Key Laboratory of Genomic and Precision Medicine, Beijing Institute of Genomics, Chinese Academy of Sciences and China National Center for Bioinformation, Beijing 100101, China. [6]Department of Thoracic/Head and Neck Medical Oncology, The University of Texas M. D. Anderson Cancer Center, Houston, Texas, USA. [7]Department of Medicine, Dan L Duncan Comprehensive Cancer Center, Institute for Clinical and Translational Research, Baylor College of Medicine, Houston, Texas, USA. [8]These authors contributed equally: Jiefei Han, Yiting Dong, Xiuli Zhu, Alexandre Reuben. ✉e-mail: Chao.Cheng@bcm.edu; zlhuxi@163.com; wangzj@cicams.ac.cn

determining the sequence variations within the peptide-binding domains of HLA-I alleles in patients[12,13]. This divergence is quantified by taking the average Grantham distance among paired alleles within each HLA-I subtype[14]. This metric offers insight into the spectrum of neoantigens that have the potential to bind to HLA-I molecules. Each classic *HLA-I* genotype exhibits distinct divergences at pairwise alleles and consequently exhibits differential neoantigen presentation capacity, highlighting the necessity to measure *HLA-I* divergence in consideration of the corresponding neoantigens. Evaluation of *HLA-I* divergences may be severely affected by loss of heterozygosity (LOH) in HLA (LOHHLA), a mechanism of immune evasion to be considered when assessing MHC presentation[15].

Considering the complex immune microenvironment, an integrated multi-parameter model may be of great value for predicting ICI efficacy. Tumor-specific immunogenetics may be largely reflected by neoantigen production and presentation[9,16]. We previously identified functional T cell receptor (TCR) repertoires to assess reinvigoration following neoantigen recognition by CD8+PD-1+T cells, to differentiate lung cancer patients who may benefit from ICI[17,18]. To our knowledge, no prior study has investigated these probabilities through combined evaluation of HLA and TCR repertoires. Thus, such a functional model is expected to improve patient stratification.

We aimed to develop a method for evaluating HLA-I-mediated neoantigen presentation [the *HLA-I* Antigen Presentation Score (HAPS)] by integrating information on predicted neoantigens, MHC-I binding affinity, and *HLA-I* allele divergences. HAPS-related tumor microenvironment (TME) features were also considered. We verified the feasibility of [1] a targeted next-generation sequencing (NGS) gene panel for HAPS construction, and [2] a blood-based application. Finally, we established a neural network (NN) integrating tumor neoantigen production, presentation, and recognition in pursuit of superior immunotherapy response prediction.

## Results

### HLA-based neoantigen presentation score

The degree of divergence in heterozygous *HLA-I* pairwise alleles may affect tumor neoantigen presentation (which may differ across *HLA-A, -B,* and *-C*). We validated the results of an identification conducted within a large pooled population: 1,125 patients with whole-exome sequencing (WES) data across seven cancer types; 792 patients treated with anti-PD-(L)1 therapy in 10 independent cohorts and 333 patients from The Cancer Genome Atlas (TCGA) database (Supplementary Data 1). Each individual's general *HLA-I* allele divergence was calculated as the average value of each *HLA-I* subtype's HLA divergence between paired alleles (by Grantham distance). The number of predicted neoantigens based on WES data and *HLA-I* genotypes was significantly higher for heterozygous compared to homozygous *HLA-I* [median $\log_{10}$(number of neoantigens+1) = 2.50 and 2.28, respectively, $P < 0.001$] (Supplementary Fig. 1). The divergence of *HLA-I* allele pairs varied across *HLA-I* subtypes. *HLA-B* presented the highest pairwise divergence (mean = 7.56), and *HLA-C* showed the lowest mean divergence (4.38; Fig. 1A), consistent with a previous report[14].

Neoantigen distribution based on MHC-I binding affinities differed from *HLA-I* divergence. The number of predicted neoantigens (according to WES data and MHC-I binding affinity) was comparable between *HLA-B* and *HLA-C* [median, $\log_{10}$(number of neoantigens+1) = 1.80, $P = 2.34e-03$; Fig. 1A], with both predicted values smaller than that for *HLA-A* [mean $\log_{10}$(number of neoantigens+1) = 2.05, $P < 0.001$]. These results reinforce evidence indicating that *HLA-I* genotypes exhibit variable capacity for neoantigen presentation, thereby highlighting the necessity to integrate HLA divergence and neoantigen presentation for ICI response prediction.

By integrating divergence in paired *HLA-I* alleles and the numbers of neoantigens presented, we devised a score for evaluating *HLA-I* with regard to neoantigen presentation (Fig. 1B, Supplementary Fig. 2). The HLA tumor-Antigen Presentation Score (HAPS) was defined as the average value of HLAi allele divergence × $\log10$(TNBi+1; i = A, B, or C). Distribution of the HAPS was generally distinct based on HLA divergence and tumor neoantigen burden (TNB) across *HLA-I* subtypes. The median HAPS derived from *HLA-A* was approximately equal to that derived from *HLA-B* (median = 13.38 and 13.32, respectively, $P = 0.85$), both of which were higher than for *HLA-C* (median = 7.74, $P < 0.001$; Fig. 1C, Supplementary Fig. 3, Supplementary Data 1).

To determine whether HAPS identifies patients likely to benefit from ICIs, we utilized the Wang-WES cohort ($n = 30$) as a Chinese training set (Training Set 1) and calculated series hazard ratios (HRs) for high vs. low HAPS according to continuous cut-off points. For optimal cutoff determination, we analyzed HAPS as a continuous variable for HR curve. Gradual decline was observed in HR for OS with the rise of HAPS, the accumulated HR curve indicated 10 as the optimal cut-off value, with almost the lowest HR (0.316, $P = 0.029$; Fig. 1D). This was close to the median HAPS (10.86) in all enrolled ICI-treated patients ($n = 792$; Supplementary Fig. 4). In the Wang-WES cohort, patients with a high HAPS demonstrated a significantly longer overall survival (OS) than those with a low HAPS [median 29.1 vs. 14.6 months, HR 0.39, 95% confidence interval (CI) 0.16−0.91 months, log-rank $P = 0.050$: Fig. 1D].

To further validate our cut-off point, we used the Rizvi 2015 cohort as an independent training set (Training Set 2), obtaining a similar optimal cut-off point of 10 (with the lowest HR of 0.256, $P = 0.016$; Fig. 1E). In the Rizvi cohort, prolonged OS was observed in patients with high HAPS (median 40.2 vs. 16.6 months, HR 0.26, 95% CI 0.1−0.62 months, log-rank $P = 0.016$; Fig. 1E). Thus, we selected 10 as a universally applicable cut-off value to determine a high or low HAPS.

To assess the immunogenicity (quality) of predicted neoantigens, we employed a model based on the antigenic distance required for a neoantigen to differentially bind to the HLA or activate a T cell compared with its wild-type peptide[19]. The results showed neoantigens in the high HAPS group were more immunogenic ($\log_{10}$(neoantigen quality+1) = 0.68 in high HAPS vs. 0.21 in low high HAPS, $P < 0.001$; Fig. 1F). To corroborate this finding, we designed and synthesized peptides corresponding to predicted neoantigens and their corresponding wild-type (WT) sequences from eight patients with available PBMCs (4 with high HAPS and 4 with low HAPS) (Supplementary Data 2) and stimulated T cells in vitro. We subsequently assessed the presence of 4-1BB on CD8 T cells using flow cytometry, with its expression serving as an indicator of recent T cell activation upon interaction with its corresponding antigen[20,21]. Importantly, higher 4-1BB expression was detected on CD8+ lymphocytes stimulated by predicted neoantigens in the high HAPS group than in the low HAPS group (2.51 times in high HAPS vs. 1.34 in low HAPS, $P = 0.0025$, Supplementary Fig. 5A, C). Moreover, the proportion of predicted neoantigens that led to a significantly up-regulated expression of 4-1BB on CD8 + T cells was higher in the high HAPS group, though statistical significance was not attained (33.3% vs. 17.2%, $P = 0.171$, Supplementary Fig. 5B).

We investigated an independently published pan-cancer cohort of 249 patients treated with anti-PD-(L)1[22] (Validation Set 1, $n = 249$, median 13.7 vs. 9.0 months, HR 0.61, 95% CI 0.41−0.91, log-rank $P = 0.006$; Fig. 2A) and 404 patients from seven other cohorts (Validation Set 2, $n = 404$, median 20.2 vs. 14.0 months; HR 0.75, 95% CI 0.58−0.98, log-rank $P = 0.028$; Fig. 2B, Supplementary Fig. 6 and Supplementary Data 3). To assess the robustness of HAPS, we performed NetMHCpan4.1 using 1% rank instead of IC50 and found no difference survival benefit from immunotherapy between the two methods (Supplementary Fig. 7).

In a pooled population of 717 ICI-treated patients with multi-parameter data, we performed univariate and multivariate Cox regression analysis to verify whether HAPS was an independent factor

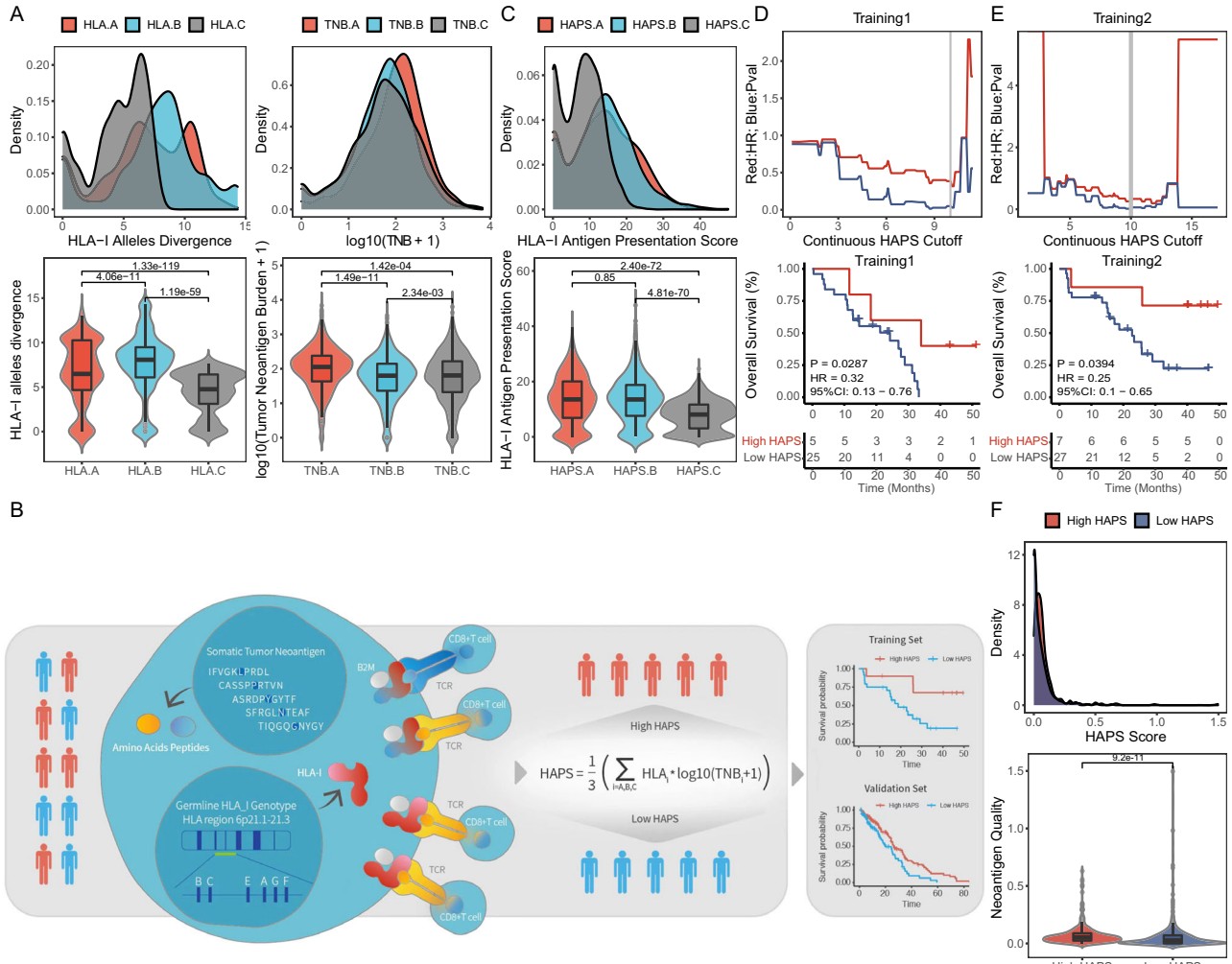

**Fig. 1 | Definition and landscape of *HLA-I* Antigen Presentation Score (HAPS).**
**A** Distribution of *HLA* allele divergence (left) and log10(TNB + 1) (right) for each *HLA* genotype in 1125 patients with whole-exome sequencing (WES) data across seven cancer types [11]immune checkpoint inhibitor (ICI)-treated cohorts (*n* = 792) and The Cancer Genome Atlas database (*n* = 333) (two-tailed *t* test). In the box plots, center line corresponds to median, box boundaries correspond to the first and third quartiles, the upper whisker is max and lower whisker is min. Source data are provided as a Source Data file. **B** Schematic of the HAPS design (HAPS = the average value of HLAi allele divergence × log10(TNBi+1; i = A, B, and C)). Patients were divided into high and low HAPS subgroups to predict ICI treatment outcomes. **C** Distribution of HAPS for each genotype (*n* = 1125) (two-tailed *t* test). In the box

plots, center line corresponds to median, box boundaries correspond to the first and third quartiles, the upper whisker is max and lower whisker is min. Source data are provided as a Source Data file. **D**, **E** Left: P and hazard ratio values following continuous HAPS cut-offs in the Training 1 and 2 cohorts. Right: Association of HAPS with Overall survival in Training 1 and 2 cohorts (Kaplan–Meier analysis with the log-rank test). Source data are provided as a Source Data file. **F** Distribution of neoantigen quality in different groups (*n* = 717). In the box plots, center line corresponds to median, box boundaries correspond to the first and third quartiles, the upper whisker is max and lower whisker is min. Source data are provided as a Source Data file.

affecting patient OS. In the univariable Cox proportional hazards regression model, higher HAPS was associated with improved OS (HAPS: HR 0.69, 95% CI 0.57–0.84, *P* < 0.001; Supplementary Fig. 8). In the multivariate Cox proportional hazards regression model adjusted for stage, gender (male vs. female), age (>65 vs. ≤65 years), PD-L1 expression (≥1% vs.<1%), TMB (>14 vs. ≤14 mutants/exome), and HAPS (high vs. low), the association between HAPS and OS remained significant (HR 0.66; 95% CI 0.50–0.86, *P* = 0.002; Fig. 2C; Supplementary Fig. 9). A similar beneficial trend with regard to progression-free survival (PFS) was observed in patients with PFS data (*n* = 488, median 4.1 vs. 3.2 months, HR 0.76, 95% CI 0.59–0.98, log-rank *P* = 0.025; Fig. 2D). Notably, the HAPS did not robustly predict survival in the TCGA cohort (*n* = 324, median 24.5 vs. 25.5 months, HR 0.72, 95% CI 0.49–1.05, log-rank *P* = 0.066; Fig. 2E), suggesting that it is predictive of ICI response rather than general prognosis. By analyzing the predictive value for ICI response of HAPS in major cancer types such as NSCLC and SKCM, we

found that it was superior to TNB and HLA divergence alone (Supplementary Fig. 10). A positive correlation was observed between the HAPS and durable clinical benefit (DCB) following immunotherapy [*n* = 562, DCB vs. non-durable benefit (NDB): mean 12.67 vs. 10.77, *P* < 0.001; Fig. 2F]. Based on a threshold of 10, patients with a higher HAPS had a higher overall response rate (ORR) than those with a lower HAPS (33.2% vs. 19.4%, *P* < 0.001; Fig. 2G and Supplementary Fig. 11). Taken together, data regarding the positive correlations between HAPS, efficacy, and survival outcomes indicated that HAPS may stratify patients likely to benefit from ICIs.

We then explored correlations between HAPS and established ICI biomarkers, obtaining a significant positive correlation between HAPS and TMB (*r* = 0.454, *P* < 0.001; Supplementary Fig. 12), as well as no correlation between HAPS and PD-L1 (*r* = −0.113, *P* = 0.261; Supplementary Fig. 12). These results further support the notion that the HAPS may represent cancer cell neoantigen presentation ability.

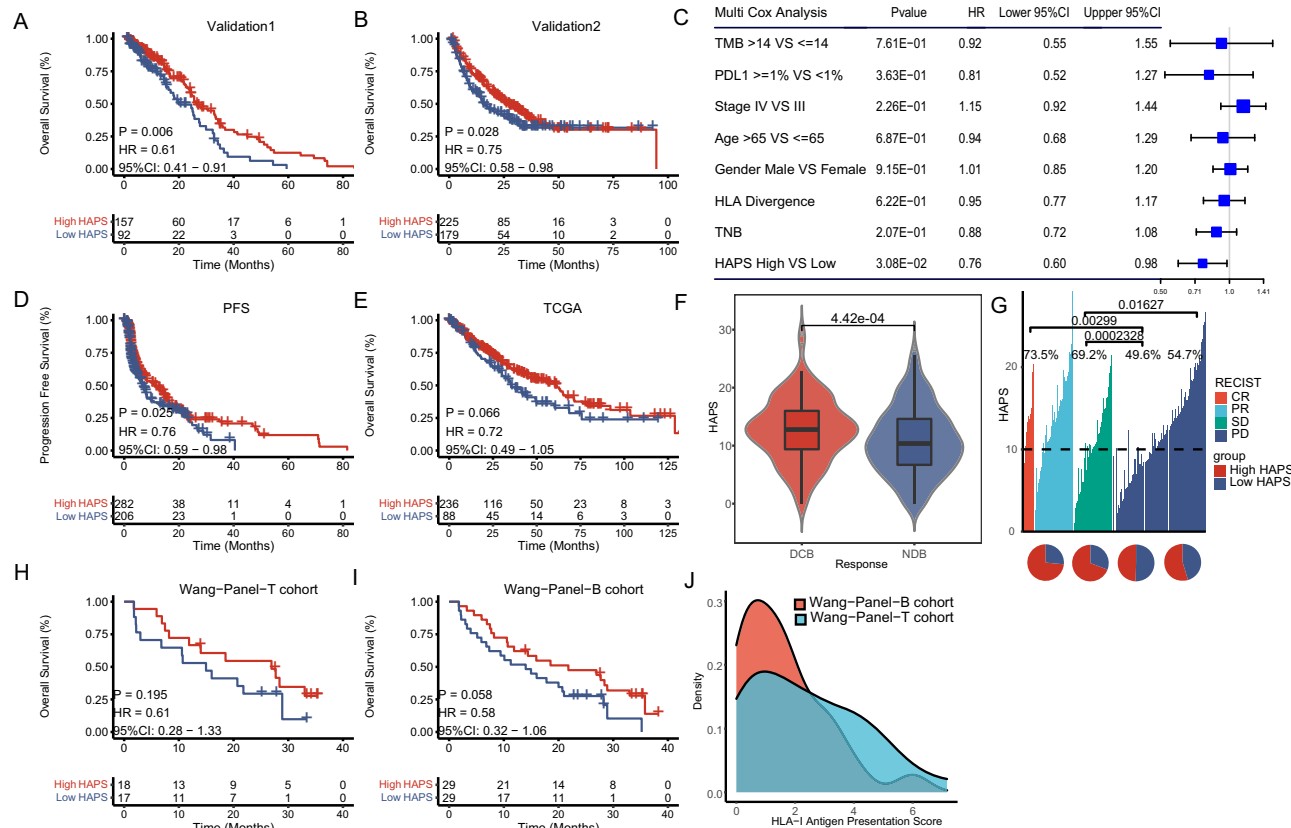

**Fig. 2 | Efficacy of *HLA-I* Antigen Presentation Score (HAPS) in predicting immune checkpoint inhibitor (ICI) outcomes. A, B** Significant association between high HAPS and improved overall survival (OS) after receiving ICIs in the Validation 1 and 2 sets (Kaplan–Meier analysis with the log-rank test). Source data are provided as a Source Data file. **C** Significantly prolonged OS in the high HAPS subgroup (by multivariate Cox analysis) ($n = 717$). The error bars indicate 95%CI for HR. Source data are provided as a Source Data file. **D** Significant association between high HAPS and improved progression-free survival after receiving ICIs in five ICI-treated cohorts (Kaplan–Meier analysis with the log-rank test). Source data are provided as a Source Data file. **E** No significant difference in OS between high and low HAPS subgroups in The Cancer Genome Atlas database (Kaplan–Meier analysis with the log-rank test). Source data are provided as a Source Data file.

**F** Significant difference in HAPS between durable clinical benefit and non-durable benefit subgroups using the Wilcoxon test and Fisher's exact test (two tailed) ($N = 562$). In the box plots, center line corresponds to median, box boundaries correspond to the first and third quartiles, the upper whisker is max and lower whisker is min. Source data are provided as a Source Data file. **G** Distribution of HAPS in complete response, partial response, stable disease, and progressive disease subgroups. Source data are provided as a Source Data file. **H, I** Association between high HAPS and improved OS after receiving ICIs in the Wang Cohort (Wang-Tissue, $n = 35$; Wang-Blood, $n = 58$; samples were sequenced using a 1021-gene panel) (Kaplan–Meier analysis with the log-rank test). Source data are provided as a Source Data file. **J** Distribution of tissue and blood panel-based HAPS.

Considering the wide clinical use of NGS panels and blood-based TMB estimation (compared with WES), we used a broad-targeted NGS-based 1021-gene panel to validate and expand on results derived from the WES-based HAPS (Supplementary Data 4). We first compared the performance of the 1021-gene panel in TMB and TNB estimation using the TCGA cohort ($n = 333$), observing high consistency with WES (TMB: $r = 0.969$, TNB: $r = 0.925$, $P < 0.001$; Supplementary Fig. 13). The 1.31 threshold for panel-based HAPS was utilized according to the lowest HR (0.581, $P = 0.058$) in an independent ICI-treated non-small cell lung cancer (NSCLC) cohort (Wang-Panel-T cohort, $n = 35$). We also evaluated the panel-based HAPS for predicting ICI benefit in the Wang-Panel-T cohort. Patients with a high panel-based HAPS showed prolonged OS and PFS compared with those with a low HAPS (median 22.7 vs. 14.9 months, HR 0.61, 95% CI 0.28–1.33, log-rank $P = 0.195$; Fig. 2H). Similar results were obtained when the blood-based 1021-gene panel was used for HAPS calculation (median 18.5 vs. 14.0 months, HR 0.58, 95% CI 0.32–1.06, log-rank $P = 0.058$; Fig. 2I) in another ICI-treated NSCLC cohort (Wang-Panel-B cohort, $n = 58$). Distributions were comparable between tissue and blood panel-based HAPS (median 2.06, range 0–7.17 in tissue; median 1.29, range 0–6.14 in blood; Fig. 2J). Consistent results across WES-based, panel-based, and blood-panel-

based HAPS highlight the potential utility of HAPS for broader clinical application.

## Combining HAPS and HLA-LOH allows for better patient stratification

Recently, LOH at the *HLA-I* locus (HLA$^{LOH}$) was identified as an important factor underlying immune escape[15] and immunotherapy resistance[23]. Herein, we analyzed the prevalence of HLA$^{LOH}$ in nine ICI-treated cohorts with available HLA$^{LOH}$ data, which failed to show differential distributions between the high and low HAPS subgroups (20.6% in high HAPS vs. 18.5 in low HAPS, $P = 0.596$; Fig. 3A, Supplementary Data 5). This suggested independence between HAPS and HLA$^{LOH}$, in turn supporting their combined use. We also investigated whether HLA$^{LOH}$ combined with HAPS has additional predictive value for ICI response. First, we stratified patients based on the occurrence of HLA$^{LOH}$. In the HLA$^{intact}$ group, patients with a high (vs. low) WES-based HAPS had a significantly longer OS (median 20.0 vs. 10.4 months, HR 0.62, 95% CI 0.48–0.80, log-rank $P < 0.001$; Fig. 3B). However, no significant difference in OS was observed between those with a high and low WES-based HAPS among patients with HLA$^{LOH}$ data (median 11.6 vs. 15.1 months, HR 0.87, 95% CI 0.54–1.40, log-rank $P = 0.555$; Fig. 3B). After resorting patients into four or two categories

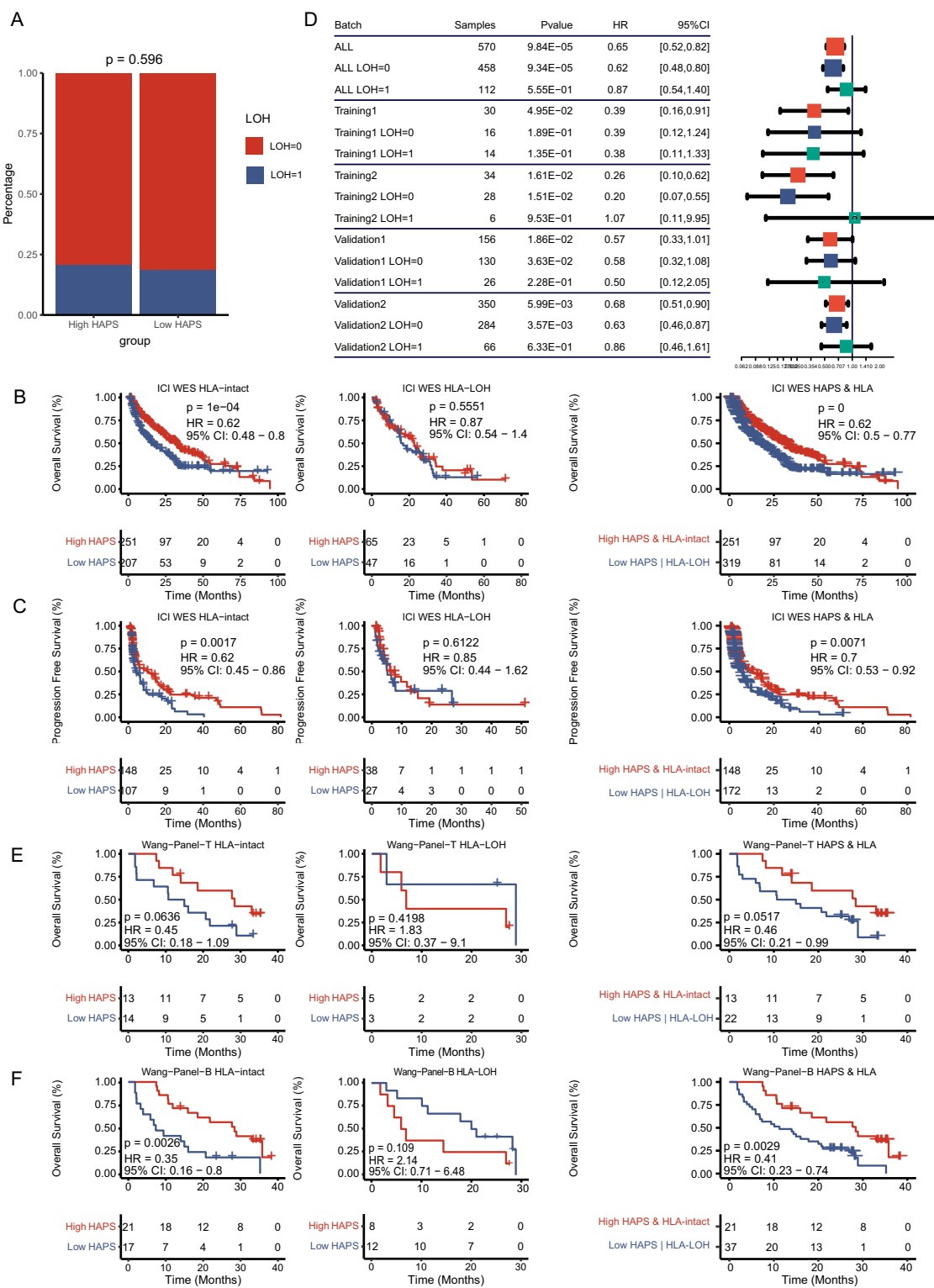

**Fig. 3 | Stratification by loss of heterozygosity (LOH). A** No significant difference in HLA-LOH prevalence between high and low *HLA-I* Antigen Presentation Score (HAPS) subgroups in nine immune checkpoint inhibitor (ICI)-treated cohorts (*n* = 616) (Fisher's exact test). Source data are provided as a Source Dat a file. **B, C** Left: We found a significant association between high HAPS and overall survival (OS) in patients with no HLA-LOH, with no association between high-HAPS and OS in patients with HLA-LOH. Right: We found a significant prolonged OS in patients with high HAPS and no LOH based on HAPS and HLA-LOH status (Kaplan–Meier analysis with the log-rank test). Source data are provided as a Source Data file.

**D** Forest plot of OS differences between high and low HAPS subgroups according to the HLA-LOH status in training and validation cohorts (*n* = 320) (Kaplan–Meier analysis with the log-rank test). The error bars indicate 95%CI for HR. Source data are provided as a Source Data file. **E, F** Left: In the Wang cohort (Wang-Tissue or Wang-Blood), we found associations between high HAPS and OS in patients based on HLA-LOH status. Right: We found prolonged OS in patients with high HAPS and no LOH based on HAPS and HLA-LOH status (Kaplan–Meier analysis with the log-rank test). Source data are provided as a Source Data file.

according to HAPS and HLA[LOH] categorization, we found that those with concurrently high WES-based HAPS and intact *HLA-I* (HAPS[high]/HLA[intact]) had the longest median OS compared to the other subgroups (HAPS[high]/HLA[LOH], HAPS[low]/HLA[intact], and HAPS[low]/HLA[LOH]: HR 0.68; 95% CI 0.47–1.0, log-rank *P* < 0.001; Fig. 3B), in addition to improved outcomes relative to non-HAPS[high]/HLA[intact] patients (HR 0.62; 95% CI 0.50–0.77, log-rank *P* < 0.001; Fig. 3B). Similar results were obtained for PFS (Fig. 3C). A forest plot of HRs for WES-based HAPS in training and validation sets illustrated that a HAPS-predicted survival benefit was mainly observed in HLA[intact] subgroups (5/8 cohorts; Fig. 3D and Supplementary Fig. 14).

We then evaluated the effects of HLA[LOH] on panel-based HAPS patient stratification in the Wang-Panel-T cohort (*n* = 35), obtaining similar results as for the WES-based HAPS. Tissue-panel-HAPS[high]/HLA[intact] group patients showed an optimal median OS compared with other subgroups (panel-HAPS[high]/HLA[LOH], panel-HAPS[low]/HLA[intact], and panel-HAPS[low]/HLA[LOH]: HR 0.37, 95% CI 0.09–1.48, log-rank *P* = 0.187) and a superior result compared to non-panel-HAPS[high]/HLA[intact] group patients (HR 0.46, 95% CI 0.21–0.99, log-rank *P* = 0.052; Fig. 3E). We explored the combined use of blood-panel-based HAPS and HLA[LOH], obtaining almost identical results in the Wang-Panel-B cohort (*n* = 58; Fig. 3F). Therefore, the combined utilization of HAPS and HLA[LOH] could better stratify patients in terms of predicted ICI benefit.

## HAPS correlates with elevated antigen presentation activity

To explore the relationship between HAPS and the TME, we utilized RNA-seq data from TCGA and four cohorts with available expression data[8,24–27]. The TME cell network depicted a comprehensive landscape of tumor–infiltrating immune cells, suggesting greater active intercellular interactions in patients with a high HAPS (Fig. 4A). Previous studies identified gene clusters involved in the active immune response, yielding a cytolytic (CYT) score [defined as the geometric mean of granzyme A (GZMA) and perforin 1 (PRF1) expression levels] and an 18-gene-based gene expression profiling (GEP) score involved in inflammatory T cell gene expression [including interferon-gamma (IFN-gamma)-responsive genes related to antigen presentation, chemokine expression, cytotoxic activity, and adaptive immune resistance][16,28]. Elevated levels of CYT- and GEP-related gene expression were observed in the high (vs. low) HAPS subgroup (Fig. 4B). CYT and GEP scores were higher in patients with high vs. low HAPS (CYT: median 134.17 vs. 86.71, *P* = 0.024; GEP: median 3.09 vs. 2.89, *P* = 0.005; Fig. 4B, Supplementary Data 6).

Next, we performed single-sample gene set enrichment analysis (ssGSEA) to explore correlations between specific immune cell subsets and HAPS distribution (Supplementary Data 7). The infiltration level of Plasmacytoid dendritic cell (pDC) were positively correlated with HAPS (*r* = 0.112, *P* < 0.001; Fig. 4C and Supplementary Fig. 15). Moreover, both the MHC_I and MHC_II immune infiltration signatures assessed via ssGSEA were higher in the high HAPS subgroup (median 0.49 vs. 0.45 for MHC_I, *P* = 0.014; median 0.39 vs. 0.33 for MHC_II, *P* = 0.044; Fig. 4D). These results illustrated that HAPS may reflect antigen presentation capacity.

Accordingly, we speculated that an improved response to ICIs (as indicated by the HAPS) may be associated with adaptive immune suppression. To verify this, we compared immune checkpoint expression levels across HAPS subgroups. Most evaluated checkpoint genes (*CD274*, *CD279*, *PDCD1LG2*, *CTLA4*, and *HAVCR2*) were expressed at higher levels in the high HAPS group (Fig. 4E), indicative of immune escape secondary to adaptive immune activation and a possible benefit from immunotherapy. To further determine the functional pathways associated with HAPS subgroups, we examined differentially expressed genes [DEGs; false discovery rate < 0.05] and extracted 13 highly expressed genes (≥1 fold increase), as well as 12 downregulated genes, in the high HAPS subgroup (Fig. 4F). Additionally, Gene set enrichment analysis (GSEA) of all protein-coding genes revealed that antigen presentation-related pathways were significantly enriched in patients with a high HAPS based on KEGG, REACTOME, and GO terms (Fig. 4G). Collectively, these data suggested that a high HAPS may be associated with the upregulation of antigen presentation pathways.

## HAPS is associated with TCR repertoire characteristics

Efficient T cell recognition of MHC-I-presented tumor neoantigens is critical for immune activation and indicative of ICI response. In the Wang-Panel-B and Wang-Panel-T cohorts, TCR sequencing was performed via NGS for TCR β-chain complementarity-determining regions (CDR3s) within sorted peripheral PD-1 + CD8 + T cells in 52 and 29 patients, respectively. Our previous study demonstrated PD-1 + CD8 + TCR repertoire diversity, as evaluated via the Shannon metric, reflects the probability of neoantigen recognition and predicts ICI response.

In this study, we investigated whether HAPS was associated with TCR diversity, observing no correlation between these indicators (*r* = −0.097, *P* = 0.343; Fig. 5A, Supplementary Data 8). Stratifying according to tissue panel-based HAPS (Wang-Panel-T cohort, *n* = 29) and TCR diversity (cut-off = 3.14, as in our previous study)[17], patients with concurrently high TCR diversity and HAPS had the most prolonged OS (median 26.9 months; Fig. 5B), but this finding was insignificant, possibly due to the limited sample size.

We integrated the blood panel-based HAPS with TCR diversity for further analysis in the Wang-Panel-B cohort, observing that patients with concurrently high TCR diversity and blood-panel based HAPS presented the most prolonged OS (median 27.3 months; Fig. 5C) among stratified subgroups. As the blood panel-based HAPS and TCR diversity can be obtained through peripheral blood analysis, their combined assessment could potentially serve for non-invasive patient stratification for ICI delivery.

We further analyzed the evolution trend with regard to TCR clonality during ICI treatment in 20 patients with longitudinal TCR repertoire data. In the blood panel-based HAPS[high]/TCR diversity[high] subgroup, TCR clonality increased by 0.08 in 50% of the patients (7 out of 14), decreased by 0.17 in the remaining 50% of patients. Conversely, all patients with HAPS[low]&TCR diversity[low] exhibited a consistent downward trend (mean = 0.22; *p* = 0.041; Fig. 5D, Supplementary Fig. 16A). The increment in TCR clonality represents an enrichment of PD-1 + CD8 + T cells, indicative of active anti-tumor immunity[17]. The number of TCR clone reads indicated a more extreme trend of increase in the high HAPS group (post-treatment vs. baseline: 1256 and 716, *P* = 0.30) than that in the low HAPS group (post-treatment vs. baseline: 1134 and 999, *P* = 0.48, Supplementary Fig. 16B), but this finding was not significant (*P* = 0.37, Supplementary Fig. 16C). Cumulatively, the increment in TCR clonality and clone numbers supports the requirement for neoantigen presentation and recognition for an effective immune response.

To illustrate TCR repertoire characteristics corresponding to HAPS and TCR status, we analyzed the edit distance between each TCR sequence, which was calculated based on CDR3 amino acid sequences and could be used to reflect similarity between TCR sequences[29]. In patients with a high HAPS (dynamic cohorts, *n* = 20), the edit distance between the top 30 clones was significantly higher than that in those with a low HAPS (median 8.93 vs. 8.53, *P* < 0.001; Fig. 5E, F), suggesting that greater structural differences between TCRs may imply the potential for recognizing more neoantigens. When stratifying patients by HAPS and TCR diversity, patients with HAPS[high]/TCR diversity[high] exhibited the highest edit distance for PD-1[+]CD8[+]TCR sequences, similar to those with HAPS[high]/TCR diversity[low] (mean 8.99 and 8.84, respectively, *P* = 0.005; Fig. 5G). Both subpopulations presented significantly higher edit distances than HAPS[low]/TCR diversity[high] or HAPS[low]/TCR diversity[low] (mean 8.75 and 8.34, respectively, *P* < 0.001; Fig. 5G).

Stratifying patients according to HAPS and timepoint (pre- or on-ICI treatment), we found that the decrease in the TCR edit distance appeared only in the high (vs. low) HAPS group (median 8.98 vs. 8.87 for pre- and on-treatment in the "high" group, respectively, *P* = 0.035;

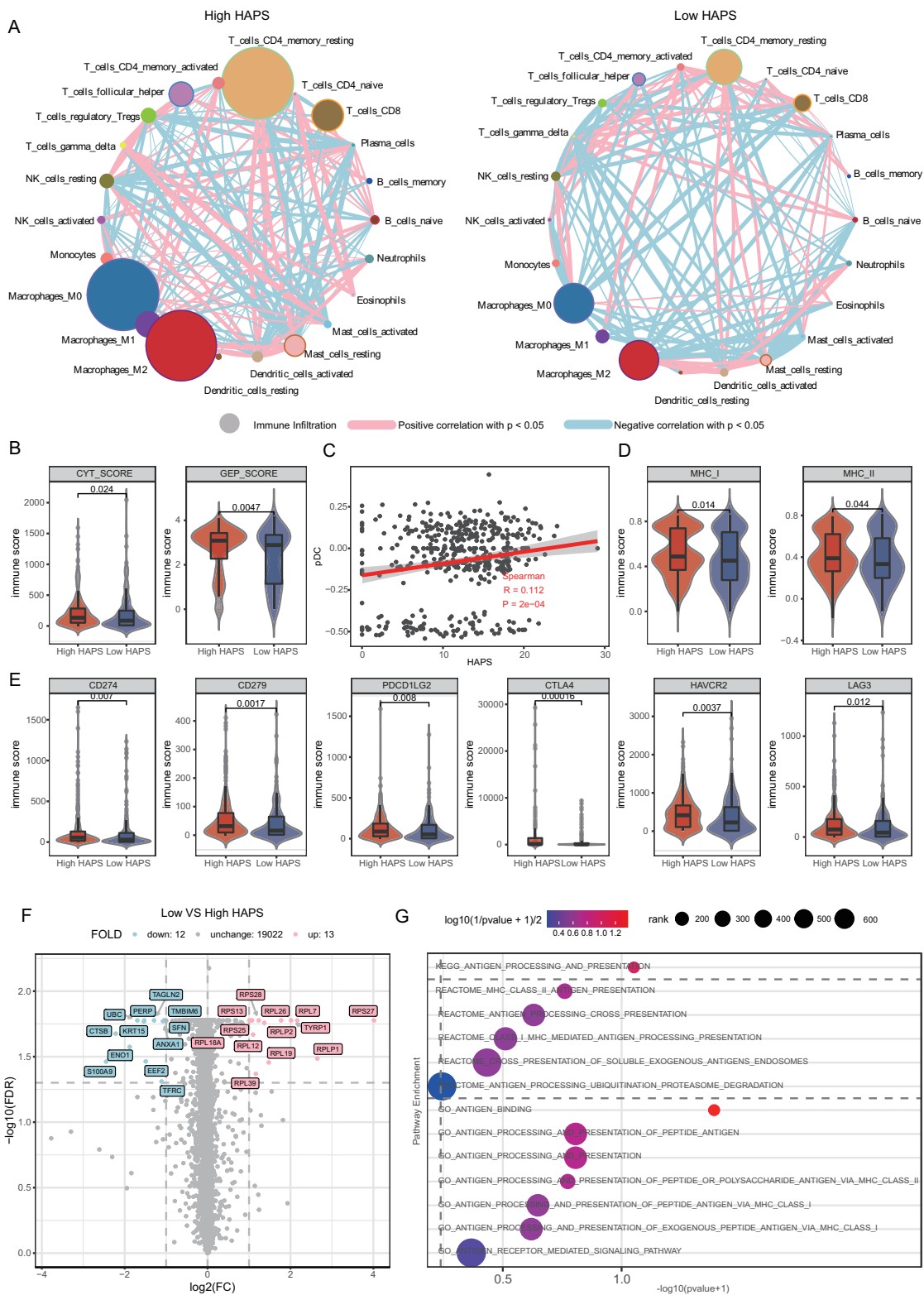

median 8.53 vs. 8.53 for pre- and on-treatment in the "low" group, respectively, $P = 0.859$; Fig. 5H), which may reflect the expansion of similar TCR clonal populations during ICI treatment in the high HAPS group.

## Multi-parameter models for predicting prognosis after ICI therapy

Although the aforementioned analysis is supportive of the potential of HAPS in patient stratification, complexities of the immune response-related process highlight the necessity for combining multiple parameters for ICI response prediction[30]. Due to the small number of cases enrolled in the Wang-Panel-T cohort, we used the Wang-Panel-B cohort for model construction. Multiple linear regression across nine factors (HAPS, TMB, TNB, TCR diversity, TCR clonality, HLA heterozygosity, HLA-LOH, smoking status, and PD-L1) together with the factor-based response [disease control rate (DCR) or PD] to ICI treatment revealed that the predictive value of enrolled factors varied between patients due to inter-individual heterogeneity (Supplementary Fig. 17).

**Fig. 4 | Immune microenvironment differences between *HLA-I* Antigen Presentation Score (HAPS) groups.** This analysis is based on patients from The Cancer Genome Atlas and four immune checkpoint inhibitor (ICI)-treated cohorts with RNA-seq data. **A** Interactions between cell types within the TME. The size of the point represents the level of immune infiltration. The thickness of the line represents the strength of the correlation estimated via Spearman correlation analysis. Source data are provided as a Source Data file. **B** Higher cytolytic (CYT) score (granzyme A, perforin 1) and gene expression profiling (GEP) score in the high vs. low HAPS subgroup ($n = 455$). In the box plots, center line corresponds to median, box boundaries correspond to the first and third quartiles, the upper whisker is max and lower whisker is min. Source data are provided as a Source Data file. **C** Significant positive correlation between plasmacytoid dendritic cell (pDC) infiltration level and HAPS by single sample gene set enrichment analysis (ssGSEA) (two tailed). Shaded area, 95% CI for the correlation. Source data are provided as a Source Data file. **D** Significantly higher major histocompatibility complex I and II binding affinities in the high (vs. low) HAPS subgroup by ssGSEA analysis ($n = 455$). In the box plots, center line corresponds to median, box boundaries correspond to the first and third quartiles, the upper whisker is max and lower whisker is min. Source data are provided as a Source Data file. **E** Significantly higher expression levels of immune checkpoint genes (*CD274*, *CD279*, *PDCD1LG2*, *CTLA4*, *HAVCR2*) in the high HAPS subgroup ($n = 455$). In the box plots, center line corresponds to median, box boundaries correspond to the first and third quartiles, the upper whisker is max and lower whisker is min. Source data are provided as a Source Data file. **F** A total of 15 upregulated and 22 downregulated genes in patients with high HAPS, obtained via DEG analysis. Source data are provided as a Source Data file. **G** Significantly enriched antigen presentation-related pathways in patients with high HAPS using GSEA. Source data are provided as a Source Data file.

We utilized a multi-parameter model to classify patients benefiting from ICIs (Fig. 6A, B) and ranked the nine candidate factors by importance using the "caret" package in R (The R Project for Statistical Computing, Vienna, Austria). The top three were HAPS, TCR diversity, and TMB (Fig. 6C). Goodness of fit was assessed using residual plots during model selection. Consequently, the NN model was determined as optimal (i.e., most appropriate), with the smallest root mean square of the residual (4.352; Fig. 6D). The cut-off value (0.436) was obtained by maximizing the Youden index (Fig. 6E).

To confirm the NN model in predicting ICI benefit, we applied it in 52 patients of the Wang-Panel-B cohort with blood panel-based HAPS, TCR diversity, and TMB values available (Supplementary Fig. 18, Supplementary Data 9). In the training set, patients with high NN scores ($\geq 0.436$, $n = 26$) exhibited a superior PFS (median 5.2 vs. 2.6 months, HR 0.55; 95% CI, 0.31–0.98, log-rank $P = 0.026$; Fig. 6F, Supplementary Fig. 19) and OS (median 21.3 vs. 9.7 months, HR 0.53, 95% CI 0.28 to 0.99, log-rank $P = 0.033$; Fig. 6F) than those with low NN scores ($< 0.436$, $n = 26$). In the validation set (the Wang-Panel-T cohort), a similar result was obtained, as patients with high (vs. low) NN scores showed longer PFS (median 4.7 vs. 2.6 months, HR 0.46, 95% CI 0.21–0.99, log-rank $P = 0.022$; Fig. 6G, Supplementary Fig. 19) and OS (median 24.4 vs. 8.2 months, HR 0.39, 95% CI 0.16–0.93, log-rank $P = 0.021$; Fig. 6G). We verified the predictive performance of the NN model in the pooled panel-based cohorts, with a significantly higher ORR in patients with a high vs. low NN score (31.5% vs. 14.8%, $P = 0.001$; Fig. 6H).

## Discussion

Immunopeptidome diversity and TME complexity highlight the necessity for integrated models of ICI response prediction. Effective neoantigen presentation via MHC-I is essential in bridging neoantigen production and recognition[31]. Herein, we developed a method to quantify pan-cancer *HLA-I* neoantigen presentation capacity by integrating *HLA-I* allele divergence and predicted neoantigens. This study integrates tumor neoantigen production, presentation, and recognition factors, as well as establishing a NN model with predictive value in patient stratification for ICI therapy.

The hypothesis of divergent allele advantage dictates that *HLA-I* genotypes with more divergent sequences at paired alleles enable the presentation of a broader immunopeptidome. Previous and current findings indicate that *HLA-I* allele divergence is distinct across classic *HLA-I* subtypes[14]. However, tumor neoantigen numbers predicted by the affinity of individual *HLA-I* genotypes were found to be distributed across varied *HLA-I* subtypes[32]. Our findings provide a rationale for integrating predicted neoantigens, MHC-I binding affinity, and HLA-I allele divergence.

HAPS presented a superior predictive capability compared with single biomarkers (TNB, HLA divergence) and a decreased HR compared to established ICI-related biomarkers (PD-L1 expression, TMB). Patients with high HAPS tend to have more tumor neoantigens that can be presented by HLA-I. The utilization of immunotherapy has the potential to overcome immune evasion mechanisms activated by immune checkpoints in these patients, thus potentially leading to enhanced therapeutic outcomes and a more favorable prognosis. Furthermore, we quantified the quality of predicted neoantigens and performed in vitro T cell stimulations. Compared with lower HAPS, higher HAPS was more strongly associated with 'high quality' neoantigens. Moreover, we validated NGS panel-based HAPS calculations that could expand the clinical utilization of HAPS, with an optimal gene panel that is yet to be defined. Overall, we provided solid evidence that the HAPS can facilitate patient stratification for ICI therapy.

HLA-LOH contributes to immune evasion and immunotherapy resistance in cancer[23], with studies highlighting HLA-LOH as an indicator for stratifying patients unlikely to benefit from ICIs. Herein, we reported an HLA-LOH prevalence of 17%, similar to a previous study[23]. Although the HAPS score was established for all patients (including those with HLA-LOH), HLA-LOH could further stratify patients when combined with HAPS. As HLA-LOH-intact status has been used to correct the prediction of established ICI-related biomarkers (for example, TMB)[33], the current observations further expand our understanding of HLA in ICI response-predictive model construction. Qualifying circulating tumor DNA (ctDNA)-based HLA-LOH, which would allow for assessing neoantigen presentation non-invasively, remains an open area of research, requiring further inquiry into the availability of ctDNA-based HLA as a surrogate of tissue-based HLA.

Although HAPS takes neoantigen production and presentation into consideration, this score largely reflected immunogenicity, immune cell types, and molecular events within the TME[28]. Herein, a high HAPS was associated with elevated numbers of dendritic cells and immune checkpoint pathways (e.g., PD-1/PD-L1, TIM3, and LAG3). Moreover, TCR similarity (indicated by the edit distance between TCR clones) was significantly correlated with the HAPS, suggesting that the latter might correspond to an enrichment of neoantigen-specific T cells. Therefore, we speculated that HAPS$^{high}$ patients with greater immunogenicity were more likely to present recognizable neoantigens, triggering effector T cell activation and subsequent adaptive upregulation of immune checkpoint pathways. We believe that HAPS could be utilized for immune genotyping, but confirmation via further functional experiments is required.

The ICI-related biomarkers explored (TMB, TNB, HAPS, and TCR repertoire) exhibited promise with regard to identifying patients most likely to benefit from ICIs. However, these data correspond to only one point during the immune cycle[34], and integrating these into a single model would be helpful in comprehensively assessing patient antitumor immunity. Unfortunately, patient cohorts with both genomic and TCR repertoire data are currently very limited.

The Wang cohort corresponds to an independent study, but its sample size is small. We used an NN model to reduce the impact of sample size on model construction, with the TMB, HAPS, and TCR diversity representing neoantigen production, presentation, and

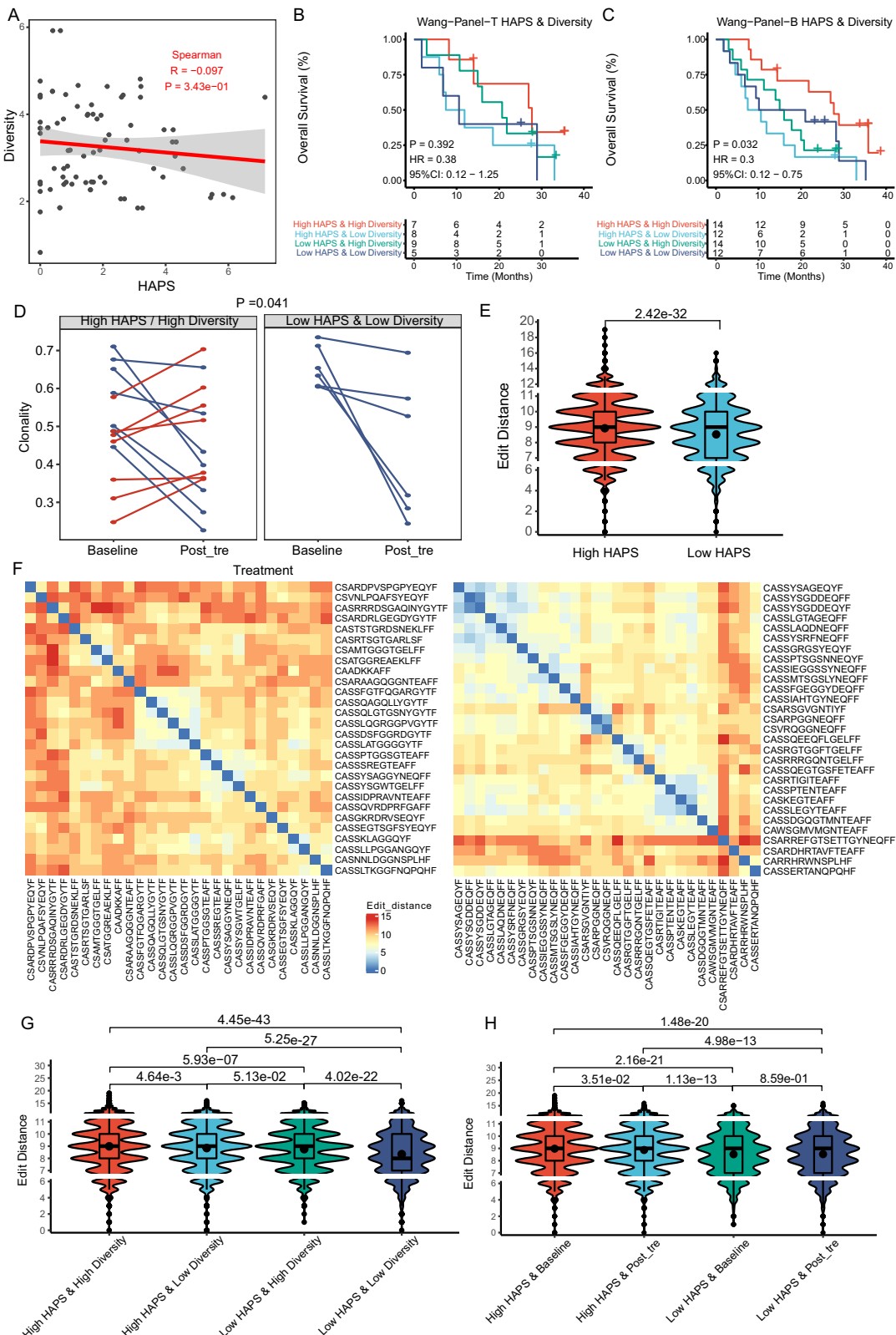

recognition. Considering the complexity of the immune response cycle, this predictive model could make up for deficiencies in a single indicator. Several models predicting ICI efficacy have been developed, including DIRECt-On and IMPRES[30,35]. Together with our model, these exhibit promising predictive value. However, we should note that they were established based on retrospective cohorts, requiring larger validation and/or prospective cohorts in the future.

Limitations of our study include its retrospective nature and the fact that we enrolled pan-cancer patients. It's important to note that HLA divergence isn't solely confined to paired alleles from the same genes. In fact, the divergence may amplify with the increasing breadth, number, and diversity of peptides presented on the cell surface. While the HLA divergence for homozygous HLA-I alleles is zero, these alleles might still present certain tumor antigens. As a result, HAPS could

**Fig. 5 | T cell receptor (TCR) repertoire differences between *HLA-I* Antigen Presentation Score (HAPS) groups.** Analysis is based on patients from the Wang-Tissue and Wang-Blood cohorts. **A** No correlation was found between TCR diversity and HAPS (two tailed). Shaded area, 95% CI for the correlation. Source data are provided as a Source Data file. **B**, **C** In the Wang Cohort (Wang-Tissue or Wang-Blood), the most prolonged progression-free survival/overall survival was found in patients with high HAPS and high TCR diversity (Kaplan–Meier analysis with the log-rank test). Source data are provided as a Source Data file. **D** In the Wang-panel-B cohort, TCR clonality increased by a mean of 0.08 in 50% (7/14) of patients with high HAPS and high TCR diversity, with a downward trend among all patients with low HAPS and low TCR diversity (i.e., by a mean of 0.22) (two tailed Wilcoxon test). Source data are provided as a Source Data file. **E** Comparison of edit distance of top 30 clones between high and low HAPS subgroups (*n* = 20) (two tailed *T* test). In the

box plots, the dots indicate mean values, center line corresponds to median, box boundaries correspond to the first and third quartiles, the upper whisker is max and lower whisker is min. Source data are provided as a Source Data file.
**F** Representative examples of low (left) and high (right) edit distance between the top 30 clones of the PD-1 + CD8 + TCR sequences. Source data are provided as a Source Data file. **G**, **H** Significantly higher edit distance in patients (*n* = 20) with high vs. low HAPS. The highest edit distance was shown in patients with high HAPS and high TCR diversity (similar to those with high HAPS and low TCR diversity). The decrement in TCR edit distance was observed only in the high HAPS subgroup (two tailed *T* test). In the box plots, the dots indicate mean values, center line corresponds to median, box boundaries correspond to the first and third quartiles, the upper whisker is max and lower whisker is min. Source data are provided as a Source Data file.

potentially underestimate the antigen presentation capability of cells that possess homozygous HLA-I alleles. Furthermore, HLA-LOH could occur only in a subset of cancer cells, and there is currently no standard method to predict HLA-LOH. Despite near-consistent trends across cancer types, the potential effects of cancer type-specific manifestations remain. Moreover, algorithms used for neoantigen prediction remain imperfect, with false positive and negative findings inevitably convoluting analyses, and the binding affinity as a read-out for neoantigen presentation need further improvement[36]. Moreover, when applying extra consideration with regard to effective model construction, the number of patients employed for establishing the NN model was relatively small; further validation in larger cohorts and/or in prospective clinical trials is necessary. Lastly, HLA-II neoantigens were not assessed in our study and should be investigated in the future considering the critical role of CD4 T cells in anti-tumor responses.

In conclusion, through integrated analysis of sequencing divergence and neoantigen peptide-binding affinity, we propose a predictive score (HAPS) to be used in evaluating antigen presentation ability with regard to *HLA-I* for the stratification of cancer patients suitable for ICI therapy. The stationary obtained cut-off value at the pan-cancer level reinforces the universal applicability of our method. Although further validation is needed, this multi-parameter model considers the entire anti-tumor immune cycle for response prediction.

## Methods
### Patient cohorts
A total of 885 pan-cancer samples from ICI-treated patients were collected [792 WES; 93 targeted sequencing (1021-gene panel)]. Pre- and post-ICI peripheral blood samples (timepoint of first imaging evaluation, 4−6 weeks after the first ICI administration) were prospectively collected, and PD-1 + CD8 + T cells were isolated through fluorescence-activated cell sorting for TCR sequencing. We enrolled one cohort from the National Cancer Center and nine previously published HLA cohorts of patients with seven cancer types treated via PD-1/PD-L1 or CTLA4 blockade (Supplementary Data 1). Clinical characteristics of these patient cohorts are provided in the original studies[8,9,22,24–27,37–39]. TCGA exome data for patients with lung cancer were obtained directly from TCGA (http://cancergenome.nih.gov). This study was approved by the ethics committees of the National Cancer Center/Cancer Hospital, Chinese Academy of Medical Sciences, and the Peking Union Medical College (NCC2016JZ-03 and NCC2018-092). All enrolled patients provided written informed consent.

### OS and treatment response
OS was defined as the time from treatment initiation to the time of the event (survival or censorship). Response data were available for some cohorts. Clinical benefit was defined as complete response, partial response, or stable disease. "No clinical benefit" was defined as progressive disease. All clinical data, including OS and clinical response data, were obtained from the original studies. Clinical data for TCGA

patients with melanoma and NSCLC were obtained through the TCGA data portal. PFS was defined as the time from ICI treatment initiation until objective disease progression (RECIST version 1.1 criteria) or death from any cause. OS was defined as the time from ICI initiation until death from any cause.

### Whole exome sequencing and analysis
For cohorts subjected to WES, FASTQ reads were aligned to the reference human genome GRCh37 using the Burrows–Wheeler aligner (BWA v.0.7.10)[40]. BWA was employed to align the clean reads to the reference human genome (hg19). Picard (version 1.98, Broad Institute, Cambridge, USA) was used to mark PCR duplicates. Realignment and recalibration were performed by using GATK (version 3.4-46-gbc02625, Broad Institute, Cambridge, MA, USA). Single nucleotide variants (SNVs) were called by using MuTect (version 1.1.4, Broad Institute, Cambridge, MA, USA). Small insertions and deletions (Indels) were called by GATK. Mutations were considered as candidate somatic mutations only when (i) the mutation was detected in at least 5 high-quality reads, (ii) the mutation with a variant allele frequency >0.01, (iii) the mutation was not present in >1% of the population in the 1000 Genomes Project (version phase 3) or dbSNP databases (The Single Nucleotide Polymorphism Database, version dbSNP 137), and (iv) the mutation was absent from a local database of normal samples.

### Targeted genomic sequencing and analysis
Genomic alterations (mutations, insertions, deletions, and amplifications) were detected in ctDNA extracted from plasma samples using a broad-targeted NGS-based 1021-gene panel, which included prevalent tumor-related genes[18]. In hybrid capture procedure, we recruited custom-designed biotinylated oligonucleotide probes (IDT, Coralville, IA, USA) covering 1.09 Mbp of the human genome. TMB was calculated as the number of all non-synonymous mutations per MB of coding regions in the sequenced genome. Coding sequence-specific mutations and small insertions/deletions were identified through analyses of tumor panel data and matched HLA peripheral blood mononuclear cells (PBMCs). We utilized GATK4 pipelines in the Terra cloud platform for somatic mutation detection[41]. Paired-end Illumina reads were aligned to the hg19 human genome reference using the Picard pipeline to yield binary alignment map (BAM) files containing aligned reads (bwa v.0.5.9) with well-calibrated quality scores. Cross-sample contamination was assessed using GATK's Calculate Contamination tool with a 5% threshold. Somatic single-nucleotide variations (SNVs) were called using the MuTect2 algorithm (https://software.broadinstitute.org/gatk/documentation/tooldocs/3.8-0/org_broadinstitute_gatk_tools_walkers_cancer_m2_MuTect2.php). Candidate tumor mutations were removed according to several criteria[1]: more than 10 reads with insertions/deletions in an 11-bp window were centered;[2] the matched DNA derived from PBL control sample carried ≥3% or ≥2% alternate allele reads, and the sum of quality scores was above 80;[3] the candidate was found in several single-nucleotide polymorphism databases

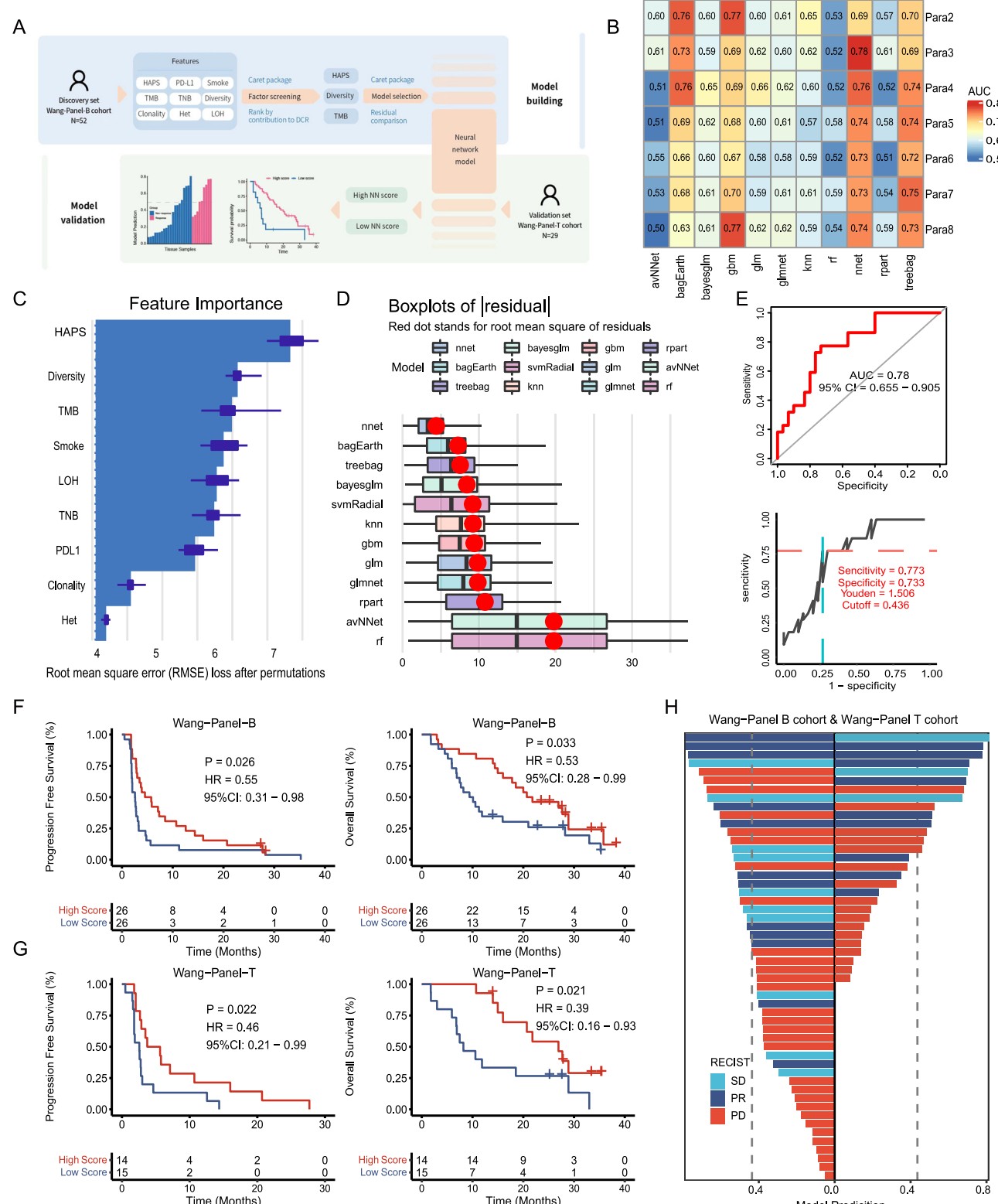

(dbsnp, https://www.ncbi.nlm.nih.gov/projects/SNP/; 1000 G, https://www.1000genomes.org/; ESP6500, https://evs.gs.washington.edu/; ExAC, http://exac.broadinstitute.org/), but not listed in the COSMIC database;[4] the candidate was supported by fewer than five high quality reads (base quality ≥30, mapping quality ≥30); and[5] the allele frequency was less than 1% for genomic DNA. For ctDNA detection, we traced back the mutations from the tumor tissues and applied a tumor-derived strategy. CtDNA mutation was identifed when identical

mutations detected in the tissue that was also found in blood with at least three high-quality reads (base quality ≥30, mapping quality ≥30).Insertions or deletions of small fragments (Indels) were called using MuTect2 with default parameters. Variants were removed if they were detected in matched control samples with three or more reads indicating Indels at the same location or in 40-bp fanking regions of experimental samples or residing near regions with low complexity or short tandem repeats.

**Fig. 6 | Neural network (NN) model combining *HLA-I* Antigen Presentation Score (HAPS), diversity, and tumor mutational burden (TMB). A** Schematic of three-factor NN model development. The Wang-Blood cohort was used as a discovery set, and the Wang-Tissue cohort was used as a validation set. Source data are provided as a Source Data file. **B** Comparison of AUCs for various combinations of models and parameter counts using Caret. **C** Top three factors [HAPS, T cell receptor (TCR) diversity, TMB] ranked by contribution to the disease control rate, as determined via the "caret" package in R ($n = 52$) (The box limits correspond to the first and third quartiles (the 25th and 75th percentiles). **D** The smallest root mean square of the residual was evident in the NN model (goodness of fit was checked using residual plots). (In boxplots, the dots indicate mean values, the center line corresponds to the median and the box limits correspond to the first and third quartiles (the 25th and 75th percentiles). **E** Identification of cutoff value for the NN model. A ROC curve was used to distinguish patients with DCR from PD. **F**, **G** In the training and validation sets, a significantly greater progression-free survival/overall survival was found in patients with high NN scores (Kaplan–Meier analysis with the log-rank test). **H** The correlation between the NN scores and the efficacy of immunotherapy. The x-axis represents the NN scores, and the y-axis shows different patients and their response to ICI. The left side is for the Wang-Panel B cohort, and the right side is for the Wang-Panel T cohort.

## Computational identification of *HLA-I* evolutionary divergence

We probed the evolutionary divergence between HLA-I alleles using a peptide sequence of 181 amino acids in length. To establish a foundation for this, our preliminary analysis of the IMGT database—which encompasses HLA-I A, B, C sequences—revealed that the HLA-I sequence of 181 bp was predominant[42]. Subsequent to this, we conducted a sequence alignment of HLA-I utilizing the widely used MAFFT software (http://mafft.cbrc.jp/alignment/software/). Intriguingly, we discerned that the peptide segment of 181 amino acids, localized within the chr6–exon2,3 peptide-binding domain, encapsulated the salient genetic variations of HLA-I.

To determine the HLA genotypes in the public datasets, we utilized existing data with tools such as Polysolver[22,24], OptiType[37,38], ATHLATES[8,9,27,39], and SOAP-HLA[26]. In the internal Wang cohort, the tool used was Polysolver. For the WES sequencing, the read length employed was 100 bp, and for the panel sequencing, it was 151 bp. Divergences between allele sequences were calculated using the Grantham distance metric[43].

## Computational identification of *HLA-I*-restricted neoantigen

We assessed antigenic potentials of the somatic mutations by calculating binding affinity between mutated epitopes and MHC class I. Binding affinities were calculated using NetMHCpan4.0[36]. MHC Class I/peptide pairs with stronger than moderate binding affinity (IC50 < 500 nM) will be determined as possibly antigenic.

## Evaluation of neoantigen quality

The assessment of neoantigen quality is consistent with previous study[19].

$$D(\mathbf{p}^{WT} \rightarrow \mathbf{p}^{MT}) = (1-w)\log\left(\frac{K_d^{WT}}{K_d^{MT}}\right) + w\log\left(\frac{EC_{50}^{MT}}{EC_{50}^{WT}}\right)$$

D, self discrimination. $P^{WT}$, sequence similarity of the wild-type neopeptide. $P^{MT}$, sequence similarity of the mutant neopeptide. $EC_{50}^{MT}/EC_{50}^{WT}$, TCRs cross-reactivity distance. $K_d^{WT}/K_d^{MT}$, differential MHC presentation. K, w sets the relative weight between the two terms. W, sets the relative weight between the two terms.

## Calculation of patient HAPS

We employed a model by integrating two main determinants: *HLA-I* genotype evolutionary divergence and neoantigen numbers for each genotype. HAPS was defined as the average value of HLAi allele divergence × log10(TNBi+1; i = A, B, or C). The cutoff for HAPS was determined based on the lowest HR for OS, which is 10 in WES cohorts and 1.31 in panel cohorts.

## LOH in HLA

To identify patient LOHHLA, which requires a tumor and germline BAM, patient-specific HLA calls [predicted by an HLA inference tool (e.g., POLYSOLVER, OptiType) or through HLA serotyping][44,45], the HLA FASTA file location, and purity and ploidy were estimated. For implementation of LOHHLA, allele-specific copy number analysis of tumors was used to estimate purity and ploidy, while HLA inference was performed using POLYSOLVER (see below). To call HLA-LOH, LOHHLA relies on five computational steps:[1] extracting HLA reads[2], creating HLA allele-specific BAM files;[3] determining coverage at mismatch positions between homologous HLA alleles[4], obtaining HLA-specific logR and B allele frequency values, and[5] determining HLA haplotype-specific copy number values. A copy number <0.5 is classified as subject to loss and is indicative of LOH. To avoid over-calling LOH, we calculated *P* values related to allelic imbalance for each HLA gene. These *P* values correspond to pairwise differences in logR values at mismatch sites between two HLA homologs and are adjusted to ensure each sequencing read is only counted once. Allelic imbalance is determined if $P < 0.01$ (using paired Student's *t* tests).

## Immune infiltration analysis

The analytical tool CIBERSORT was used for quantifying the percentage of different tumor-infiltrating cell types[46] under a complex "gene signature matrix" based on 547 genes. Overall, 455 patients with available RNA-seq data were included for immune infiltration analysis[8,24–27]. Enrichment of cell type meta-genes was calculated using ssGSEA, previously used to analyze samples for immune/stromal infiltrates and implemented in the "GSVA" R package (with z-scoring across samples). The MHC-I immune infiltration signature included the expression of B2M, TAP1, TAP2, TAPBP, HLA-A, HLA-B, HLA-C, HLA-E, HLA-F, and HLA-G. The MHC-II immune infiltration signature included HLA-DMA, HLA-DMB, HLA-DOA, HLA-DOB, HLA-DPA1, HLA-DPA2, HLA-DPA3, HLA-DPB1, HLA-DPB2, HLA-DQA1, HLA-DQA2, HLA-DQB1, HLA-DQB2, HLA-DRA, HLA-DRB1, HLA-DRB2, HLA-DRB3, HLA-DRB4, HLA-DRB5, HLA-DRB6, HLA-DRB7, HLA-DRB8 and HLA-DRB9[47]. The GEP was composed of 18 genes related to antigen presentation, chemokine expression, cytolytic activity, and adaptive immune resistance[28]. We utilized a simple and quantitative measure of immune cytolytic activity ('CYT') based on the transcript levels of two key cytolytic effectors (GZMA, PRF1) dramatically upregulated upon CD8 + T cell activation[16]. Biological process enrichment and KEGG pathway enrichment analysis of DEGs were performed using GSEA (http://software.broadinstitute.org/gsea/msigdb/annotate.jsp).

## TCR β-chain sequencing and analysis

PBMCs were extracted from 10 mL of fresh peripheral blood with anticoagulant through density gradient centrifugation with Lymphoprep (Progen). Using FACS analysis (BD FACSAriaTM), PD-1 + CD8 + T cells were obtained. The specific antibodies used, sourced from eBioscience, included CD3-efluor 450 (OKT3) (catalog: 48-0037-42), CD8-APC (RPA-T8) (catalog: 17-0088-42), and CD279 (PD-1)-PE (MIH4) (catalog: 12-9969-42). CD8-FITC (T8) was purchased from MBL (catalog: K0227-4). 4-1BB-APC (4B4-1) was purchased from biolegend (catalog: 309810). DNA was isolated from the PD-1 + CD8 + T cells using the QIAamp DNA Mini Kit (QIAGEN, catalog: 51306). The DNA yield was ≥1 mg with UV absorption ratios at wavelengths 260/280 and 260/230 being ≥1.8 and 2, respectively. A multiplex PCR approach was applied to amplify the CDR3 in TCRb chain (TRB) both inclusively and semi-quantitatively. This involved a two-round PCR process, with primer sequences listed under Chinese patent CN105087789A[17] (Supplementary Data 10).

During the first PCR round, 10 cycles were conducted to amplify CDR3 sequences using specific primers. The initial round involved a reaction system comprising of 600 ng of template DNA, QIAGEN Multiplex PCR Master Mix, Q solution, and primer set pools using a Multiplex PCR Kit (QIAGEN). Post this, a purification step was conducted with magnetic beads (Agencourt no. A63882, Beckman). All products from this round were used as templates for the second amplification step, adding pooled primers and a Phusion High-Fidelity PCR Kit. This was followed by a cycle program: one cycle at 98 °C for 1 min; subsequently, 25 cycles consisting of denaturation at 98 °C for 20 s, annealing at 65 °C for 30 s, and extension at 72 °C for 30 s; followed by a final extension at 72 °C for 5 min. The amplified products were size-selected via agarose gel electrophoresis, targeting fragments between 200 and 350 bp, purified using the QIAquick Gel Purification Kit (QIAGEN). Finally, paired-end sequencing of these samples was executed using the Illumina HiSeq 3000 platform, achieving a read length of 151 bp. The main quality control steps included the removal of reads bearing adapter sequences at the 5' end or those with over 5% "N" bases. Subsequently, we calculated the average base quality for each read after eliminating low-quality bases (base quality <10) from the 3' end. Further filtration excluded reads with an average quality lower than 15. The sequences of the V, D, and J genes were compared with the ImMunoGeneTics (IMGT) database using the MIXCR software to identify the CDR3 sequence of the TCR. Then, $10^6$ qualified reads per sample were randomly selected for downstream analysis.

TCR repertoire diversity was calculated based on the Shannon−Wiener index (Shannon index), which ranges from 0 to 1: $Shannon\ index\ H = -\sum_{i=0}^{n} p_i \ln p_i$, where pi refers to the frequency of clonotype i for a sample with n unique clonotypes. Clonality was defined as 1 - (Shannon index)/ln(# of productive unique sequences).

### Edit distance between baseline and post-treatment TCR

Edit distances were calculated between pairs of TCRs in 20 enrolled patents. The similarity of TCR sequences (specific to defined antigens) was evaluated. The network of VDJdb records constructed using Hamming distances was computed for pairs of CDR3 amino acid sequences. Edges (alignments) connect sequences that differ by up to three amino acid substitutions[29]. We constructed a heatmap showing the normalized number of alignments between each pair of epitope specificities, wherein the diagonal indicates alignments within the same epitope specificity. Normalization was performed by dividing each entry of the alignment count matrix by the product of corresponding row and column sums. The scoring is described in the database specification (https://github.com/antigenomics/vdjdb-db/blob/master/README.md).

### In vitro T cell stimulation

PBMCs were collected from the patient peripheral blood samples. The total cells were cultured in AIM V serum free medium (Gibco, Grand Island, NY) and allowed to adhere for 4 h. The adherent cells were cultured using DC serum-free medium (CellGenix, Germany) containing 1000 μ/ml IL-4 and 500 μ/ml granulocyte-macrophage colony-stimulating factor (GM-CSF). Following next 6 days for facilitating DC growth, another 10 ng/ml of TNF-a was added to the DCs to induce maturation on the 6th day. The non-adherent cells were collected at the first day and induced to become T cells followed by stimulating 50 ng/ml anti-CD3 antibody (Ab) and 1000 μ/ml recombinant human interferon-γ (Peprotech, USA) at 37 °C with 5% CO2 for 24 h. Then, 1000 μ/ml recombinant human IL-2 (Peprotech) was added to the medium. IL-2-containing medium was added to the culture system every 2 days.

For T cell activation, after 7 days of T cell culturing, the T cells were mixed with DCs loaded with 10 mM peptide (synthesized by GL Biochem, Shanghai, Ltd.) at ratio of 20:1 and co-cultured using conditioned medium supplemented with 1000 μ/ml IL-2, 500 μ/ml IL-4,

250 μ/ml GM-CSF and 10 ng/ml TNFα (Peprotech) for another 7 days. The T cells were harvested and analyzed by flow cytometric analysis.

### Neural Network probit model establishment and verification

To classify DCR vs. PD, we utilized an NN probit approach. In brief, patients were split into plasma training (*n* = 52) and tissue validation (*n* = 29) cohorts. Features associated with ultimate outcomes were identified in the training cohort. Scoring for human antigen presentation, TMB, and T lymphocyte distribution are independent and can be used as variable factors. Among caret methods (https://topepo.github.io/caret/index.html), the Neural Network (NN) was chosen due to the smallest root mean square of residuals in model selection. We employed the R package "neuralnet" to construct a neural network model. This model used HAPS, TMB, and Diversity as input layers, with Response designated as the output layer. The configuration was designed with a singular hidden layer and a non-linear output, leveraging the RPROP algorithm and the "sse" error function to assess error rates. Within this network, neurons were interlinked, and the strength of these connections was characterized by node link weights. The model training underwent 20 iterations and concluded when the absolute change in the error function fell below 0.01. Error assessments were based on the AIC criterion. The intercepts for the hidden neurons were found to be 1.9214, with weight predictions being 1.54006, 1.4383, and 2.46875. The output layer showcased intercepts of −0.42557 and −1.76532. The models were trained in the training cohort, and DCR vs. PD was stratified based on the best threshold (Youden's J). When the Youden index reached its maxima, the NN prediction was selected as the cut-off for all enrolled patients. The model and threshold were then applied to the tissue validation cohorts.

### Statistical analyses

All statistical analyses were performed using R version 4.0.3 (http://www.r-project.org). Continuous variables are expressed as the mean (±standard deviation) and were compared using Wilcoxon signed rank test (paired). For data that follows a normal distribution, we used the *T* test. Categorical variables were compared using the χ2 or two-tailed Fisher's exact test. Spearman correlation coefficient was used to evaluate the correlation between two variables. Kaplan−Meier analysis with the log-rank test was used to compare PFS and OS. The Youden index (J=sensitivity+specificity-1), a measure of overall diagnostic effectiveness, was used in conjunction with ROC analysis. DEGs between high and low HAPS groups were identified using the DESeq function in edgeR. An adjusted *p* value < 0.05 and absolute $\log_2$foldchange > 1 were set as the thresholds for significant differential expression of genes. All reported *P* values are two-tailed. *P* < 0.05 indicates statistical significance.

### Reporting summary

Further information on research design is available in the Nature Portfolio Reporting Summary linked to this article.

## Data availability

All data needed to evaluate the conclusions of the current study are present in the manuscript and/or the Supplementary Information. The data associated with this paper are also available at https://github.com/zxl2014swjx/HAPS.git. Whole-exome sequencing, T-cell repertoire sequencing and panel sequencing data have been deposited in Genome Sequence Archive under accession code PRJCA018167. All GSA data from the study are under controlled access are available from Dr. Zhijie Wang (wangzj@cicams.ac.cn) upon request. Applicants should have obtained ethical approvals from their ethics committees and submitted a research proposal for the data request. Timescale for access to be granted would be around one month, and there are no restrictions on the duration of access. Source data are provided with this paper.

## Code availability

The source code associated with this paper are publicly available at https://github.com/zxl2014swjx/HAPS.git.

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

## Acknowledgements

We would like to thank the Geneplus-Beijing Institute, Beijing, China for the support, as well as all the patients. We thank Editage (www.editage.cn) for English language editing. Support for the study was provided by National key research and development project (2022YFC2505004, 2022YFC2505000 to Z.W. and J.W.), CAMS Innovation Fund for Medical Sciences (2021-1-I2M-012 to Z.W.), CAMS Innovation Fund for Medical Sciences (CIFMS 2022-I2M-1-009 to J.W.), CAMS Key lab of translational research on lung cancer (2018PT31035 to J.W.), National Natural Sciences Foundation of China (82272796 and 82241229 to J.W.; 81871889 and 82072586 to Z.W.; 82003275 to J.H.), Beijing Natural Science Foundation (7212084 to Z.W.), Aiyou Foundation (KY201701 to J.W.), Beijing Municipal Administration of Hospitals' Youth Program (QML20210504 to J.H.), National Youth Talent (to Z.W.), and Beijing Natural Science Foundation (7214249 to R.W.).

## Author contributions

Conceptualization: J.W. and Z.W. Methodology: J.H., X.Z., J.B. and C.C. Investigation: J.H., J.X., H.B., J.D., R.W., J.Zhao., J.W. and Z.W. Visualization: J.H., Y.D., X.Z., J.B., X.X. X.Y. and Z.W. Funding acquisition: J.H., R.W., J.W. and Z.W. Project administration: J.W. and Z.W. Supervision: J.W. and Z.W. Writing – original draft: J.H., Y.D. and X.Z. Writing – review & editing: A.R., J.Zhang., C.C., J.W. and Z.W.

## Competing interests

The authors declare no competing interests.
