## [Peer Review File · Nature Communications]

Assessment of human leukocyte antigen-based neoantigen presentation to determine pan-cancer response to immunotherapyREVIEWER COMMENTS

Reviewer #1 (Remarks to the Author): expertise in TCR-seq and WES bioinformatics

Han et al. present "Assessment of human leukocyte antigen-based neoantigen presentation to determine pan-cancer response to immunotherapy," their manuscript describing a score (HLA tumor-Antigen Presentation Score [HAPS]) used to stratify patients more likely to benefit from ICI therapy. The manuscript presents extensive correlative data. Results are presented that show HAPS may have some advantages over established markers (HLA-LOH, TMB, PD-L1, etc..) but it is not clear that HAPS is robust enough to inform clinical decision making, which is a central claim. Many methodological details and justifications are missing. Providing clarification on the points below would strengthen the paper.

- Overall, figures and their text are undersized. Even viewed on the computer at 200% zoom, they were hard to read. Many figures lacked sufficient explanation in the figure legend or text to be easily interpretable.
- The GitHub code repository (<https://github.com/zxl2014swjx/HAPS>) appears to only contain code to generate figures, and not the actual analysis (calculation of HAPS score, TCR annotation, neoantigen predictions, differential gene expression analysis, gene set enrichment analysis, CIBERSORT, etc). The entirety of the README is: "The codes and files are for the study of HAPS (Human Antigen Presentation Score)." This is insufficiently descriptive.
- Line 104: Is the entire HLA sequence being compared? Just the binding groove? What is the justification for the choice?
- Line 104: Define what a predicted neoantigen is (mutation that can yield a peptide capable of binding an HLA? A peptide-MHC pair?) How is this counted – one per mutation, one per peptide, one per peptide-HLA pair? How are the predictions done?
- Line 106-107: Why is the log10 of number of neoantigens used? Is there biological relevance for the neoantigen burden on a log scale compared to a linear scale? What statistical test was done? The distributions appear extremely overlapping – is the statistical difference biologically relevant?
- Line 107-108: Was divergence calculated for all possible HLA pairs, or naturally occurring within patients (in your cohorts)?
- Line 110: What binding affinity threshold?
- Line 116-117: Given $HAPS = HLA \text{ divergence} \times \log_{10}(TNB)$, and divergence is a linear scale while TNB is used on a log scale, divergence will have a stronger impact on HAPS. Does divergence alone predict ICI response? Figure S6 seems to suggest so, where the survival curves for HLA diversity are very similar to survival curves for HAPS in almost all cohorts. Also, wouldn't TNB be correlated to divergence, since heterozygous HLA loci would have more possible neoantigens and higher divergence? Does the HAPS metric for patients that are homozygous at all loci make biological sense (where the score would be 0, but it is very possible they have bonafide neoantigens)?
- Line 132: How was it "indicated" that 10 was the optimal cut-off? It has "almost" the lowest HR, so by what criterion was it selected?
- Line 136: The HAPS threshold of 10 was selected using the Wang-WES data, and yielded a HR of 0.316, $p=0.029$ (line 132), however in line 136 it is reported that the same data and threshold had HR of 0.39 and $p=0.05$. Why do the HRs not match?
- Line 139: The text states this obtained a "similar" optimal cut-off, but it appears to not be just similar, but actually the same (10).
- Line 147: As with neoantigen counts, what is the justification for using logs here for neoantigen quality?
- Line 149: Were all peptides corresponding to predicted neoantigens synthesized? How were the 8 patients selected?
- Figure S8: What do the size of the blue HR boxes mean? TNB is very large. How does TNB have a HR=1.00 (CI 1.00-1.00) but $p=0.00790$? This plot says neither TNB nor HLA divergence have any significant effect on OS alone (with tight HR CI around 1.0), but they do when combined into HAPS score. However, Figure S6 shows that HLA divergence and TNB may have an effect on OS, not reaching significance, but having wide HR CIs. These results do not seem consistent.
- Line 162-164: This sentence needs context for those readers not intimately familiar with pMHC binding predictions. Also, Figure S7 shows that the test is NetMHCpan4.1 vs NetMHCpan4.0, and doesn't say which generated %rank predictions and which generated IC50 predictions (both

versions can produce both metrics).

- Line 176. It is problematic that HAPS did not predict survival in the TCGA cohort. Support for HAPS is built on predicting OS (Figs 1, 2, 3). It is difficult to reconcile a claim that "HAPS may predict ICI response but not general prognosis" with the rest of the study. If ICI response is of interest, then this should be the main metric assessed, not OS. More validation cohorts should be assessed.
- Line 179-180: "...HAPS...superior to TNB and HLA divergence alone" – Fig S10 shows that in NSCLC, TNB is better than HAPS.
- Figure S12: What are the units for PD_L1, why does it go from 0 to 1?
- Figure S13: Which cohort was used for this, Wang-Panel-T or Wang-Panel-B?
- Figure 2I, 2J, and 2K references are incorrect (Line 203-208).
- Figure 3B: The third column/panel is just the previous two combined/overlaid. Are the previous 2 really needed? It makes it hard to interpret what is different/the same between them.
- Line 189. Why does HAPS correlate with TMB, but TMB is not predictive in the multivariate Cox analysis (Fig 2c)
- Line 198-201: Does the panel-based HAPS threshold of 1.31 make sense compared to the genome-wide based HAPS threshold of 10 when considering the size difference of the 1021 gene panel vs the genome? Is it justified to select the threshold using the Wang-Panel-T cohort and then predict ICI benefit in the same cohort?
- Line 216-217: Could you test whether HAPS and HLA LOH are correlated or not? Likewise, does HLA LOH correlate with HLA diversity? Should/could HLA LOH be taken into account when calculating HLA diversity (if there is HLA LOH, the effective HLA diversity should be 0)?
- Line 246-247: Is there any way to quantify if there is greater intercellular interactions in patients with high HAPS?
- Line 237: Fig 3D ref is wrong.
- Line 253-254, Fig 4B: Distributions are so overlapping, is there any biological relevance to this small statistical difference?
- Line 259: how are the counts defined?
- Line 261-263: is there any biological relevance to the small difference? Can you really conclude that HAPS may reflect antigen presentation capacity from this?
- Line 266. If checkpoint genes are expressed more highly in the HAPS group, could this alone explain and association between HAPS and ICI response?
- Line 271-272: Is it realistic to expect that there are ~16,000 genes that are differentially expressed between HAPS high and low groups? That is the majority of genes. If this is a true result, why are there three times as many downregulated as upregulated?
- Line 283-284: which "complementarity-determining regions" were used?
- Line 296, Line 308. If a finding is not statistically significant it is not a finding.
- Line 301: The authors state that TCR clonality is increased (by 0.08) over time in half the patients with high HAPS/high diversity, and that it decreases for low HAPS and low diversity. However, the other half of patients in the high/high group showed decreased clonality, apparently with greater magnitude. This is a questionable description of the data – the high high group likely has no overall change in clonality if a statistical test was done.
- Line 313: what is meant by "epitope-specific amino acid residues"? Are these just the amino acids comprising the CDR3, or something else?
- Line 316: There is a small statistical difference here, but the medians are the same. What is the biological relevance of such a small change?
- Figure 5F: This figure needs labels to help guide interpretation.
- Figure 5G, H: As above, can you justify that there is biological relevance to these small differences between completely overlapping distributions?
- Figure 6A: I do not understand what this plot is showing.
- Line 341-342: Were only these top three used for model building? If not, why call them out?
- Line 346-347: This sounds like you applied the NN model to the same patients that were used to train it. Presumably this is not the case.
- Figure 6H: This figure needs more description to be interpretable.
- Line 595: "encoding the neoantigen" I think you mean "MHC".
- Line 597: "amino acid length equal to 181 nucleotides" – do you mean 181 amino acids?
- Line 599: "protein sequences of exons 2 and 3" – is it these exons, or the 181nt/aa section?
- Line 611: How were HLA genotypes determined? Existing data, or analyzed by the authors? What tools were used (by authors or existing data)? If it was NGS-based, what read length(s) were

used?

- Line 612: "of the top recurrent somatic mutations" – Does this mean that neoantigens were not predicted for all mutations in a patient?
- Line 615: Which IEDB algorithms were used?
- Line 617: "when neoepitopes are available" – what does this mean? In general, for neoantigen predictions, were all overlapping peptides tested (ie. All possible 8-11mer peptides with the mutation in every possible position)?
- Line 621-622: This is possibly confusing, as the HAPS threshold for the panel was a different value. Also, what is meant by a "median score" of 10?
- Line 638-641: Unclear what is meant by this.
- Line 655: "we devised" – since this metric was taken from an existing publication, it was already devised.
- Line 661: Significantly more detail is needed on how the TCR-b sequencing was done. How were the cells isolated? How were CDR3 regions selected? Were primers used? Was this from RNA? DNA? What sequence machine was used? What read length? What sequence depth was achieved? Provide a summary of the sequence data obtained. How was the resulting sequencing quality filtered and annotated with TCR features?
- Line 672-674: In Figure 5F, the diagonal is not always the same TCR (and so may not have the same epitope specificity). This is not consistent with the text description at these lines.
- Line 679: I found this section unclear. How were features associated with outcomes? How is it known that human antigen presentation, TMB, and T lymphocyte distributions are independent? What is the architecture of the model?
- Line 698: Is it appropriate to use a non-parametric paired test for all comparisons? Many distributions appear normally distributed, and a T test would perform well. There are also many comparisons that do not seem appropriate for a paired test (comparisons of high and low HAPS patients, for instance – what would the pairing be?).

Reviewer #2 (Remarks to the Author): expertise in HLA sequencing and ICI response

The manuscript " Assessment of human leukocyte antigen-1 based neoantigen presentation to determine pan-cancer response to immunotherapy" by Han et al. describes the integration of multiple factors for better prediction of ICI treatment outcomes.

Overall, the study is interesting but it is quite complex and it was a little difficult to follow the rationale throughout. Several essential points need to be addressed for clarity before this manuscript should be considered for publication.

1.

Authors define a novel HLA presentation score (HAPS). The score is based on (i) HLA allele divergence, which has been previously shown to strongly predict ICI treatment outcomes (Ref14), and (ii) the number of predicted neoantigens, also known to predict treatment outcomes (Ref9 and others). HLA divergence should be introduced in more detail.

I did not understand the rationale for combining both factors for patient stratification. Does HLA divergence and number of presented neoantigens correlate overall (as expected and observed in Ref14)? Even if variability for different HLA alleles is observed, it is not clear to me why both factors should be combined for patient stratification. It is in my view therefore essential that the authors show how HAPS compares to the performance of HLA divergence and number of neoantigens separately (on patient cohort level).

2.

A framework is then applied to estimate the immunogenicity of the predicted immunogenicity of neoantigens, in alignment to Ref19. Authors show that patients with high HAPS score present more immunogenic peptides than patients with low HAPS score. If I understand correctly, the HAPS score contains quantitative predicted neoantigen numbers in each patient, so is this creating

a bias and suggest direct correlation between the number of neoantigens, and the number of immunogenic neoantigens?

3.

Could you explain the rationale of measuring 4-1BB on CD8 T cells?

4.

The observation that HAPS predicts ICI response outcomes rather than overall survival is interesting. How do the authors think this relates to observations of tumour evolution and tumour-immune editing? Please discuss.

5.

Immune signatures were defined in the methods section. It seems that MHC-I immune infiltration was simply defined by expression levels of MHC class I alpha chains. This does not give an understanding of immune-infiltration, but of HLA expression levels? MHC class II immune infiltration signatures included MHC class I pathway molecules TAP1, TAP2, TAPBP which is confusing? Did the authors look at expression of immune cell markers etc? Much more detail and discussion needed.

6.

The authors final model needs to be benchmarked to related models to proof superiority.

Minor comments:

To note that I found it a little difficult to follow the paper as the formatting was not consistent, i.e. section headers were not highlighted in bold, and abstract title missing.

In figures, it would be good if axis titles could be reviewed, as they were often not clear, i.e. Figure 1A/C and 2J. What is "the ratio of HLA divergence" etc. Figure D/E: both HR and p-value are given in one graph, but only one axis is shown. Please add p-values, too.

Reviewer #3 (Remarks to the Author): expertise in HLA neoantigen prediction

The authors describe an integrated scoring metric combining tumor neoantigen burden along with HLA allele divergence to develop a combined HLA tumor-Antigen Presentation Score (HAPS). This score is then evaluated in patient cohorts that have undergone immune checkpoint blockade therapy for survival benefit as well as correlation with various other metrics such as gene expression relating to the tumor microenvironment. This is an interesting idea, however I'm not fully convinced of the added value of using HAPS with the currently presented data compared with previous published studies, please see detailed comments below:

Major comments:

The authors suggest that HAPS consists of two elements, tumor neoantigen burden and HLA allele divergence score. Can the authors please clarify details on how tumor neoantigen burden is defined for all sample cohorts, including details of handling cases such as 1) multiple HLA alleles of the patient bind strongly to the same peptide, 2) discrepancies among different prediction algorithms used?

Furthermore, it seems that NetMHCpan4.0 is one of prediction algorithms used, which in itself utilizes a "neighboring" estimation approach for HLA alleles with no prediction data. This estimation approach I believe is based on HLA allele "similarity" in key sequences for peptide binding. If the authors are using NetMHCpan4.0, it seems that the idea of HLA allele divergence might be somewhat already accounted for. For example, for HLA alleles with low divergence, it is likely that prediction algorithms will also predict them to be of similar binding to the same group of

peptides and thus if you were to deduplicate and remove peptides in such cases, the resulting TNB is naturally lower. Can authors demonstrate the added value of using HAPS over using a properly defined tumor neoantigen burden?

Previously, Chan et al have demonstrated a similar concept regarding HLA allele divergence and ICI treatment response in their 2019 manuscript (PMID: 31700181). Here, authors stratify patients groups based on high and low HAPS throughout the manuscript to demonstrate differences in survival outcomes. I think it's important that authors more explicitly spell out what is novel in this study, particularly the added value of using HAPS versus solely using a tumor neoantigen burden and/or HLA allele divergence score, which I believe is a key finding of the paper that would demonstrate the novelty of this approach. Supplemental figure S6 seems to show this to some degree but a quantitative summarization of this across different cohorts would be more convincing. Additionally the authors should specify how the cutoffs for high vs low TNB and HLA alleles divergence are determined and what the cutoffs are exactly.

Building on top of the previous comment, the current definition of TNB seems to be solely based on binding affinity predictions. Could the authors add more complexity to the definition of TNB, such as additional filtering using RNA expression data (gene or transcript)? I understand this might be for a limited number of cohorts but would be nice to see if it improves survival stratification.

Since HAPS puts an emphasis on allele-specific differences, I believe it would be more appropriate to either use percentile predictions (e.g. 2% cutoff) and or elution rankings (e.g. common threshold rank of ~ 1.1 in NetMHCpan 4.0) when determining TNB instead of IC50 binding affinity or at least use allele-specific binding affinity cutoffs where available. Please see: <https://help.iedb.org/hc/en-us/articles/114094151811-Selecting-thresholds-cut-offs-for-MHC-class-I-and-II-binding-predictions>. I noticed that supplementary figure S7 shows a comparison between using IC50 vs percentile rankings, but lacks description of percentile cutoff that was used to determine TNB.

It would be a great resource to the community if authors could make their HAPS calculation pipeline available for others to use to calculate scores between any HLA allele of their choice or provide a supplementary table with HAPS scores between common HLA alleles of each class for others to look up.

Minor comments:

The authors use "neoantigen", "neoepitope" and "neopeptide" throughout the manuscript without explaining a clear difference between the three terms.

Figure captions should be more detailed and figure title/labels need additional attention. Some examples would be Figure S2 subplot titles and an explanation of what authors mean by "Ratio of HLA divergence", "Ratio of $\log_{10}(\text{TNB}+1)$ ", "Ratio of HAPS" in Figure 1A. Also please define all acronyms when they first occur in the manuscript, including those in figure legends.

For violin/box plots such as Fig S1, could the authors provide details on the sample sizes for homozygous and heterozygous cases?

The authors could probably improve on their alignment and somatic variant detection pipeline by using a more updated human reference and more than just Mutect for somatic variant detection. Also the precise details for how somatic variants are called were lacking in the methods.

It appears that methods sections are missing for the in vitro T-cell stimulation performed and neoantigen quality calculations. Specifically for neoantigen quality calculations, are the scores averaged across the number of neoantigens per patient?

Enclosed herewith is our revised manuscript (NCOMMS-23-11649A) entitled "Assessment of human leukocyte antigen-based neoantigen presentation to determine pan-cancer response to immunotherapy"

We truly appreciate the reviewers' constructive comments and suggestions. We have carefully revised our manuscript based on the reviewers' comments. Please kindly find the enclosed point-by-point response letter and the revised manuscript.

We hope that you will find that the reviewers' concerns have been adequately addressed, and this updated manuscript is now acceptable for publication in *Nature Communications*.

The point-by point response to the reviewers is as follows:

Reviewer #1 (Remarks to the Author): expertise in TCR-seq and WES bioinformatics

Han et al. present "Assessment of human leukocyte antigen-based neoantigen presentation to determine pan-cancer response to immunotherapy," their manuscript describing a score (HLA tumor-Antigen Presentation Score [HAPS]) used to stratify patients more likely to benefit from ICI therapy. The manuscript presents extensive correlative data. Results are presented that show HAPS may have some advantages over established markers (HLA-LOH, TMB, PD-L1, etc..) but it is not clear that HAPS is robust enough to inform clinical decision making, which is a central claim. Many methodological details and justifications are missing. Providing clarification on the points below would strengthen the paper.
Response: We appreciate the reviewer's recognition and important suggestions. We have carefully supplemented the methodological details and justifications, as well as the comments below point by point.

- Overall, figures and their text are undersized. Even viewed on the computer at 200% zoom, they were hard to read. Many figures lacked sufficient explanation in the figure legend or text to be easily interpretable.

Response: We apologize for the poor image presentation. We have revised and embellished all the figures and rewrote the figure legends to be easily interpretable.

- The GitHub code repository (<https://github.com/zxl2014swjx/HAPS>) appears to only contain code to generate figures, and not the actual analysis (calculation of HAPS score, TCR annotation, neoantigen predictions, differential gene expression analysis, gene set enrichment analysis, CIBERSORT, etc). The

entirety of the README is: “The codes and files are for the study of HAPS (Human Antigen Presentation Score).” This is insufficiently descriptive.

Response: Thanks a lot for your kindly reminding. We have supplemented the GitHub code repository so as to contain code for the present actual analyses. We also revised the descriptive of README accordingly.

- Line 104: Is the entire HLA sequence being compared? Just the binding groove? What is the justification for the choice?

Response: We thank the reviewer for the comments. In our investigation, we probed the evolutionary divergency between HLA-I alleles using a peptide sequence of 181 amino acids in length. To establish a foundation for this, the preliminary analysis by using the IMGT database—which encompasses the sequences of HLA-I A, B, C genotypes—revealed that the HLA-I sequence of 181 amino acids represent the predominant ones(1). Subsequently, we conducted a sequence alignment of HLA-I by utilizing the widely acclaimed MAFFT software (<http://mafft.cbrc.jp/alignment/software/>). Intriguingly, we discerned that the peptide segment of 181 amino acids, localized within the chr6—exon2,3 peptide-binding domain, encapsulated the salient genetic variations of HLA-I sequences. The results obtained through this computational method were found to be in accordance with previous study(2) (see the following Figure). As a result, we adopted the peptide sequence consisting of the aforementioned 181 amino acids to facilitate the assessment of HAPS calculations. We have updated “Computational identification of HLA-I evolutionary divergence” in the Method section, please kindly refer to line 632-640.

Reviewer Fig. RF1. Correlation analysis of HLA evolutionary divergence between precious study and 181-Amino Acid based calculation.

- Line 104: Define what a predicted neoantigen is (mutation that can yield a peptide capable of binding an HLA? A peptide-MHC pair?) How is this counted

– one per mutation, one per peptide, one per peptide-HLA pair? How are the predictions done?

Response: Thank the reviewer for this comment. The neoantigens predicted in our research were based on NetMHCpan4.0, a method trained on binding affinity and eluted ligand data leveraging the information from the both data types (3), which is commonly used to predict neoantigens(4, 5). The predicted IC50 binding affinities and rank percentage scores, representing the rank of the predicted affinity compared to a set of 400,000 random natural peptides, were calculated for all peptides binding to each of the patient’s HLA alleles using netMHCpan-4.0. The predicted putative neoantigens refer to those peptides with a predicted binding affinity < 500nM based on NetMHCpan4.0.

- Line 106-107: Why is the log10 of number of neoantigens used? Is there biological relevance for the neoantigen burden on a log scale compared to a linear scale? What statistical test was done? The distributions appear extremely overlapping – is the statistical difference biologically relevant?

Response: Actually, the neoantigen numbers followed an exponential distribution; the median, mean, 25-percentile and 75-percentile were 278.0, 528.1, 114.0 and 544, respectively. By taking the log transformation, a more normal-like distribution was obtained. That is the reason why we used log TNB (tumor neoantigen burden) for HAPS calculation. After carefully comparing the distribution and predictive value comprehensively, we consider log more suitable for HAPS, just like in the previous study (2). Additionally, the choice to employ a logarithmic transformation on the TNB of patients does not affect its biological interpretation. In Fig S1, as cited in manuscript line 106-107, the data follows a normal distribution, and we accordingly employed the T-test as our chosen statistical approach. The distribution of TNB reveals statistically significant variations among HLA-I A, B, and C alleles, as well as between homozygous and heterozygous HLA-I genotypes. The biological significance of these findings is as follows: Firstly, different HLA-I alleles display distinct capacities for presenting tumor antigens. Additionally, heterozygous HLA-I genotypes generally demonstrate a greater ability to present a larger repertoire of tumor antigens compared to homozygous HLA-I genotypes (6).

Reviewer Fig. RF2. Distribution of TNB (left) and $\log_{10}(\text{TNB}+1)$ (right) for in all enrolled patients with whole-exome sequencing (WES) data.

- Line 107-108: Was divergence calculated for all possible HLA pairs, or naturally occurring within patients (in your cohorts)?

Response: In our study, the divergences were calculated for all possible pairs of HLA-A, B, C genotypes. We developed and validated the HAPS index based on both our internal and public external cohorts, which means HAPS is widely applicable. We have uploaded the pipeline used for HAPS calculation and HLA-I allele divergence data to github (<https://github.com/zx12014swjx/HAPS/tree/main/hed/db/>). Other researchers can obtain the patient's HAPS score using the data and methods in this link, by providing the patient's TNB and HLA genotyping.

- Line 110: What binding affinity threshold?

Response: Thanks for the reviewer's question. As described in line 646 of the manuscript, MHC Class I/peptide pairs exhibiting binding affinities stronger than moderate ($\text{IC}_{50} < 500 \text{ nM}$) will be classified as potentially antigenic.

- Line 116-117: Given $\text{HAPS} = \text{HLA divergence} \times \log_{10}(\text{TNB})$, and divergence is a linear scale while TNB is used on a log scale, divergence will have a stronger impact on HAPS. Does divergence alone predict ICI response? Figure S6 seems to suggest so, where the survival curves for HLA diversity are very similar to survival curves for HAPS in almost all cohorts. Also, wouldn't TNB be correlated to divergence, since heterozygous HLA loci would have more possible neoantigens and higher divergence? Does the HAPS metric for patients that are homozygous at all loci make biological sense (where the score would be 0, but it is very possible they have bonafide neoantigens)?

Response: We truly appreciate the reviewer's insightful comments. The reason for using log TNB was that the distribution of neoantigen numbers was observed following an exponential distribution, the median, mean, 25-percentile and 75-

percentile were 278.0, 528.1, 114.0 and 544, respectively. By taking the log transformation, a normal distribution was obtained (Reviewer Fig. RF2). To clarify the impact of log transformation on TNB for HAPS, we first compared the immunotherapy predictive effects of HAPS with and without log transformation. The survival curves demonstrated very similarly as well as the hazard ratios (Reviewer Fig. RF3). We further refined our selection to include patients who experienced a change in grouping post-logarithmic transformation. We then juxtaposed the survival analysis of HAPS pre- and post-logarithmic transformation (Reviewer Fig. RF4). By contrasting the survival curves of these two subsets of patients (those who experienced a change post-log transformation versus those who did not), our findings indicate that the "high to low" group (originally low HAPS pre-log and high HAPS post-log) exhibited survival patterns more congruent with the High HAPS group. In contrast, the "low to high" group mirrored the survival patterns of the low HAPS cohort (Reviewer Fig. RF5). Consequently, patients who register as high HAPS following the logarithmic transformation of TNB appear to derive greater benefit from immunotherapy.

Pursuant to the reviewer's suggestion, we performed survival analyses for HAPS, HLA divergence, and TNB. The survival curves illustrated that all these indicators can distinguish patients benefiting from immunotherapy, however HAPS displaying the lowest P value and HR value (Reviewer Fig. RF6). For a definitive comparison of the independent predictive values of these factors, we incorporated TNB and HLA divergence into the multivariate regression, and pinpointed only HAPS as an independent determinant of OS (Reviewer Fig. RF7 and Fig. 2C). Additionally, we compared HAPS, TNB, and HLA divergence at the cohort level. Among ten ICI treatment cohorts, HAPS could distinguish the OS benefit population in six cohorts, whereas TNB and HLA divergence could only do so in one cohort, indicating that HAPS is superior to both TNB and HLA divergence (Reviewer Fig. RF8).

We further analyzed the correlation between TNB and HLA divergence. The results revealed a very weak correlation ($P=0.004$, $r=0.137$, Reviewer Fig. RF9), which is consistent with the previous study(2).

In this study, based on the calculation method for HLA allelic evolutionary divergence, patients who are homozygous have a HAPS score of 0. We definitely concur with the reviewer's perspective that homozygous HLA might still possess antigen presentation capabilities. In such, we made a comparative analysis of the number of neoantigens corresponding to homozygous HLA versus heterozygous HLA. The findings revealed that the neoantigen count for homozygous HLA is significantly lower than that for heterozygous HLA ($P<0.01$) (Reviewer Fig. RF10). Additionally, we evaluated the efficacy of immunotherapy in three distinct patient groups: those with homozygous HLA-I, those with high HAPS, and those with low HAPS. The results revealed that patients with homozygous HLA-I exhibited survival patterns that closely aligned with the low HAPS cohort in response to immunotherapy (Reviewer Fig. RF11). We have incorporated a discussion on this limitation in the revised manuscript, please refer to line 422-455.

Reviewer Fig. RF2. Distribution of TNB (left) and log10(TNB+1) (right) for in all enrolled patients with whole-exome sequencing (WES) data.

Reviewer Fig. RF3. Association of HAPS (left) and HAPS without log (right) with Overall survival in ICI treated patients.

Reviewer Fig. RF4. Association of HAPS without log (left) and HAPS log (right) with Overall survival in patients whose grouping changed after log transformation.

Reviewer Fig. RF5. Survival analysis of patients in different group.

Reviewer Fig. RF6. Summarized overall survival stratified by HLA divergence, TNB and HAPS.

Association of HLA divergence, TNB and HAPS with overall survival in ICI treated cohorts.

Reviewer Fig. RF7 (Fig. 2C). Multivariate Cox regression model assessing the correlation of clinical variables with OS.

Reviewer Fig. RF8. Comparison of predictive value of HLA-I Antigen Presentation Score (HAPS), TNB and HLA divergence. Overall survival of immune checkpoint inhibitor (ICI) stratified by HLA divergence (left), TNB (middle) and HAPS (right) in 6 cohorts.

Reviewer Fig. RF9. Correlation analyses of TNB with HLA divergence.

Reviewer Fig. RF10. Comparing the number of neoantigens corresponding to homozygous and heterozygous HLA in HLA-A, HLA-B, and HLA-C respectively.

Reviewer Fig. RF11. Overall survival stratified by HAPS and heterozygotes.

- Line 132: How was it “indicated” that 10 was the optimal cut-off? It has “almost” the lowest HR, so by what criterion was it selected?

Response: We thank the reviewer for the comments. For optimal cut-off determination, we selected the cut-off value that could best stratify patients with the lowest HR for OS in set Training 1 (the Wang-WES cohort, n=30). We continuously increased the HAPS cut-off to divide patients into two subgroups and calculated the corresponding HR. The accumulated HR curve illustrated an initial gradually dropping and then rising. The lowest HR (0.316, P=0.029; Fig. 1D) was obtained at HAPS=10, which was close to the median HAPS (10.86) in all enrolled ICI-treated patients (n=792; Supplementary Fig. S4). We therefore chose HAPS=10 as the cut-off for patient stratification. To further validate this cut-off value, we used the Rizvi 2015 cohort as an independent Caucasian training set (Training Set 2), obtaining a similar optimal cut-off point of 10 (with the lowest HR of 0.256, P=0.016; Fig. 1E). In such, we selected 10 as a universally applicable cut-off value to determine a high or low HAPS. Please kindly refer to line 139-144.

- Line 136: The HAPS threshold of 10 was selected using the Wang-WES data, and yielded a HR of 0.316, p=0.029 (line 132), however in line 136 it is reported that the same data and threshold had HR of 0.39 and p=0.05. Why do the HRs not match?

Response: We apologize for this mistake and thank the reviewer for the kind reminder. We have re-checked the data and the correct HR is 0.316 with a p-value of 0.029. We have made the corrections to Figure 1D and 1E in the revised manuscript.

Reviewer Fig. RF13 (Fig. 1D,E). Association of HAPS with Overall survival in Training 1 and 2 cohorts

- Line 139: The text states this obtained a “similar” optimal cut-off, but it appears to not be just similar, but actually the same (10).

Response: Thank the reviewer for pointing that out. We have now replaced 'similar' with 'same' in the revised manuscript.

- Line 147: As with neoantigen counts, what is the justification for using logs here for neoantigen quality?

Response: Thank you for the suggestion. We sincerely apologize for the oversight in our calculation of neoantigen quality. We have recalculated the neoantigen quality without using log transformations. Please refer to the revised Figure 1F.

Reviewer Fig. RF14. Distribution of neoantigen quality (left) and $\text{Log}_{10}(\text{quality}+1)$ (right) in different groups.

- Line 149: Were all peptides corresponding to predicted neoantigens synthesized? How were the 8 patients selected?

Response: Thanks a lot for the reviewer's comment. All peptides corresponding to predicted neoantigens were synthesized. The reason only 8 patients were included in the validation experiment is that only these 8 patients consented and provided peripheral blood specimens suitable for in vitro validation during the period of in vitro experiments from April 2019 to October 2019. Hopefully, enlarged case validation was performed.

- Figure S8: What do the size of the blue HR boxes mean? TNB is very large. How does TNB have a HR=1.00 (CI 1.00-1.00) but $p=0.00790$? This plot says neither TNB nor HLA divergence have any significant effect on OS alone (with tight HR CI around 1.0), but they do when combined into HAPS score. However, Figure S6 shows that HLA divergence and TNB may have an effect on OS, not reaching significance, but having wide HR CIs. These results do not seem consistent.

Response: We sincerely apologize for this oversight. The size of the blue HR boxes represents the magnitude of the HR value. In our study, we mistakenly conflated categorical variables with continuous ones. We have now re-conducted the univariate COX regression analysis, and the results indicated that HLA divergence, TNB, and HAPS all exhibit statistically significant differences (Reviewer Fig. RF15 and Fig. S8). Further multivariate regression analysis revealed that only HAPS possesses independent prognostic value (Reviewer Fig. RF7 and Fig. 2C).

Reviewer Fig. RF15 (Fig. S8). Univariate Cox regression analysis in ICI-treated patients

Reviewer Fig. RF7 (Fig. 2C). Multivariate Cox regression analysis in ICI-treated patients

- **Line 162-164: This sentence needs context for those readers not intimately familiar with pMHC binding predictions. Also, Figure S7 shows that the test is NetMHCpan4.1 vs NetMHCpan4.0, and doesn't say which generated %rank predictions and which generated IC50 predictions (both versions can produce both metrics).**

Response: Thank the reviewer for the constructive suggestions. We used IC50 in NetMHCpan 4.0, and 1% rank in NetMHCpan 4.1. We have added clarifications in the revised manuscript, please kindly refer to line 164 and figure legend for Figure S7.

- **Line 176. It is problematic that HAPS did not predict survival in the TCGA cohort. Support for HAPS is built on predicting OS (Figs 1, 2, 3). It is difficult to reconcile a claim that "HAPS may predict ICI response but not general prognosis" with the rest of the study. If ICI response is of interest, then this should be the main metric assessed, not OS. More validation cohorts should be assessed.**

Response: Thanks a lot for this comment. According to our analysis, the HAPS did not robustly predict overall survival in the TCGA cohort. Actually, almost all patients included in TCGA database did not receive subsequent immunotherapy. This result meant that HAPS is not a prognostic biomarker correlated with overall survival. The purpose of developing HAPS was to predict the benefit from immunotherapy, and therefore constructed based on ICI-treated patient cohorts. As a result, patients with high HAPS exhibited a longer PFS as well as OS than those with low HAPS in several ICI-treated cohorts, implying that survival benefit based on HAPS was due to immunotherapy. These results further indicated that HAPS is a predictor for response to immunotherapy rather than a prognostic biomarker for OS. Furthermore, according to the reviewer's suggestion, we identified an additional public dataset available for analysis, which contains information on 309 NSCLC patients who underwent immunotherapy(7). Within this cohort, HAPS effectively differentiated survival outcomes (whatever PFS or OS, Fig. RF16).

Reviewer Fig. RF16. OS (left) and PFS (right) stratification by HAPS in the newly added NSCLC cohort.

- Line 179-180: “...HAPS...superior to TNB and HLA divergence alone” – Fig S10 shows that in NSCLC, TNB is better than HAPS.

Response: Thank you for the comment. To compare the predictive significance of TNB and HAPS for NSCLC immunotherapy, we initially analyzed the cohorts included in Figure S10 separately. The results indicated that HAPS could distinguish survival benefits in 2 out of 4 cohorts, while TNB does so in only one cohort (Reviewer Fig. RF17). Furthermore, we identified an additional public dataset available for analysis, which contains information on 309 NSCLC patients who underwent immunotherapy(7). Upon calculating HAPS and TNB in this cohort, our findings illustrated that HAPS offered a superior distinction in survival benefit compared to TNB, with a lower HR and P value (Reviewer Fig. RF18), though both HAPS and TNB presented significant P values.

Reviewer Fig. RF17. OS stratification by HLA divergence, TNB and HAPS in cohorts with patients.

Reviewer Fig. RF18. OS stratification by HAPS and TNB in newly added NSCLC patients treated with ICI.

- Figure S12: What are the units for PD_L1, why does it go from 0 to 1?

Response: We apologize for the labeling error. The unit for PD-L1 expression level is percentage. We have corrected this in Fig S12.

- Figure S13: Which cohort was used for this, Wang-Panel-T or Wang-Panel-B?

Response: We are sorry for the unclear expression. The patients mentioned in Fig. S13 refer to those included in this study with WES data. From the WES data, we calculated TMB and TNB based on the genes corresponding to the 1021 panel and then conducted a correlation analysis with the WES data. The results indicated that both TMB and TNB showed good correlations of calculated by the 1021 panel and WES.

- Figure 2I, 2J, and 2K references are incorrect (Line 203-208).

Response: Thank you for pointing out the error. We have made the correction in the revised manuscript.

- Figure 3B: The third column/panel is just the previous two combined/overlaid. Are the previous 2 really needed? It makes it hard to interpret what is different/the same between them.

Response: Thanks a lot for this comment. The first two columns categorize patients into the HLA-intact group and the HLA-LOH group, respectively. The results indicate that HAPS can better differentiate immunotherapy efficacy when HLA-LOH is not present. The third column summarizes the previous two columns. Based on the reviewers' request, we have removed the third column from Figure 3.

- Line189. Why does HAPS correlate with TMB, but TMB is not predictive in the multivariate Cox analysis (Fig 2c)

Response: We thank the reviewer for this comment. Since TMB based TNB is one of the factors used for HAPS calculation, it is reasonable that HAPS is correlated with

TMB. Actually, in our study, there is a moderate correlation ($r=0.454$, less than 0.5) between HAPS and TMB. The purpose of the multivariate Cox regression is to adjust for the effects of confounding factors. HAPS has a stronger prognostic significance than other factors, such as TMB. After adjustment through multivariate Cox regression, only HAPS retained its independent prognostic value.

- Line 198-201: Does the panel-based HAPS threshold of 1.31 make sense compared to the genome-wide based HAPS threshold of 10 when considering the size difference of the 1021 gene panel vs the genome? Is it justified to select the threshold using the Wang-Panel-T cohort and then predict ICI benefit in the same cohort?

Response: We thank the reviewer for the kindly reminding. In our study, the HAPS threshold of 10 based on WES has been validated across multiple independent cohorts. However, limited by the currently available data, the used threshold of 1.31 originating from 1021-gene panel was just settled down based on one single-center cohort, which might bring more bias compared with WES based HAPS threshold. We also agree with the reviewer that the used gene panel size might impact the HAPS threshold. However, regardless of WES or gene panel was utilized for HAPS calculation, the method for determining the threshold remained consistent. Although the HAPS threshold is different, the positive correlation between HAPS and efficacy of immunotherapy indicates the rationale of the method for determining the threshold. To further verify the rationale of threshold in our panel cohort, we employed a 10-fold cross-validation approach. The results showed that in 8 out of the 10 models, the optimal threshold was 1.31 consistently, emphasizing the appropriateness of the current cutoff (Reviewer Fig. RF18).

Reviewer Fig. RF19. Determination of the optimal cutoff value in Wang-panel-T cohort by 10-fold Cross-Validation.

- Line 216-217: Could you test whether HAPS and HLA LOH are correlated or not? Likewise, does HLA LOH correlate with HLA diversity? Should/could HLA LOH be taken into account when calculating HLA diversity (if there is HLA LOH, the effective HLA diversity should be 0)?

Response: We appreciate the reviewer's constructive comments. Following the reviewer's recommendation, we investigated the relationships among HAPS, HLA divergence, and HLA LOH. The results indicated that in the HAPS grouping, the distribution of HLA-LOH shows no difference. However, by HLA divergence grouping, patients with higher HLA divergence have a higher proportion of carrying HLA-LOH (Reviewer Fig. RF20). Theoretically, higher HLA divergence meant that more diverse neoantigens are presented and therefore face higher likelihood of immune escape due to secondary HLA-LOH. With the occurrence of HLA-LOH, the ability of neoantigen presentation will significantly decrease. These analyses suggested that both HLA divergence and HLA-LOH reflect the ability of neoantigen presentation and influence each other. In addition, HLA divergence and HLA-LOH belong to continuous and binary variables respectively, incorporating those in one model always difficult. Actually, we have tried to establish a model including both HLA divergence and HLA-LOH, however failed. Also generally, a model including two correlated parameter depending on each other might influence the feasibility for model construction. By using the established HAPS formula, if there is HLA LOH, both HAPS and HLA divergence are "0", suggesting a decreased ability of neoantigen presentation and inferior response to immunotherapy, which is consistent with biological relevance and current understanding.

Besides of these considerations, there are several other explanations why HLA LOH was excluded from HAPS model. First, the parameters used for HAPS construction included HLA-I evolutionary divergence and predicted neoantigens, both of which are based on germline HLA sequencing, different from HLA-LOH (somatic mutations). Second, to our knowledge, there is no gold standard method for HLA-LOH assessment, which will lead to more extent of bias and uncertainty of the HAPS model, if HLA LOH is used. Finally, one goal of this study was to apply HAPS in peripheral blood as a noninvasive predictor of response to immunotherapy. However, up to date, peripheral blood based HLA-LOH has not been established due to HLA-LOH only occurring in tumor cells. To better determine the impact of HLA-LOH on the predictive value of HAPS, we conducted a subgroup analysis, and the results showed a better performance of HAPS with intact HLA (Fig 3B). A significant association was found between high HAPS and overall survival (OS) in patients with intact HLA, with no association between high-HAPS and OS in patients with HLA-LOH (Reviewer Fig. RF21). These results demonstrate that HLA-LOH is not associated with high or low HAPS, and combined use of HAPS with HLA-LOH may better predict response to immunotherapy.

Reviewer Fig. RF20. HLA-LOH prevalence between high/low HAPS (left) and high/low HLA divergence (right) in ICI treated cohorts.

Reviewer Fig. RF21. OS stratification by loss of heterozygosity (LOH)

- Line 246-247: Is there any way to quantify if there is greater intercellular interactions in patients with high HAPS?

Response: We are grateful for the reviewer's suggestion. We initially analyzed the number and intensity of cell types that exhibited both positive and negative correlations (quantified as $r \times -\log_{10}(pvalue)$). Our findings revealed that, within the High HAPS group, correlations were identified between 232 cell pairs: 94 of these were positively correlated with a median value of (median $r \times -\log_{10}(pvalue) = 0.623$), while 138 pairs exhibited negative correlations (median $r \times -\log_{10}(pvalue) = -0.789$). In contrast, the Low HAPS group demonstrated correlations for 168 cell pairs, with 62 pairs showing positive correlations (median $r \times -\log_{10}(pvalue) = 0.423$) and 106 pairs indicating negative correlations (median $r \times -\log_{10}(pvalue) = -0.519$). These observations suggested that the High HAPS group consistently exhibits higher values than the Low HAPS group in terms of both the number of correlated cell types and the intensity of these correlations. This underscores the possibility of stronger

intercellular interactions within the tumor microenvironment in the High HAPS group.

- Line 237: Fig 3D ref is wrong.

Response: Thank you for pointing out the error. We have removed “Figure 3D” in the revised manuscript.

- Line 253-254, Fig 4B: Distributions are so overlapping, is there any biological relevance to this small statistical difference?

Response: We thank the reviewer for this challenging question. After calculating the GEP and CYT scores in 455 patients, the results demonstrated that both scores are higher in the high HAPS group (CYT: median 134.17 vs. 86.71, GEP: median 3.09 vs. 2.89). Due to the large data dispersion, the difference doesn't appear pronounced in the figure. Both scores epitomize the body's immune cytotoxic potential. Enhanced antigen-presentation capability, as indicated by high HAPS, tends to bolster the host immune response. Immune checkpoints, in this context, are pivotal for facilitating immune escape. Consequently, high HAPS is associated with augmented immune cytotoxic scores, suggesting potential benefits from immunotherapy. These consistent analyses indicated the positively biological relevance, though overlapping distributions indeed exist.

- Line 259: how are the counts defined?

Response: Thank you for pointing out the error. The infiltration level of pDCs in the immune microenvironment was assessed using ssGSEA. This method cannot quantify the count of immune cells. Actually, it works by analyzing the known immune-related gene sets, which enables the determination of the relative abundance of immune cell subtypes in each sample (8, 9). We have corrected this error in the revised manuscript. Please refer to line 259.

- Line 261-263: is there any biological relevance to the small difference? Can you really conclude that HAPS may reflect antigen presentation capacity from this?

Response: We thank the reviewer for the comments. We analyzed immune infiltration in 455 patients and revealed elevated levels of both MHC-I and MHC-II in the high HAPS group compared to their counterparts in the low HAPS group, albeit the magnitude of this increase was modest. Theoretically, the HAPS score introduced in this study reflects a patient's tumor antigen presentation capability. With higher levels of neoantigens and a more diverse set of HLA-I alleles, there is also an expected increase in the expression of antigen-presentation molecules MHC.

- Line 266. If checkpoint genes are expressed more highly in the HAPS group, could this alone explain and association between HAPS and ICI response?

Response: Thanks a lot for the reviewer's suggestion. Initially, we performed survival analyses for the checkpoint genes that were overexpressed in the high HAPS group. The results revealed that CD279, CTLA4, and LAG3 can differentiate populations

that benefit from immunotherapy (Reviewer Fig. RF22). Further, we incorporated these three factors along with HAPS into a multivariate Cox regression analysis, which indicated that HAPS is an independent prognostic factor for immunotherapy outcomes (Reviewer Fig. RF23). In addition, considering the significantly positive correlation between HAPS and checkpoint gene mRNA level (Fig. 4E), the positive effect of checkpoint gene mRNA level on response to immunotherapy is mainly due to HAPS. Therefore, according to our data, checkpoint genes could not alone explain the association between HAPS and ICI response.

Reviewer Fig. RF22. Survival analysis of checkpoint genes.

Overall survival of immune checkpoint inhibitor (ICI) stratified by checkpoint genes.

Reviewer Fig. RF23. Multivariate Cox regression model assessing the correlation of checkpoint genes and HAPS with OS.

- Line 271-272: Is it realistic to expect that there are ~16,000 genes that are differentially expressed between HAPS high and low groups? That is the

majority of genes. If this is a true result, why are there three times as many downregulated as upregulated?

Response: We are truly sorry for the mistake. We conducted a revised differential gene expression analysis. Initially, genes localizing in non-coding regions were excluded. We then standardized the gene expression levels. Using the limma package, we performed the differential gene expression assessment. Genes with an absolute fold change exceeding 1 and an FDR below 0.05 were identified as differentially expressed.

As a result, 13 genes were found to be upregulated in the high HAPS group, whereas 12 were downregulated. These changes have been made in the revised manuscript and Figure 4F; please kindly see lines 273 to 274.

Reviewer Fig. RF24 (Figure 4F). Differential gene expression analysis between high HAPS and low HAPS.

- Line 283-284: which “complementarity-determining regions” were used?

Response: The “complementarity-determining regions” here refers to TCR β chain CDR3s, We have made the necessary revisions to the revised manuscript, please kindly refer to line 283.

- Line 296, Line 308. If a finding is not statistically significant it is not a finding.

Response: We are grateful for the reviewer's suggestion. As depicted in line 296 and Figure 5C, we performed the survival analysis based on HAPS and TCR diversity within the Wang-Panel-B cohort. Although limited by the number of patients included in each subgroup, the analysis did yield statistically significant differences ($P=0.032$). From the illustration, it is discernible that patients with high HAPS and elevated TCR diversity have notably superior prognosis than the other three groups. Furthermore, due to other three groups largely overlapping, we incorporated the three groups and compared to group with high HAPS and elevated TCR diversity, the difference presented more apparently as well as significant p value (Reviewer Fig. RF25). there's minimal overlap with the other groups, hinting at the likelihood of these patients

benefiting from immunotherapy. Following the reviewer's suggestion, we've revised the state that “their combined assessment could potentially serve for non-invasive patient stratification for ICI delivery” Please refer to line 297-298.

Line 309 and sFig 17C depicted the change in TCR clone numbers before and after treatment across different HAPS groups. Similarly, due to the limited number of patients included, the results suggest a trend where the high HAPS group displays an increased number of TCR clones compared to the low group.

Reviewer Fig. RF25. Survival analysis of patients with high HAPS and TCR diversity versus others.

- **Line 301:** The authors state that TCR clonality is increased (by 0.08) over time in half the patients with high HAPS/high diversity, and that it decreases for low HAPS and low diversity. However, the other half of patients in the high/high group showed decreased clonality, apparently with greater magnitude. This is a questionable description of the data – the high group likely has no overall change in clonality if a statistical test was done.

Response: We appreciate the reviewer’s suggestion. In the high HAPS/high diversity group, 7 patients showed a decline with a mean decrease of 0.17. Meanwhile, in the low HAPS & low diversity group, the average decline was 0.22. Although there was no statistical difference between them (P=0.63), the magnitude of the decline in the low HAPS & low diversity group appeared greater than that in the high HAPS/high diversity group. Overall, patients with low HAPS/low diversity presented more degree of decrement of TCR clonality compared with those with high HAPS/high diversity (p=0.041, Fig. 5D).

- **Line 313:** what is meant by “epitope-specific amino acid residues”? Are these just the amino acids comprising the CDR3, or something else?

Response: We apologize to the reviewer for this issue. The edit distance was calculated based on CDR3 amino acid sequences, we have made the correction in the revised manuscript, please kindly refer to line 313.

- **Line 316:** There is a small statistical difference here, but the medians are the same. What is the biological relevance of such a small change?

Response: We thank the reviewer for this challenging comment.

TCR edit distance refers to the similarities between TCR sequences. This distance serves as an indication of the immune system's capacity to discern a broad spectrum of antigens(10). A more pronounced edit distance may suggest a richer diversity within the TCR repertoire. Given the vast array of TCRs each patient carries, determining an individual's edit distance necessitates comprehensive comparisons across all TCRs. We indeed noticed that, while there might be statistical differences, the absolute value of the edit distance among distinct groups was slightly different, which might be due to the limited sample size. However, some typical cases were observed. In addition, as a supportive data for HAPS related characteristic, this can still provide insights into the extent of TCR sequence variations between groups.

- Figure 5F: This figure needs labels to help guide interpretation.

Response: We thank the reviewer for the kind reminder. The x and y axes of Figure 5F both represent TCR sequences. We have added this to the figure.

- Figure 5G, H: As above, can you justify that there is biological relevance to these small differences between completely overlapping distributions?

Response: We thank the reviewer for the comments. Figure 5G and 5H illustrated the variations or disparities in the TCR edit distance derived from comparisons among different patient groups. As mentioned in the response above, this involves a comparison of edit distances across all TCRs. Consequently, while there might be statistically significant differences, the absolute values of the edit distances among different groups could be subtly varied. Nevertheless, these distinctions still offer insights into the extent of TCR sequence variations between the groups. In addition, all the consistent trends with biological rationality including edit distance and above results mentioned in Fig. 5 enhanced the reliability of the analyses and interpretation.

- Figure 6A: I do not understand what this plot is showing.

Response: Thanks a lot for the reviewer's kindly reminding. Briefly, to devise a predictive model for immunotherapy efficacy, we initially undertook a multiple linear regression analysis spanning nine determinants: HAPS, TMB, TNB, TCR diversity, TCR clonality, HLA heterozygosity, HLA-LOH, smoking status, and PD-L1 expression. Notably, we observed significant inter-individual variations in the predictive value of these factors. As illustrated in Fig. 6A, each colored trajectory corresponds to a distinct patient, and the dots along this trajectory denote the relative influence of each factor on the therapeutic response of that particular patient. For a more comprehensive explanation, we have expanded the figure legend and relocated figure 6A to the supplementary materials. Please kindly refer to the Fig. S17 in the revised supplementary materials.

- Line 341-342: Were only these top three used for model building? If not, why call them out?

Response: In our study, during the process of model selection, we leveraged the "caret" package to accommodate the AUC value fitting for diverse models and varying numbers of included parameters. Our results identified the optimal performance of the neural network model that is based on three parameters (HAPS, TCR diversity, and TMB), achieving the optimal AUC value (AUC=0.78), as delineated in the newly appended Figure 6B. Subsequently, we ranked the nine candidate factors based on their importance and selected the top three - HAPS, TCR diversity, and TMB - to establish our model (Figure 6C).

Reviewer Fig. RF26 (newly added Figure 6B). Comparison of AUCs for various combinations of models and parameter counts using Caret.

- Line 346-347: This sounds like you applied the NN model to the same patients that were used to train it. Presumably this is not the case.

Response: We appreciate the feedback from the reviewer. Due to the limited number of immunotherapy patients in our study with both TCR and genomic sequencing data, we divided the samples into tissue cohort and blood cohort based on the sample type. The model was trained in the blood cohort, which had a larger sample size, and then validated in the tissue cohort. Importantly, while the model's training was based on the therapeutic response of immunotherapy, the finalized model effectively differentiated the survival benefits of immunotherapy across both cohorts.

- Figure 6H: This figure needs more description to be interpretable.

Response: Thanks a lot for the suggestion. The correlation between the NN (neural network) scores and the response to ICI treatment. The x-axis represents the NN scores, and the y-axis shows different patients and their response to ICI. The left side is for the Wang-Panel B cohort, and the right side is for the Wang-Panel T cohort. We have added the description to the revised legend of figure 6H.

- Line 595: “encoding the neoantigen” I think you mean “MHC”.

Response: We appreciate the reviewer pointing out the error. We have rewritten the "Computational identification of HLA-I evolutionary divergence" in the Method section. Please kindly refer to line 632-640.

- Line 597: “amino acid length equal to 181 nucleotides” – do you mean 181 amino acids?

Response: We are truly sorry for the oversight. The length included in the analysis was 181 amino acids. We have made the correction in the revised manuscript.

- Line 599: “protein sequences of exons 2 and 3” – is it these exons, or the 181nt/aa section?

Response: We apologize to the reviewer for this issue and have updated “Computational identification of HLA-I evolutionary divergence” in the Method section, please kindly refer to line 632-640. The length included in the analysis was 181 amino acids.

- Line 611: How were HLA genotypes determined? Existing data, or analyzed by the authors? What tools were used (by authors or existing data)? If it was NGS-based, what read length(s) were used?

Response: We thank the reviewer for the comments. In this study, our cohort includes both public cohorts and internal datasets from our center. For the public datasets, we utilized existing data with tools such as Polysolver(11, 12), OptiType(13, 14), ATHLATES(15-18), and SOAP-HLA(19). In the internal Wang cohort, the tool used was Polysolver. For the WES sequencing, the read length employed was 100 bp, and for the panel sequencing, it was 151 bp.

- Line 612: “of the top recurrent somatic mutations” – Does this mean that neoantigens were not predicted for all mutations in a patient?

Response: We sincerely apologize for the oversight. In our research, neoantigens were predicted for every mutation in one patient. We have removed “top recurrent” from the revised manuscript.

- Line 615: Which IEDB algorithms were used?

Response: We employed NetMHCpan 4.0 in our study, which was recommended by IEDB. (<http://tools.iedb.org/mhci/help/#Method>).

- Line 617: “when neoepitopes are available” – what does this mean? In general, for neoantigen predictions, were all overlapping peptides tested (ie. All possible 8-11mer peptides with the mutation in every possible position)?

Response: We thank the reviewer for the comments. For neoantigen predictions, we tested all overlapping peptides. We have removed the phrase "when neoepitopes are available" from the revised manuscript.

- Line 621-622: This is possibly confusing, as the HAPS threshold for the panel was a different value. Also, what is meant by a “median score” of 10?

Response: We apologize for this mistake. The HAPS threshold was determined based on the lowest HR for OS, which is 10 in WES cohorts. We have made the revision in the manuscript, please kindly refer to line 657-659 in the Method section.

- Line 638-641: Unclear what is meant by this.

Response: We sincerely apologize for the oversight and have removed the ambiguous description.

- Line 655: “we devised” – since this metric was taken from an existing publication, it was already devised.

Response: We thank the reviewer for the kind reminder. We have modified "devised" to "utilized" in revised manuscript.

- Line 661: Significantly more detail is needed on how the TCR-b sequencing was done. How were the cells isolated? How were CDR3 regions selected? Were primers used? Was this from RNA? DNA? What sequence machine was used? What read length? What sequence depth was achieved? Provide a summary of the sequence data obtained. How was the resulting sequencing quality filtered and annotated with TCR features?

Response: We thank the reviewer for this feedback and suggestion. PBMCs were extracted from 10 mL of fresh peripheral blood with anticoagulant through density gradient centrifugation with Lymphoprep (Progen). Using FACS analysis (BD FACSAria™), PD-1+ CD8+ T cells were obtained. The specific antibodies used, sourced from eBioscience, included CD3-eFluor 450 (OKT3), CD8-APC (RPA-T8), and CD279 (PD-1)-PE (MIH4). DNA was isolated from the PD-1+ CD8+ T cells using the QIAamp DNA Mini Kit (QIAGEN, catalog: 51306). The DNA yield was ≥ 1 mg with UV absorption ratios at wavelengths 260/280 and 260/230 being ≥ 1.8 and 2, respectively. A multiplex PCR approach was applied to amplify the CDR3 in TCR β chain (TRB) both inclusively and semi-quantitatively. This involved a two-round PCR process, with primer sequences listed under Chinese patent CN105087789A (20).

During the first PCR round, 10 cycles were conducted to amplify CDR3 sequences using specific primers. The initial round involved a reaction system comprising of 600 ng of template DNA, QIAGEN Multiplex PCR Master Mix, Q solution, and primer

set pools using a Multiplex PCR Kit (QIAGEN). Post this, a purification step was conducted with magnetic beads (Agencourt no. A63882, Beckman). All products from this round were used as templates for the second amplification step, adding pooled primers and a Phusion High-Fidelity PCR Kit. This was followed by a cycle program: one minute at 98°C, 25 cycles at various temperatures, and a final extension. The amplified products were size-selected via agarose gel electrophoresis, targeting fragments between 200-350 bp, purified using the QIAquick Gel Purification Kit (QIAGEN). Finally, paired-end sequencing of these samples was executed using the Illumina HiSeq 3000 platform, achieving a read length of 151 bp. We have added this part to the revised manuscript. Please kindly refer to line 696-716 in the Method section.

- Line 672-674: In Figure 5F, the diagonal is not always the same TCR (and so may not have the same epitope specificity). This is not consistent with the text description at these lines.

Response: We apologize for this mistake and thank the reviewer for the kind reminder. The diagonal is the same TCR, we have modified Figure 5F in the revised manuscript.

- Line 679: I found this section unclear. How were features associated with outcomes? How is it known that human antigen presentation, TMB, and T lymphocyte distributions are independent? What is the architecture of the model?

Response: We thank the reviewer for the comments. In response to the reviewer's request, we performed survival analysis of HAPS, TMB and TCR diversity. The results suggested that none of these parameters could differentiate the survival outcomes of immunotherapy (Reviewer Fig. RF27). Furthermore, we conducted a correlation analysis on the three factors included in the neural network model. The results showed that HAPS is positively correlated with TMB, while HAPS and TCR diversity, as well as TMB and TCR diversity, are not correlated (Reviewer Fig. RF28). In our study, we employed the R package "neuralnet" to construct a neural network model. This model used HAPS, TMB, and Diversity as input layers, with Response designated as the output layer (Reviewer Fig. RF29 and newly added Fig. S18). The configuration was designed with a singular hidden layer and a non-linear output, leveraging the RPROP algorithm and the "sse" error function to assess error rates. Within this network, neurons were interlinked, and the strength of these connections was characterized by node link weights. The model training underwent 20 iterations and concluded when the absolute change in the error function fell below 0.01. Error assessments were based on the AIC criterion. The intercepts for the hidden neurons were found to be 1.9214, with weight predictions being 1.54006, 1.4383, and 2.46875. The output layer showcased intercepts of -0.42557 and -1.76532. We have added this part to the revised manuscript. Please kindly refer to line 756 to 765 in the Method section and Fig. S18.

Reviewer Fig. RF27. Overall survival of immune checkpoint inhibitor (ICI) stratified by HLA divergence (left), TCR diversity (middle) and TMB (right) in the Wang-Blood cohort.

Reviewer Fig. RF28. Correlation analyses of features included in the neural network model.

Reviewer Fig. RF29 (newly added Fig. S18). The architecture of the neural network model.

- Line 698: Is it appropriate to use a non-parametric paired test for all comparisons? Many distributions appear normally distributed, and a T test would perform well. There are also many comparisons that do not seem appropriate for a paired test (comparisons of high and low HAPS patients, for instance – what would the pairing be?).

Response: We apologize to the reviewer for this issue. We re-evaluated the data included in the study for normal distribution. For data that adhered to a normal distribution, we performed a t-test again and corrected the results in the revised manuscript. Additionally, we apologize for the inaccuracies in our methodology regarding the statistical methods used for comparisons. In fact, the comparisons of high and low HAPS patients utilized an unpaired test. We have corrected the statistical description in the methods section of the manuscript, please refer to line 772-773.

Reviewer #2 (Remarks to the Author): expertise in HLA sequencing and ICI response

The manuscript " Assessment of human leukocyte antigen-1 based neoantigen presentation to determine pan-cancer response to immunotherapy' by Han et al. describes the integration of multiple factors for better prediction of ICI treatment outcomes.

Overall, the study is interesting but it is quite complex and it was a little difficult to follow the rationale throughout. Several essential points need to be addressed for clarity before this manuscript should be considered for publication.

Response: We appreciate the positive feedback and constructive criticism from the reviewer.

1. Authors define a novel HLA presentation score (HAPS). The score is based on (i) HLA allele divergence, which has been previously shown to strongly predict ICI treatment outcomes (Ref14), and (ii) the number of predicted neoantigens, also known to predict treatment outcomes (Ref9 and others). HLA divergence should be introduced in more detail.

I did not understand the rationale for combining both factors for patient stratification. Does HLA divergence and number of presented neoantigens correlate overall (as expected and observed in Ref14)? Even if variability for different HLA alleles is observed, it is not clear to me why both factors should be combined for patient stratification. It is in my view therefore essential that the authors show how HAPS compares to the performance of HLA divergence and number of neoantigens separately (on patient cohort level).

Response: We are grateful to the reviewer for this insight. First, the divergence observed in HLA alleles plays a pivotal role in determining the sequence variations

within the peptide-binding domains of HLA-I alleles in patients. This divergence is quantified by taking the average Grantham distance among paired alleles within each HLA-I subtype. This metric offers insight into the spectrum of neoantigens that have the potential to bind to HLA-I molecules. On the other hand, the number of tumor neoantigens, predicted based on the affinity of individual HLA-I genotypes, is observed to be distributed among various HLA-I subtypes (21). The HAPS model we developed, grounded in bioinformatics, biologically reflects tumor immunogenicity (TNB) and neoantigen presentation (HLA allele divergence). We performed survival analyses for HAPS, HLA divergence, and TNB. The findings illustrate that all these indicators can distinguish patients reaping benefits from immunotherapy, with HAPS displaying the lowest P value and HR value (Reviewer Fig. RF1 and Fig. S8). For a definitive comparison of the independent predictive values of these factors, we incorporated TNB and HLA divergence into the multivariate regression, and pinpointed HAPS as an independent determinant of OS (Reviewer Fig. RF2 and Fig. 2C).

Following the reviewer's suggestion, we analyzed the correlation between TNB and HLA divergence. The results revealed a weak correlation ($P=0.004$, $r=0.137$, see the Reviewer Fig. RF3), which is consistent with the previous study(2). Additionally, we compared HAPS, TNB, and HLA divergence at the cohort level. Among ten ICI treatment cohorts, HAPS could distinguish the OS benefit population in six cohorts, whereas TNB and HLA divergence could only do so in one cohort, indicating that HAPS is superior to both (Reviewer Fig. RF4 and Fig. S6). In addition, as mentioned by the reviewer, we introduced the HLA allele divergence in more detail in line 69-74 in the Introduction section of the revised manuscript.

Reviewer Fig. RF1 (Fig. S6). Single Cox regression model assessing the correlation of clinical variables with OS.

Multi Cox Analysis	Pvalue	HR	Lower 95%CI	Upper 95%CI	
TMB >14 VS <=14	0.761051174	0.923089725	0.551108316	1.546147312	
PDL1 >=1% VS <1%	0.363414478	0.813921371	0.522103519	1.268844154	
Stage IV VS III	0.226330261	1.150054912	0.916970669	1.442386704	
Age >65 VS <=65	0.686855385	0.936203945	0.679479261	1.289925796	
Gender Male VS Female	0.915067153	1.009566869	0.847504253	1.202619644	
HLA Divergence	0.621666186	0.948741617	0.769817762	1.169251607	
TNB	0.207259412	0.877484948	0.716188061	1.075108447	
HAPS High VS Low	0.030829966	0.762994736	0.596878819	0.975341977	

Reviewer Fig. RF2 (Fig. 2C). Multivariate Cox regression model assessing the correlation of clinical variables with OS.

Reviewer Fig. RF3. Correlation analyses of TNB with HLA divergence.

Reviewer Fig. RF4. Comparison of predictive value of HLA-I Antigen Presentation Score (HAPS), TNB and HLA divergence. Overall survival of immune checkpoint inhibitor (ICI) stratified by HLA divergence (left), TNB (middle) and HAPS (right) in 6 cohorts.

2.A framework is then applied to estimate the immunogenicity of the predicted immunogenicity of neoantigens, in alignment to Ref19. Authors show that patients with high HAPS score present more immunogenic peptides than patients with low HAPS score. If I understand correctly, the HAPS score contains quantitative predicted neoantigen numbers in each patient, so is this creating a bias and suggest direct correlation between the number of neoantigens, and the number of immunogenic neoantigens?

Response: We completely agree with the reviewer's opinion that “predicted neoantigen” are not equal to “immunogenic peptides” which may be more likely to mediate robust immune responses and lead to clinical benefit. However, it is challenging to verify predicted neoantigens *in vitro* for each patient for clinical feasibility and model construction. Therefore, it is essential to develop a relatively convenient evaluation method that can be used in clinical practice, which was also one of the goals of our research. To quantify the immunogenic (quality) of predicted neoantigens, we referred to an established model based on the antigenic distance required for a neoantigen to differentially bind the HLA or activate a T cell compared with its wild-type peptide(22). The results showed that neoantigens in the high HAPS group were more immunogenic (Reviewer Fig. RF5 and Fig. 1F). To corroborate these findings, we designed and synthesized 44 neoantigen peptides and their corresponding wild-type (WT) counterparts in eight patients with available peripheral blood (4 with high HAPS and 4 with low HAPS) (Table S2) and stimulated T cells *in vitro*. The result showed higher upregulation of 4-1BB on CD8+ lymphocytes in the high HAPS group than in the low HAPS group. Moreover, the proportion of predicted neoantigens that led to a significantly up-regulated expression of 4-1BB on CD8+ T cells was higher in the high HAPS group (33.3% vs. 17.2%, $P=0.171$, Reviewer Fig. RF6 and Fig. S5). These results indicate neoantigens from patients with high HAPS are more likely to be presented and activate CD8+ T cells. Taken together, compared with lower HAPS, higher HAPS cannot pinpoint but is more likely to encompass immunogenic neoantigens.

Reviewer Fig. RF5 (Fig. 1F). Higher Neoantigen quality in the high HAPS group.

Log10(quality+1) of predicted neoantigens in different groups.

Table S2. Amino acid sequences of predicted neoantigens.

Amino acid sequences of predicted neoantigens in patients with high (n=4) and low (n=4) HAPS.							
Patients (group)	Gene	cHGVS	Mutant peptide	MHC class I restriction	Predicted mutant IC50 (50nM)	WT peptide	Predicted WT IC50 (50nM)
Patient No. 1 (high)	CDKN2A	c.221delA	APEAVTMPA	HLA-B07:02	16.5	AAREGFLDT	277.9
	TP53	c.731G>T	MVGMNRRPI	HLA-A30:01	14.1718	MGGMNRPI	2134.866
	DDR2	c.1867C>G	ARNDFVKEI	HLA-C06:02	20.5034	ARNDFLKEI	358.8707
Patient No. 2 (high)	KRAS	c.35G>T	VVGAVGVGK	HLA-A11:01	89.6342	VVGAGGVGK	7639.657
Patient No. 3 (high)	CDKN2A	c.278C>A	AAREGFLDK	HLA-A11:01	303.3602	AAREGFLDT	32862.93
	TBX3	c.209G>T	VAAETGIPF	HLA-C03:02	7.6854	GAAETGIPF	166.3957
	CDH23	c.6910C>T	FGITYYMEW	HLA-B58:01	26.5251	FGITYYMER	26074.11
	PRKDC	c.5605A>C	STFDTQITQK	HLA-A11:01	9.1593	STFDTQITKK	140.2333
	FMN2	c.2879C>G	AAIPPPPLR	HLA-A11:01	398.4271	AAIPPPPLP	33524.46
	MS4A1	c.304G>T	GSLLSATEK	HLA-A11:01	26.962	GSLLAATEK	922.4379
Patient No. 4 (high)	SERPIN3	c.955G>T	MTWSRGLVL	HLA-C07:02	374.6986	MTGSRGLVL	2409.3018
	IGF1R	c.3115G>A	KTVNEATSMR	HLA-A11:01	271.1686	KTVNEAASMR	439.3816
	TET2	c.4964C>T	YSPQSQMDLY	HLA-A01:01	71.4678	YSPQSQMDLY	134.362
	TP53	c.874A>G	KEGEPHHEL	HLA-B40:01	36.8813	KKGEPHHEL	13006.0527
	RET	c.2731G>A	RSQSRIPVKW	HLA-B57:01	15.3849	RSQGRIPVKW	19.1893
	ABCG2	c.404C>T	VVMGTLMVR	HLA-A11:01	48.7093	VVMGTLTVR	132.0888
	LRP1B	c.7531A>T	VTKNSSCYAY	HLA-A01:01	470.529	VTKNSSCNAY	822.0314
	LRP1B	c.1819G>T	AVYWIGNNLY	HLA-A11:01	141.2093	AVDWIGNNLY	904.4221
	ARID1B	c.4286G>A	GQYPYPYSK	HLA-A11:01	38.318	GQYPYPYSR	320.3552
	FAM46C	c.659A>C	MYGDFEAEF	HLA-C07:02	142.9464	MYGDFEAEF	402.1778
	EPHA4	c.490A>T	QVDIGDRILK	HLA-A11:01	124.934	QVDIGDRIMK	259.5792
	LRP1B	c.1132G>A	AALALNLVNK	HLA-A11:01	25.0089	AALALDLVNK	30.2811
	SLCO1B3	c.730A>T	LSSIRITPK	HLA-A11:01	41.8146	LSTIRITPK	75.3174
	FAT1	c.9803C>T	IEANAEITYL	HLA-B40:01	64.9053	IEANAEITYS	16948.0586
Patient NO.5 (low)	CSMD1	c.8601C>A	KAVLTGELF	HLA-B57:01	119.7433	NAVLTGELF	5581.089
	TSC2	c.2112G>T	KQESDWNVLK	HLA-A11:01	36.8813	KQESDWKVLK	1106.965
Patient No.6 (low)	ESR1	c.1754C>T	YYIMGEAEGF	HLA-A24:02	41.864	YYITGEAEGF	3206.409
Patient No.7 (low)	TP53	c.745A>G	MNRGPILTI	HLA-C12:03	166.7363	MNRRPILTI	418.306
	PPM1D	c.472G>T	AMWKKLSEWPK	HLA-A03:01	75.3468	AMWKKLAEWPK	105.432
	MLL3	c.2306C>T	SLFSSADISK	HLA-A03:01	36.4934	SSFSSADISK	183.0018
	IL7R	c.1130G>T	NVSAFDAPI	HLA-A02:06	266.431	NVSACDAPI	637.5781
	LRP1B	c.4666T>A	CASPHLMKL	HLA-C12:03	59.5363	CACPHLMKL	402.1778
	IGF1R	c.1267C>T	LEGNYSFYVF	HLA-B18:01	57.618	LEGNYSFYVL	658.2709
	NF1	c.2999G>T	MMLNLVRYVL	HLA-A02:06	54.6346	MMLNLVRYVR	482.0457
Patient No.8 (low)	ERRFI1	c.989C>T	LRTSPKSL	HLA-C07:02	263.7465	SRTSPKSL	368.2361
	MLL	c.2990G>T	STPSSITVK	HLA-A11:01	58.2713	STPSSSTVK	149.6165
	MLL2	c.10405G>T	LSGGPSSYL	HLA-C03:04	324.7736	LSGGPSSDL	2124.6111
	MAP2K1	c.1023C>G	FQDFVNKWL	HLA-A02:01	75.3761	FQDFVNKCLI	411.2826
	ABCC11	c.800A>T	GVVNYLFEGV	HLA-A02:01	20.9501	GDVNYLFEGV	2189.3938
	POLD1	c.1331G>T	RLDTKVSSMV	HLA-A02:01	185.0427	RRDTKVSSMV	29606.3418
	BAP1	c.1562G>T	SANPTRPSI	HLA-C03:04	32.9808	SANPTRPSS	9319.5254
	ATR	c.326G>T	LLIAATPSCHL	HLA-A02:01	127.6918	LRIAATPSCHL	22439.4004
	IDH2	c.515G>T	KPITIGMHA	HLA-B55:02	119.7705	KPITIGRHA	401.174
CIC	c.3614G>T	SGRPGPAPL	HLA-C03:04	293.506	SGRPGPAPR	32670.084	

Abbreviations: HLA, human leukocyte antigen; WT, wild type

A

B

C

Reviewer Fig. RF6 (Fig S5). In vitro validation of functional T cells stimulated by predicted neoantigens.

A. Flow cytometry analysis for the expression of 4-1BB on CD8+ lymphocytes after co-culture with predicted neoantigens or wild-type counterparts in patients with high (upper) and low HAPS (lower). B. Proportion of predicted neoantigens leading to up-regulation of 4-1BB on CD8+ T cells in different groups. C. Higher activation of 4-1BB on CD8+ lymphocytes stimulated by predicted neoantigens in the high HAPS group than in the low HAPS group.

3. Could you explain the rationale of measuring 4-1BB on CD8 T cells?

Response: We thank the reviewer for this comment. 4-1BB (also known as CD137) is a member of the tumor necrosis factor receptor superfamily that is expressed on a variety of immune cells, including CD8+ T cells, following activation. Its expression indicates that the T cell has recently encountered its specific antigen and has been activated (23, 24). In the context of cancer immunotherapy, 4-1BB is an important costimulatory receptor. Its stimulation can enhance T cell function, promote survival, and induce proliferation of CD8+ T cells. Therefore, measuring the expression of 4-1BB on CD8+ T cells can be useful to understand and evaluate the effectiveness of immune-based therapies.

4. The observation that HAPS predicts ICI response outcomes rather than overall survival is interesting. How do the authors think this relates to observations of tumour evolution and tumour-immune editing? Please discuss.

Response: Thanks a lot for this comment. According to our analysis, the HAPS did not robustly predict overall survival in the TCGA cohort. Almost all patients included in TCGA database did not receive subsequent immunotherapy. This result meant that HAPS is not a prognostic biomarker correlated with overall survival. We developed HAPS to predict the benefit from immunotherapy. As a result, patients with a high HAPS exhibited a longer PFS and OS than those with a low HAPS in several ICI-treated cohorts, implying that survival benefit based on HAPS was due to immunotherapy. These results further indicated that HAPS is a predictor for response to immunotherapy rather than a prognostic biomarker for OS. From the perspectives of tumor evolution and immune editing, patients with high HAPS tend to have more tumor neoantigens that can be presented by HLA-I. As a result, these neoantigens are more likely to be recognized and targeted by T cells. This means that tumor clones capable of being presented are more susceptible to elimination during the immune response, a process known as immune editing. However, tumors often develop the ability to evade this immune response due to the blockade of immune checkpoints. Fortunately, the use of immunotherapy can counteract this inhibition, potentially leading to improved therapeutic outcomes and better prognosis for these patients.

5. Immune signatures were defined in the methods section. It seems that MHC-I immune infiltration was simply defined by expression levels of MHC class I alpha chains. This does not give an understanding of immune-infiltration, but of HLA expression levels? MHC class II immune infiltration signatures included MHC class I pathway molecules TAP1, TAP2, TAPBP which is confusing? Did the authors look at expression of immune cell markers etc? Much more detail and discussion needed.

Response: We thank the reviewer for these comments. Single-sample Gene Set Enrichment Analysis (ssGSEA) we used in our analyses is a widely adopted bioinformatics approach. It works by analyzing the known immune-related gene sets,

which enables the determination of the relative abundance of immune cell subtypes in each sample. This, in turn, provides insights into the extent and types of immune cell infiltration in the tumor microenvironment. It has been validated as an effective tool to evaluate immune infiltration in previous studies(8, 9). In this study, beyond evaluating the expression levels of HLA, we also incorporated both the GEP score and the CYT score. These scores potentially reflect aspects such as T cell infiltration, antigen presentation, chemokine expression, and cytotoxic activity. Collectively, these metrics highlight that patients with high HAPS demonstrate an elevated potential for immune response.

We sincerely apologize for the mistake. The MHC-I immune infiltration signature includes B2M, TAP1, TAP2, TAPBP, HLA-A, HLA-B, HLA-C, HLA-E, HLA-F, and HLA-G, whereas MHC-II includes HLA-DMA, HLA-DMB, HLA-DOA, HLA-DOB, HLA-DPA1, HLA-DPA2, HLA-DPA3, HLA-DPB1, HLA-DPB2, HLA-DQA1, HLA-DQA2, HLA-DQB1, HLA-DQB2, HLA-DRA, HLA-DRB1, HLA-DRB2, HLA-DRB3, HLA-DRB4, HLA-DRB5, HLA-DRB6, HLA-DRB7, HLA-DRB8 and HLA-DRB9. We have made the corrections in the revised manuscript, please refer to line 683-688. Beyond the aforementioned analyses, this study employed ssGSEA to investigate the relative infiltration levels of 25 immune cell types. The results revealed that only dendritic cells (pDC) exhibited a weak positive correlation with HAPS ($r=0.112$, $P<0.001$; see Fig. 4C, Supplementary Fig. S15).

6.The authors final model needs to be benchmarked to related models to proof superiority.

Response: We are grateful for the reviewer's insightful suggestion. In our study, during the process of model selection, we leveraged the "caret" package to accommodate the AUC value fitting for diverse models and varying numbers of included parameters. Our results identified the optimal performance of the neural network model that is based on three parameters (HAPS, TCR diversity, and TMB), achieving an AUC=0.78, as delineated in the Reviewer Fig. RF4 (newly added Figure 6B). Subsequently, we ranked the nine candidate factors based on their importance and selected the top three - HAPS, TCR diversity, and TMB - to establish our model (Figure 6C). Further validation was conducted by comparing residuals among models, which underscored the Neural Network (NN) as the most suitable choice (Figure 6D).

Reviewer Fig. RF7 (Figure 6B). Comparison of AUCs for various combinations of models and parameter counts using Caret.

Minor comments:

To note that I found it a little difficult to follow the paper as the formatting was not consistent, i.e. section headers were not highlighted in bold, and abstract title missing.

Response: We appreciate the reviewer's kind suggestion. We have now highlighted the section headers in bold and supplemented abstract title.

In figures, it would be good if axis titles could be reviewed, as they were often not clear, i.e. Figure 1A/C and 2J. What is "the ratio of HLA divergence" etc. Figure D/E: both HR and p-value are given in one graph, but only one axis is shown. Please add p-values, too.

Response: We appreciate the reminder from the reviewer. Accordingly, we have revised the y-axis label from "the ratio of HLA divergence" to "Density" in Figures 1A/C and 2J. In Figures 1D/E, the y-axis label "Red:HR; Blue:Pval" encompasses both HR value and P value, with the red line representing HR and the blue line representing P value.

Reviewer #3 (Remarks to the Author): expertise in HLA neoantigen prediction

The authors describe an integrated scoring metric combining tumor neoantigen

burden along with HLA allele divergence to develop a combined HLA tumor-Antigen Presentation Score (HAPS). This score is then evaluated in patient cohorts that have undergone immune checkpoint blockade therapy for survival benefit as well as correlation with various other metrics such as gene expression relating to the tumor microenvironment. This is an interesting idea, however I'm not fully convinced of the added value of using HAPS with the currently presented data compared with previous published studies, please see detailed comments below:

Response: We thank the reviewer for the positive comments and constructive suggestions.

Major comments:

The authors suggest that HAPS consists of two elements, tumor neoantigen burden and HLA allele divergence score. Can the authors please clarify details on how tumor neoantigen burden is defined for all sample cohorts, including details of handling cases such as 1) multiple HLA alleles of the patient bind strongly to the same peptide, 2) discrepancies among different prediction algorithms used?

Response: We appreciate the reviewer's insightful comments. Within the NetMHCpan4.0 neoantigen prediction algorithm, a single peptide could potentially bind with two similar HLA-I alleles, resulting in repeated calculations. At present, there is no established method to circumvent this issue. To understand the implications of this phenomenon in our study, we performed a manual check in the Training1 cohort to ascertain the frequency of the instance. Out of 30 cases, this occurrence was observed in two patients with a very low rate (2/30), both falling within the low HAPS group. These results indicated the phenomenon of multiple HLA alleles binding strongly to the same peptide is rare. Furthermore, to assess the influence of varying neoantigen prediction strategies on our study outcomes, we employed NetMHCpan4.1, which utilized a 1% rank instead of IC50 in NetMHCpan4.0 for neoantigen prediction. Both methods exhibited no difference in predicting survival benefits from immunotherapy compared with primary calculation methods, as illustrated in Fig S7.

Furthermore, it seems that NetMHCpan4.0 is one of prediction algorithms used, which in itself utilizes a “neighboring” estimation approach for HLA alleles with no prediction data. This estimation approach I believe is based on HLA allele “similarity” in key sequences for peptide binding. If the authors are using NetMHCpan4.0, it seems that the idea of HLA allele divergence might be somewhat already accounted for. For example, for HLA alleles with low divergence, it is likely that prediction algorithms will also predict them to be of similar binding to the same group of peptides and thus if you were to deduplicate and remove peptides in such cases, the resulting TNB is naturally lower. Can authors demonstrate the added value of using HAPS over using a properly defined tumor neoantigen burden?

Response: We concur with the reviewer's comments that a single mutation, predicted by NetMHCpan4.0 to bind with HLA alleles of low divergence, might result in an inflated TNB. To delineate the influence of this phenomenon on our study results, we undertook a manual check in the Training1 cohort to ascertain the proportion of instances where a single mutation was presented by two different alleles. Out of 30 cases, this occurrence was observed only in two patients, both falling within the low HAPS group. Following the removal of redundantly calculated neoantigens, we compared the prognostic value between HAPS and the revised TNB. The findings revealed that HAPS still outperforms TNB. The results remain consistent with those presented in the manuscript, demonstrating superior performance of HAPS.

Reviewer Fig. RF1. Overall survival stratified by HAPS and refined TNB

Association of HAPS and refined TNB and HAPS with overall survival in the Wang-WES cohort.

Previously, Chan et al have demonstrated a similar concept regarding HLA allele divergence and ICI treatment response in their 2019 manuscript (PMID: 31700181). Here, authors stratify patients groups based on high and low HAPS throughout the manuscript to demonstrate differences in survival outcomes. I think it's important that authors more explicitly spell out what is novel in this study, particularly the added value of using HAPS versus solely using a tumor neoantigen burden and/or HLA allele divergence score, which I believe is a key finding of the paper that would demonstrate the novelty of this approach. Supplemental figure S6 seems to show this to some degree but a quantitative summarization of this across different cohorts would be more convincing. Additionally the authors should specify how the cutoffs for high vs low TNB and HLA alleles divergence are determined and what the cutoffs are exactly.

Response: We thank the reviewer for these constructive comments. Firstly, the divergence in HLA alleles represents the variety of neoantigens that bind to HLA-I molecules, whereas the number of tumor neoantigens, predicted by the affinity of individual HLA-I genotypes, has been found to be distributed across different HLA-I

subtypes (21). The HAPS model we developed, grounded in bioinformatics, biologically reflects tumor immunogenicity (TNB) and neoantigen presentation (HLA allele divergence). Pursuant to the reviewer's suggestion, we performed survival analyses comparing HAPS, HLA divergence, and TNB. The findings illustrated that all these indicators can distinguish patients reaping benefits from immunotherapy, with HAPS displaying the lowest P value and HR value (Reviewer Fig. RF2). For a definitive comparison of the independent predictive values of these factors, we incorporated TNB and HLA divergence into the multivariate regression, and pinpointed HAPS as an independent determinant of OS (Reviewer Fig. RF3). In addition, the cutoff values for TNB and HLA divergence were both set at the median, which were 219 and 6.48, respectively.

Reviewer Fig. RF2. Summarized overall survival stratified by HLA divergence, TNB and

Association of HLA divergence, TNB and HAPS with overall survival in ICI treated cohorts.

Multi Cox Analysis	Pvalue	HR	Lower 95%CI	Uppper 95%CI	
TMB >14 VS <=14	0.761051174	0.923089725	0.551108316	1.546147312	
PDL1 >=1% VS <1%	0.363414478	0.813921371	0.522103519	1.268844154	
Stage IV VS III	0.226330261	1.150054912	0.916970669	1.442386704	
Age >65 VS <=65	0.686855385	0.936203945	0.679479261	1.289925796	
Gender Male VS Female	0.915067153	1.009566869	0.847504253	1.202619644	
HLA Divergence	0.621666186	0.948741617	0.769817762	1.169251607	
TNB	0.207259412	0.877484948	0.716188061	1.075108447	
HAPS High VS Low	0.030829966	0.762994736	0.596878819	0.975341977	

Reviewer Fig. RF3. Multivariate Cox regression model assessing the correlation of clinical variables with OS.

Building on top of the previous comment, the current definition of TNB seems to be solely based on binding affinity predictions. Could the authors add more complexity to the definition of TNB, such as additional filtering using RNA

expression data (gene or transcript)? I understand this might be for a limited number of cohorts but would be nice to see if it improves survival stratification.

Response: We truly thank the reviewer for the constructive suggestion. In response to the editor's suggestions, we have integrated RNA expression data into neoantigen prediction utilizing NetMHCpan4.0, and subsequently applied refined HAPS to three immunotherapy cohorts presented within the study (Reviewer Fig. RF4). However, HAPS with RNA did not yield improvement in stratifying survival benefit in comparison to the current HAPS.

Reviewer Fig. RF4. Overall survival stratified by HAPS with RNA.

Association of HAPS with RNA and overall survival in four ICI-treated cohorts with RNA data.

Since HAPS puts an emphasis on allele-specific differences, I believe it would be more appropriate to either use percentile predictions (e.g. 2% cutoff) and or elution rankings (e.g. common threshold rank of ~1.1 in NetMHCpan 4.0) when determining TNB instead of IC50 binding affinity or at least use allele-specific binding affinity cutoffs where available. Please see: <https://help.iedb.org/hc/en-us/articles/114094151811-Selecting-thresholds-cut-offs-for-MHC-class-I-and-II-binding-predictions>. I noticed that supplementary figure S7 shows a comparison between using IC50 vs percentile rankings, but lacks description of percentile cutoff that was used to determine TNB.

Response: We truly thank the reviewer for the constructive suggestion. The neoantigens predicted in our research were based on NetMHCpan4.0, a method trained on binding affinity and eluted ligand data leveraging the information from the both data types (3), which is commonly used to predict neoantigens(4, 5). We then applied NetMHCpan4.1 and used %Rank < 1% for neoantigen prediction used for HAPS construction. The two methods showed very similar survival curves and performance in predicting survival benefit from immunotherapy. We have added the description of percentile cutoff to line 164.

It would be a great resource to the community if authors could make their HAPS calculation pipeline available for others to use to calculate scores between any

HLA allele of their choice or provide a supplementary table with HAPS scores between common HLA alleles of each class for others to look up.

Response: We appreciate the reviewer's suggestion. We have uploaded the pipeline used for HAPS calculation and HLA-I allele divergence data to github (<https://github.com/zxl2014swjx/HAPS/tree/main/hed/db/>). Other researchers can obtain the patient's HAPS score using the data and methods in this link, provided that they have obtained the patient's TNB and HLA typing.

Minor comments:

The authors use “neoantigen”, “neoepitope” and “neopeptide” throughout the manuscript without explaining a clear difference between the three terms.

Response: We apologize for the mistake. In fact, "neoantigen," "neoepitope," and "neopeptide" all refer to the same concept of neoantigen. In the revised manuscript, we have unified the descriptions and used "neoantigen" consistently throughout.

Figure captions should be more detailed and figure title/labels need additional attention. Some examples would be Figure S2 subplot titles and an explanation of what authors mean by “Ratio of HLA divergence”, “Ratio of log₁₀(TNB+1)”, “Ratio of HAPS” in Figure 1A. Also please define all acronyms when they first occur in the manuscript, including those in figure legends.

Response: We apologize to the reviewer for this error. We have revised the y-axis label from "the ratio of HLA divergence" to "Density" in Figures 1A and have corrected the subplot titles in Figure S2. Moreover, we have provided definitions for all acronyms used throughout the manuscript.

For violin/box plots such as Fig S1, could the authors provide details on the sample sizes for homozygous and heterozygous cases?

Response: We thank the reviewer for the comments. We have annotated the sample sizes for both homozygous and heterozygous cases in Figure S1.

The authors could probably improve on their alignment and somatic variant detection pipeline by using a more updated human reference and more than just Mutect for somatic variant detection. Also the precise details for how somatic variants are called were lacking in the methods.

Response: We apologize for the lack of a detailed description in the methodology. We have added the details of both "Whole exome sequencing and analysis" and "Targeted genomic sequencing and analysis" to the revised manuscript. Please refer to lines 584-630 in the Method section.

It appears that methods sections are missing for the in vitro T-cell stimulation

performed and neoantigen quality calculations. Specifically for neoantigen quality calculations, are the scores averaged across the number of neoantigens per patient?

Response: We apologize for this oversight and thank the reviewer for the kind reminder. We have added the method for T cell stimulation and neoantigen quality calculations to the Method section. Please refer to line 647-653 and 732-748. The methodology for quantifying neoantigen quality involves an individual assessment of each unique neoantigen. Subsequently, the overall neoantigen quality within each patient is derived from the mean value of all individual neoantigen masses present. This approach ensures that the calculation remains unaffected by the sheer number of neoantigens.

$$D(\mathbf{p}^{\text{WT}} \rightarrow \mathbf{p}^{\text{MT}}) = (1 - w) \log \left(\frac{K_d^{\text{WT}}}{K_d^{\text{MT}}} \right) + w \log \left(\frac{EC_{50}^{\text{MT}}}{EC_{50}^{\text{WT}}} \right)$$

D , self discrimination. P^{WT} , sequence similarity of the wild-type neopeptide. P^{MT} , sequence similarity of the mutant neopeptide. $EC_{50}^{\text{MT}}/EC_{50}^{\text{WT}}$, TCRs cross-reactivity distance. $K_d^{\text{WT}}/K_d^{\text{MT}}$, differential MHC presentation. K , w sets the relative weight between the two terms. W , sets the relative weight between the two terms.

1. J. Robinson, J. A. Halliwell, J. D. Hayhurst, P. Flicek, P. Parham, S. G. Marsh, The IPD and IMGT/HLA database: allele variant databases. *Nucleic Acids Res* **43**, D423-431 (2015).
2. D. Chowell, C. Krishna, F. Pierini, V. Makarov, N. A. Rizvi, F. Kuo, L. G. T. Morris, N. Riaz, T. L. Lenz, T. A. Chan, Evolutionary divergence of HLA class I genotype impacts efficacy of cancer immunotherapy. *Nat Med* **25**, 1715-1720 (2019).
3. V. Jurtz, S. Paul, M. Andreatta, P. Marcatili, B. Peters, M. Nielsen, NetMHCpan-4.0: Improved Peptide-MHC Class I Interaction Predictions Integrating Eluted Ligand and Peptide Binding Affinity Data. *J Immunol* **199**, 3360-3368 (2017).
4. M. Montesion, K. Murugesan, D. X. Jin, R. Sharaf, N. Sanchez, A. Guria, M. Minker, G. Li, V. Fisher, E. S. Sokol, D. C. Pavlick, J. A. Moore, A. Braly, G. Singal, D. Fabrizio, L. A. Comment, N. A. Rizvi, B. M. Alexander, G. M. Frampton, P. S. Hegde, L. A. Albacker, Somatic HLA Class I Loss Is a Widespread Mechanism of Immune Evasion Which Refines the Use of Tumor Mutational Burden as a Biomarker of Checkpoint Inhibitor Response. *Cancer Discov* **11**, 282-292 (2021).
5. Z. Mao, P. A. Nesterenko, J. McLaughlin, W. Deng, G. Burton Sojo, D. Cheng, M. Noguchi, W. Chour, D. C. DeLucia, K. A. Finton, Y. Qin, M. B. Obusan, W. Tran, L. Wang, N. J. Bangayan, L. Ta, C. C. Chen, C. S. Seet, G. M. Crooks, J. W. Phillips, J. R. Heath, R. K. Strong, J. K. Lee, J. A. Wohlschlegel, O. N. Witte, Physical and in silico immunopeptidomic profiling of a cancer antigen prostatic acid phosphatase reveals targets enabling TCR isolation. *Proc Natl Acad Sci U S A* **119**, e2203410119 (2022).
6. D. Chowell, L. G. T. Morris, C. M. Grigg, J. K. Weber, R. M. Samstein, V. Makarov, F. Kuo, S. M. Kendall, D. Requena, N. Riaz, B. Greenbaum, J. Carroll, E. Garon, D. M. Hyman, A. Zehir, D. Solit, M. Berger, R. Zhou, N. A. Rizvi, T. A. Chan, Patient HLA class I genotype influences cancer response to checkpoint blockade immunotherapy. *Science* **359**, 582-587 (2018).
7. A. Ravi, M. D. Hellmann, M. B. Arniella, M. Holton, S. S. Freeman, V. Naranbhai, C. Stewart,

- I. Leshchiner, J. Kim, Y. Akiyama, A. T. Griffin, N. I. Vokes, M. Sakhi, V. Kamesan, H. Rizvi, B. Ricciuti, P. M. Forde, V. Anagnostou, J. W. Riess, D. L. Gibbons, N. A. Pennell, V. Velcheti, S. R. Digumarthy, M. Mino-Kenudson, A. Califano, J. V. Heymach, R. S. Herbst, J. R. Brahmer, K. A. Schalper, V. E. Velculescu, B. S. Henick, N. Rizvi, P. A. Jänne, M. M. Awad, A. Chow, B. D. Greenbaum, M. Luksza, A. T. Shaw, J. Wolchok, N. Hacohen, G. Getz, J. F. Gainor, Genomic and transcriptomic analysis of checkpoint blockade response in advanced non-small cell lung cancer. *Nat Genet* **55**, 807-819 (2023).
8. S. Krishna, F. J. Lowery, A. R. Copeland, E. Bahadiroglu, R. Mukherjee, L. Jia, J. T. Anibal, A. Sachs, S. O. Adebola, D. Gurusamy, Z. Yu, V. Hill, J. J. Gartner, Y. F. Li, M. Parkhurst, B. Paria, P. Kvistborg, M. C. Kelly, S. L. Goff, G. Altan-Bonnet, P. F. Robbins, S. A. Rosenberg, Stem-like CD8 T cells mediate response of adoptive cell immunotherapy against human cancer. *Science* **370**, 1328-1334 (2020).
9. H. Hackl, P. Charoentong, F. Finotello, Z. Trajanoski, Computational genomics tools for dissecting tumour-immune cell interactions. *Nat Rev Genet* **17**, 441-458 (2016).
10. M. Shugay, D. V. Bagaev, I. V. Zvyagin, R. M. Vroomans, J. C. Crawford, G. Dolton, E. A. Komech, A. L. Sycheva, A. E. Koneva, E. S. Egorov, A. V. Eliseev, E. Van Dyk, P. Dash, M. Attaf, C. Rius, K. Ladell, J. E. McLaren, K. K. Matthews, E. B. Clemens, D. C. Douek, F. Luciani, D. van Baarle, K. Kedzierska, C. Kesmir, P. G. Thomas, D. A. Price, A. K. Sewell, D. M. Chudakov, VDJdb: a curated database of T-cell receptor sequences with known antigen specificity. *Nucleic Acids Res* **46**, D419-d427 (2018).
11. D. Miao, C. A. Margolis, N. I. Vokes, D. Liu, A. Taylor-Weiner, S. M. Wankowicz, D. Adeegbe, D. Keliher, B. Schilling, A. Tracy, M. Manos, N. G. Chau, G. J. Hanna, P. Polak, S. J. Rodig, S. Signoretti, L. M. Sholl, J. A. Engelman, G. Getz, P. A. Jänne, R. I. Haddad, T. K. Choueiri, D. A. Barbie, R. Haq, M. M. Awad, D. Schadendorf, F. S. Hodi, J. Bellmunt, K. K. Wong, P. Hammerman, E. M. Van Allen, Genomic correlates of response to immune checkpoint blockade in microsatellite-stable solid tumors. *Nat Genet* **50**, 1271-1281 (2018).
12. E. M. Van Allen, D. Miao, B. Schilling, S. A. Shukla, C. Blank, L. Zimmer, A. Sucker, U. Hillen, M. H. G. Foppen, S. M. Goldinger, J. Utikal, J. C. Hassel, B. Weide, K. C. Kaehler, C. Loquai, P. Mohr, R. Gutzmer, R. Dummer, S. Gabriel, C. J. Wu, D. Schadendorf, L. A. Garraway, Genomic correlates of response to CTLA-4 blockade in metastatic melanoma. *Science* **350**, 207-211 (2015).
13. V. Anagnostou, N. Niknafs, K. Marrone, D. C. Bruhm, J. R. White, J. Naidoo, K. Hummelink, K. Monkhorst, F. Lalezari, M. Lanis, S. Rosner, J. E. Reuss, K. N. Smith, V. Adleff, K. Rodgers, Z. Belcaid, L. Rhymee, B. Levy, J. Feliciano, C. L. Hann, D. S. Ettinger, C. Georgiades, F. Verde, P. Illei, Q. K. Li, A. S. Baras, E. Gabrielson, M. V. Brock, R. Karchin, D. M. Pardoll, S. B. Baylin, J. R. Brahmer, R. B. Scharpf, P. M. Forde, V. E. Velculescu, Multimodal genomic features predict outcome of immune checkpoint blockade in non-small-cell lung cancer. *Nat Cancer* **1**, 99-111 (2020).
14. M. D. Hellmann, T. Nathanson, H. Rizvi, B. C. Creelan, F. Sanchez-Vega, A. Ahuja, A. Ni, J. B. Novik, L. M. B. Mangarin, M. Abu-Akeel, C. Liu, J. L. Sauter, N. Rekhtman, E. Chang, M. K. Callahan, J. E. Chaft, M. H. Voss, M. Tenet, X. M. Li, K. Covello, A. Renninger, P. Vitazka, W. J. Geese, H. Borghaei, C. M. Rudin, S. J. Antonia, C. Swanton, J. Hammerbacher, T. Merghoub, N. McGranahan, A. Snyder, J. D. Wolchok, Genomic Features of Response to Combination Immunotherapy in Patients with Advanced Non-Small-Cell Lung Cancer.

- Cancer Cell* **33**, 843-852.e844 (2018).
15. W. Hugo, J. M. Zaretsky, L. Sun, C. Song, B. H. Moreno, S. Hu-Lieskovan, B. Berent-Maoz, J. Pang, B. Chmielowski, G. Cherry, E. Seja, S. Lomeli, X. Kong, M. C. Kelley, J. A. Sosman, D. B. Johnson, A. Ribas, R. S. Lo, Genomic and Transcriptomic Features of Response to Anti-PD-1 Therapy in Metastatic Melanoma. *Cell* **165**, 35-44 (2016).
 16. N. A. Rizvi, M. D. Hellmann, A. Snyder, P. Kvistborg, V. Makarov, J. J. Havel, W. Lee, J. Yuan, P. Wong, T. S. Ho, M. L. Miller, N. Rekhtman, A. L. Moreira, F. Ibrahim, C. Bruggeman, B. Gasmi, R. Zappasodi, Y. Maeda, C. Sander, E. B. Garon, T. Merghoub, J. D. Wolchok, T. N. Schumacher, T. A. Chan, Cancer immunology. Mutational landscape determines sensitivity to PD-1 blockade in non-small cell lung cancer. *Science* **348**, 124-128 (2015).
 17. W. Roh, P. L. Chen, A. Reuben, C. N. Spencer, P. A. Prieto, J. P. Miller, V. Gopalakrishnan, F. Wang, Z. A. Cooper, S. M. Reddy, C. Gumbs, L. Little, Q. Chang, W. S. Chen, K. Wani, M. P. De Macedo, E. Chen, J. L. Austin-Breneman, H. Jiang, J. Roszik, M. T. Tetzlaff, M. A. Davies, J. E. Gershenwald, H. Tawbi, A. J. Lazar, P. Hwu, W. J. Hwu, A. Diab, I. C. Glitza, S. P. Patel, S. E. Woodman, R. N. Amaria, V. G. Prieto, J. Hu, P. Sharma, J. P. Allison, L. Chin, J. Zhang, J. A. Wargo, P. A. Futreal, Integrated molecular analysis of tumor biopsies on sequential CTLA-4 and PD-1 blockade reveals markers of response and resistance. *Sci Transl Med* **9**, (2017).
 18. A. Snyder, V. Makarov, T. Merghoub, J. Yuan, J. M. Zaretsky, A. Desrichard, L. A. Walsh, M. A. Postow, P. Wong, T. S. Ho, T. J. Hollmann, C. Bruggeman, K. Kannan, Y. Li, C. Elipenahli, C. Liu, C. T. Harbison, L. Wang, A. Ribas, J. D. Wolchok, T. A. Chan, Genetic basis for clinical response to CTLA-4 blockade in melanoma. *N Engl J Med* **371**, 2189-2199 (2014).
 19. N. Riaz, J. J. Havel, V. Makarov, A. Desrichard, W. J. Urba, J. S. Sims, F. S. Hodi, S. Martin-Algarra, R. Mandal, W. H. Sharfman, S. Bhatia, W. J. Hwu, T. F. Gajewski, C. L. Slingluff, Jr., D. Chowell, S. M. Kendall, H. Chang, R. Shah, F. Kuo, L. G. T. Morris, J. W. Sidhom, J. P. Schneck, C. E. Horak, N. Weinhold, T. A. Chan, Tumor and Microenvironment Evolution during Immunotherapy with Nivolumab. *Cell* **171**, 934-949.e916 (2017).
 20. J. Han, J. Duan, H. Bai, Y. Wang, R. Wan, X. Wang, S. Chen, Y. Tian, D. Wang, K. Fei, Z. Yao, S. Wang, Z. Lu, Z. Wang, J. Wang, TCR Repertoire Diversity of Peripheral PD-1(+)CD8(+) T Cells Predicts Clinical Outcomes after Immunotherapy in Patients with Non-Small Cell Lung Cancer. *Cancer Immunol Res* **8**, 146-154 (2020).
 21. M. Yadav, S. Jhunjunwala, Q. T. Phung, P. Lupardus, J. Tanguay, S. Bumbaca, C. Franci, T. K. Cheung, J. Fritsche, T. Weinschenk, Z. Modrusan, I. Mellman, J. R. Lill, L. Delamarre, Predicting immunogenic tumour mutations by combining mass spectrometry and exome sequencing. *Nature* **515**, 572-576 (2014).
 22. M. Łuksza, Z. M. Sethna, L. A. Rojas, J. Lihm, B. Bravi, Y. Elhanati, K. Soares, M. Amisaki, A. Dobrin, D. Hoyos, P. Guasp, A. Zebboudj, R. Yu, A. K. Chandra, T. Waters, Z. Odgerel, J. Leung, R. Kappagantula, A. Makohon-Moore, A. Johns, A. Gill, M. Gigoux, J. Wolchok, T. Merghoub, M. Sadelain, E. Patterson, R. Monasson, T. Mora, A. M. Walczak, S. Cocco, C. Iacobuzio-Donahue, B. D. Greenbaum, V. P. Balachandran, Neoantigen quality predicts immunoediting in survivors of pancreatic cancer. *Nature* **606**, 389-395 (2022).
 23. E. Tran, M. Ahmadzadeh, Y. C. Lu, A. Gros, S. Turcotte, P. F. Robbins, J. J. Gartner, Z. Zheng, Y. F. Li, S. Ray, J. R. Wunderlich, R. P. Somerville, S. A. Rosenberg, Immunogenicity of somatic mutations in human gastrointestinal cancers. *Science* **350**, 1387-1390 (2015).

24. Z. Hu, D. E. Leet, R. L. Allesøe, G. Oliveira, S. Li, A. M. Luoma, J. Liu, J. Forman, T. Huang, J. B. Iorgulescu, R. Holden, S. Sarkizova, S. H. Gohil, R. A. Redd, J. Sun, L. Elagina, A. Giobbie-Hurder, W. Zhang, L. Peter, Z. Ciantra, S. Rodig, O. Olive, K. Shetty, J. Pyrdol, M. Uduman, P. C. Lee, P. Bachireddy, E. I. Buchbinder, C. H. Yoon, D. Neuberg, B. L. Pentelute, N. Hacohen, K. J. Livak, S. A. Shukla, L. R. Olsen, D. H. Barouch, K. W. Wucherpfennig, E. F. Fritsch, D. B. Keskin, C. J. Wu, P. A. Ott, Personal neoantigen vaccines induce persistent memory T cell responses and epitope spreading in patients with melanoma. *Nat Med*, (2021).

REVIEWER COMMENTS

Reviewer #1 (Remarks to the Author):

The manuscript is substantially improved but the following issues and oversights require further attention.

1. Original remark: Overall, figures and their text are undersized. Even viewed on the computer at 200% zoom, they were hard to read. Many figures lacked sufficient explanation in the figure legend or text to be easily interpretable.

>> the work to improve the figures is appreciated, but there are still some graphical issues, with overlapping text and labels, and panels blocking parts of other panels. Survival curve lines are so thick as to obscure the censoring ticks. Overall, all figures should be re-reviewed carefully.

2. GitHub code availability

>> The GitHub repository is improved. However, it offers two links to Zenodo archives of relevant code (one for Neoantigen Pipeline, one for HAPS platform, notably these point to the same URL). A "Permission Denied" error occurred when attempting to access these, so they remain un-reviewed.

3. Original remark: Line 107-108: Was divergence calculated for all possible HLA pairs, or naturally occurring within patients (in your cohorts)?

>> The authors response says that divergence was calculated for all possible pairs of HLA (ie. Not just those within a given patient). This seems incorrect, as they say later that a patient that is homozygous for an allele would have a HAPS score of 0 (as there is no divergence between two of the same alleles).

4. Figure S12

>> The authors responded that PD-L1 expression units are percentage, and have corrected the figure. However, the figure seems unchanged (the axis still is from 0 to 1). What does percentage mean when being used as an expression value? What does 100% PD-L1 expression mean?

5. Original remark: Line 271-272: Is it realistic to expect that there are ~16,000 genes that are differentially expressed between HAPS high and low groups? That is the majority of genes. If this is a true result, why are there three times as many downregulated as upregulated?

>> The authors have provided a corrected differential gene expression analysis, with 13 up and 12 downregulated genes. This is a major change to reported results. What effect has this had on the downstream GSEA analysis which uses these genes? The presented results remain unchanged between initial submission and this revision.

6. Original remark: Line 301: The authors state that TCR clonality is increased (by 0.08) over time in half the patients with high HAPS/high diversity, and that it decreases for low HAPS and low diversity. However, the other half of patients in the high/high group showed decreased clonality, apparently with greater magnitude. This is a questionable description of the data – the high group likely has no overall change in clonality if a statistical test was done.

>> rewording of this section is recommended, highlighting that only half of the high/high group decrease while all of the low/low group decrease (which seems to be what the authors are trying to say), and directly compare/test the magnitude of the decreases in each group.

7. Original remark: Line 346-347: This sounds like you applied the NN model to the same patients that were used to train it. Presumably this is not the case.

>> The authors have responded that the model was trained on the blood cohort (Wang-Panel-B), and validated on the tissue cohort (Wang-Panel-T). However, the text is still not clear regarding this. Revised manuscript lines 332-334: "Due to the small number of cases enrolled in the Wang-Panel-T cohort, we used the Wang-Panel-B cohort for model construction" (this is consistent with Authors response). Revised manuscript lines 346-347: "To validate the NN model in predicting ICI benefit, we applied it in 52 patients of the Wang- Panel-B cohort with blood panel-based HAPS, TCR diversity, and TMB values available" (this is inconsistent, and says it was validated on the blood cohort).

8. Original remark: Line 611: How were HLA genotypes determined? Existing data, or analyzed by the authors? What tools were used (by authors or existing data)? If it was NGSbased, what read length(s) were used?

>> The authors have responded with the methods used. Please add this description to the Methods section.

9. Original remark: Line 615: Which IEDB algorithms were used?

>> The authors have clarified that only NetMHCpan was used. They should remove "and The Immune Epitope Database (IEDB) algorithms" from revised manuscript lines 644-645.

10. Original remark: Line 661: Significantly more detail is needed on how the TCR-b sequencing was done. How were the cells isolated? How were CDR3 regions selected? Were primers used? Was this from RNA? DNA? What sequence machine was used? What read length? What sequence depth was achieved? Provide a summary of the sequence data obtained. How was the resulting sequencing quality filtered and annotated with TCR features?

>> The PCR thermocycling description ("one minute at 98°C, 25 cycles at various temperatures, and a final extension") is insufficiently detailed for reproducibility. This section is also still missing details for the depth of sequencing obtained for the samples (total number of reads per sample), any QC steps, and methods for how the raw sequence data was annotated with TCR features.

11. Figure 5F

>> The authors have revised this figure to have the diagonal show the same TCR. However, the sequences displayed in the figure have changed between the original submission and this revision. What is the source of the different data presented?

12. Figure 4G

>> Please label the axes.

13. Figure 3D

>> The end ticks for two of the error bars are misplaced.

14. Figure 2F, 2G

>> The p-values have changed on these figures between the original submission and revised manuscript – what is the basis of these changes?

Reviewer #3 (Remarks to the Author):

The authors have generally addressed my questions and suggestions but I have two remaining concerns.

1. The authors make their code available. This seems important from a reproducibility perspective. However, these two zenodo links could not be accessed when I attempted to do so. They appear to be set to private and should be made public prior to publication.

Neoantigen Pipeline: <https://zenodo.org/deposit/8287023>

HAPS Platform: <https://zenodo.org/deposit/8287023>

2. An open source license of some kind should be explicitly applied to the source code above AND in the git repository to make it clear under what conditions the code is being shared.

- Response to Reviewer 2 point 3. This explanation is helpful background context, not just for the reviewers but also for the readers.

- Response to Reviewer 2 point 4. Same thing. Good response. Would be nice to see a brief version of this in the manuscript.

Reviewer #1 (Remarks to the Author):

The manuscript is substantially improved but the following issues and oversights require further attention.

Response: We appreciate the recognition of the reviewer upon our previous response. Further comments have been carefully responded point-by-point.

1. Original remark: Overall, figures and their text are undersized. Even viewed on the computer at 200% zoom, they were hard to read. Many figures lacked sufficient explanation in the figure legend or text to be easily interpretable.

>> the work to improve the figures is appreciated, but there are still some graphical issues, with overlapping text and labels, and panels blocking parts of other panels. Survival curve lines are so thick as to obscure the censoring ticks. Overall, all figures should be re-reviewed carefully.

Response: Thanks a lot for the suggestions. We have re-checked all figures and texts to make clearly reading.

2. GitHub code availability

>> The GitHub repository is improved. However, it offers two links to Zenodo archives of relevant code (one for Neoantigen Pipeline, one for HAPS platform, notably these point to the same URL). A “Permission Denied” error occurred when attempting to access these, so they remain un-reviewed.

Response: Thank the reviewer for the kindly reminding about making our resources and code available on GitHub and Zenodo. We have already made them publicly accessible.

3. Original remark: Line 107-108: Was divergence calculated for all possible HLA pairs, or naturally occurring within patients (in your cohorts)?

>> The authors response says that divergence was calculated for all possible pairs of HLA (ie. Not just those within a given patient). This seems incorrect, as they say later that a patient that is homozygous for an allele would have a HAPS score of 0 (as there is no divergence between two of the same alleles).

Response: We apologize for the unclear expression. In our cohorts, the divergence was calculated for HLA pairs naturally occurring within patients. Specifically, divergence calculations were conducted for HLA-A, HLA-B, and HLA-C alleles individually in each patient, and the average value was computed as the HAPS of an indicated patient, as detailed in lines 103-105. If all HLA-I alleles of a patient were identified as homozygous, the resulting HAPS score would be zero. However, in cases where a patient had at least one heterozygous HLA allele, their HAPS score deviated from zero. In fact, the probability of all subtypes of HLA-I being homozygous is very low, as evidenced by the fact that among our cohort of 1125

patients, only 42 (3.7%) had a HAPS score of 0. The majority of patients had at least one heterozygous HLA-I allele, making this metric applicable across a broad range of cases.

4. Figure S12

>> The authors responded that PD-L1 expression units are percentage, and have corrected the figure. However, the figure seems unchanged (the axis still is from 0 to 1). What does percentage mean when being used as an expression value? What does 100% PD-L1 expression mean?

Response: We sincerely apologize for the unintentional upload of the pre-updated images. The unit for PD-L1 expression level is a percentage. The method employed to assess PD-L1 expression levels in the cohort was immunohistochemistry, using the E1L3N antibody clone(1). The percentage represents the proportion of PD-L1-positive tumor cells among the total tumor cells. We have included the revised Figure S12 in the supplementary materials.

5. Original remark: Line 271-272: Is it realistic to expect that there are ~16,000 genes that are differentially expressed between HAPS high and low groups? That is the majority of genes. If this is a true result, why are there three times as many downregulated as upregulated?

>> The authors have provided a corrected differential gene expression analysis, with 13 up and 12 downregulated genes. This is a major change to reported results. What effect has this had on the downstream GSEA analysis which uses these genes? The presented results remain unchanged between initial submission and this revision.

Response: We are grateful to the reviewer for this insight and appreciate the constructive comment. In the pathway enrichment analysis, we performed Gene Set Enrichment Analysis (GSEA) utilizing all 19,047 protein-coding genes, encompassing both differentially expressed genes (DEGs) and those not identified as such (2, 3). The methods and genes included remained consistent between the initial and revised versions. Consequently, Figure 4G remained unchanged.

6. Original remark: Line 301: The authors state that TCR clonality is increased (by 0.08) over time in half the patients with high HAPS/high diversity, and that it decreases for low HAPS and low diversity. However, the other half of patients in the high/high group showed decreased clonality, apparently with greater magnitude. This is a questionable description of the data – the high group likely has no overall change in clonality if a statistical test was done.

>> rewording of this section is recommended , highlighting that only half of the high/high group decrease while all of the low/low group decrease (which seems to be what the authors are trying to say), and directly compare/test the magnitude of the decreases in each group.

Response: We appreciate the reviewer's suggestion. The expected clonal expansion of neoantigen-specific T cells in response to immunotherapy, which could be reflected

by an increase in PD-1+CD8+TCR clonality, is indeed a noteworthy aspect of our study. In the blood panel-based HAPS^{high}/TCR diversity^{high} subgroup, TCR clonality showed an increase of 0.08 in 50% of the patients (7 out of 14) and a decrease of 0.17 in the remaining 50% of the patients. Conversely, all patients in the HAPS^{low}&TCR diversity^{low} subgroup consistently exhibited a decline (mean = 0.22; p = 0.041; Figure 5D, Supplementary Figure S16A). The "P" value reflects the statistical significance of the differences in clonality changes between these two groups, as determined by the Wilcoxon test. As for the magnitude of decline in patients between the two groups, there was no significant difference (P=0.63). We have reworded this section according to the reviewer's suggestion; please kindly refer to lines 303-306 in the revised manuscript.

7. Original remark: Line 346-347: This sounds like you applied the NN model to the same patients that were used to train it. Presumably this is not the case.

>> **The authors have responded that the model was trained on the blood cohort (Wang-Panel-B), and validated on the tissue cohort (Wang-Panel-T). However, the text is still not clear regarding this. Revised manuscript lines 332-334: "Due to the small number of cases enrolled in the Wang-Panel-T cohort, we used the Wang-Panel-B cohort for model construction" (this is consistent with Authors response). Revised manuscript lines 346-347: "To validate the NN model in predicting ICI benefit, we applied it in 52 patients of the Wang- Panel-B cohort with blood panel-based HAPS, TCR diversity, and TMB values available" (this is inconsistent, and says it was validated on the blood cohort).**

Response: We apologize for any lack of clarity in our description. The blood cohort (Wang-Panel-B) served for training the dataset and for model construction. We have now substituted the term "validate" with "confirm" to enhance clarity. Please refer to line 349 for your reference.

8. Original remark: Line 611: How were HLA genotypes determined? Existing data, or analyzed by the authors? What tools were used (by authors or existing data)? If it was NGSbased, what read length(s) were used?

>> **The authors have responded with the methods used. Please add this description to the Methods section.**

Response: We thank the reviewer for the kind reminder. We have added this description to the Methods section. Please refer to lines 647-651 in the revised manuscript.

9. Original remark: Line 615: Which IEDB algorithms were used?

>> **The authors have clarified that only NetMHCpan was used. The should remove "and The Immune Epitope Database (IEDB) algorithms" from revised manuscript lines 644-645.**

Response: We appreciate the reviewer's kind suggestion. We have removed "and The Immune Epitope Database (IEDB) algorithms" in the revised manuscript.

10. Original remark: Line 661: Significantly more detail is needed on how the TCR-b sequencing was done. How were the cells isolated? How were CDR3 regions selected? Were primers used? Was this from RNA? DNA? What sequence machine was used? What read length? What sequence depth was achieved? Provide a summary of the sequence data obtained. How was the resulting sequencing quality filtered and annotated with TCR features?

>> The PCR thermocycling description (“one minute at 98° C, 25 cycles at various temperatures, and a final extension”) is insufficiently detailed for reproducibility. This section is also still missing details for the depth of sequencing obtained for the samples (total number of reads per sample), any QC steps, and methods for how the raw sequence data was annotated with TCR features.

Response: We thank the reviewer for the comments. The specific steps of the PCR thermocycling are as follows: one cycle at 98°C for 1 minute; subsequently, 25 cycles consisting of denaturation at 98°C for 20 seconds, annealing at 65°C for 30 seconds, and extension at 72°C for 30 seconds; followed by a final extension at 72°C for 5 minutes. The amplified products were size-selected via agarose gel electrophoresis, targeting fragments between 200-350 bp, purified using the QIAquick Gel Purification Kit (QIAGEN). Finally, paired-end sequencing of these samples was executed using the Illumina HiSeq 3000 platform, achieving a read length of 151 bp. The main quality control steps included the removal of reads bearing adapter sequences at the 5' end or those with over 5% "N" bases. Subsequently, we calculated the average base quality for each read after eliminating low-quality bases (base quality <10) from the 3' end. Further filtration excluded reads with an average quality lower than 15. The sequences of the V, D, and J genes were compared with the ImMunoGeneTics (IMGT) database using the MIXCR software to identify the CDR3 sequence of the TCR. Then, 10⁶ qualified reads per sample were randomly selected for downstream analysis. We have incorporated these detailed steps into the Method section of the manuscript, please refer to line 724-737.

11. Figure 5F

>> The authors have revised this figure to have the diagonal show the same TCR. However, the sequences displayed in the figure have changed between the original submission and this revision. What is the source of the different data presented?

Response: Thank you for your comments. In the initial version, the diagonal elements were unequal between rows and columns due to the lack of self-self edit distance. In response to this, we have revised Figure 5F by including the complete TCR sequences on both the horizontal and vertical axes. With this rearrangement, the diagonal now displays the same TCR sequences.

12. Figure 4G

>> Please label the axes.

Response: We appreciate the reminder from the reviewer. In Figure 4G, the x-axis represents $-\log_{10}(p\text{-value} + 1)$, while the y-axis indicates the antigen presentation-related pathways significantly enriched in patients with a high HAPS based on KEGG, REACTOME, and GO terms. We have added labels to the axes in the updated Figure 4G.

13. Figure 3D

>> **The end ticks for two of the error bars are misplaced.**

Response: We apologize to the reviewer for this error. We have corrected the error bars in Figure 3D.

14. Figure 2F, 2G

>> **The p-values have changed on these figures between the original submission and revised manuscript – what is the basis of these changes?**

Response: We appreciate the reviewer's feedback. In accordance with the previous suggestions, we have employed the Student's t-test to analyze data that adheres to a normal distribution. This pertains specifically to the datasets in Figure 2F and 2G, which have been confirmed to exhibit a normal distribution. Consequently, we have conducted the t-test and revised the corresponding p-values accordingly.

Reviewer #3 (Remarks to the Author):

The authors have generally addressed my questions and suggestions but I have two remaining concerns.

Response: We are grateful to the reviewer's recognition upon our previous response. Further comments have been carefully responded point-by-point.

1. The authors make their code available. This seems important from a reproducibility perspective. However, these two zenodo links could not be accessed when I attempted to do so. They appear to be set to private and should be made public prior to publication.

Neoantigen Pipeline: <https://zenodo.org/deposit/8287023>

HAPS Platform: <https://zenodo.org/deposit/8287023>

Response: Thanks a lot for the reminder regarding making our resources and code available on GitHub and Zenodo. We have indeed made them publicly accessible.

2. An open source license of some kind should be explicitly applied to the source code above AND in the git repository to make it clear under what conditions the code is being shared.

Response: We thank the reviewer for the kindly reminding. We initially planned to make the data publicly available after publication. Following the reviewer's suggestion, we have now updated the data and code to be "publicly available."

Response to Reviewer 2 point 3. This explanation is helpful background context, not just for the reviewers but also for the readers.

Response: We appreciate the reviewer 3's positive feedback and valuable suggestions. We have included a brief explanation regarding 4-1BB in the revised manuscript to be easily read, please refer to lines 152-155.

Response to Reviewer 2 point 4. Same thing. Good response. Would be nice to see a brief version of this in the manuscript.

Response: Thanks a lot for the comments. We have revised and provided a brief version correspondingly as follows. "Patients with high HAPS tend to have more tumor neoantigens that can be presented by HLA-I. The utilization of immunotherapy has the potential to overcome immune evasion mechanisms activated by immune checkpoints in these patients, thus potentially leading to enhanced therapeutic outcomes and a more favorable prognosis." This discussion has been incorporated into the "Discussion" section. Please refer to lines 379-382 in the revised manuscript.

1. V. Anagnostou, N. Niknafs, K. Marrone, D. C. Bruhm, J. R. White, J. Naidoo, K. Hummelink, K. Monkhorst, F. Lalezari, M. Lanis, S. Rosner, J. E. Reuss, K. N. Smith, V. Adleff, K. Rodgers, Z. Belcaid, L. Rhymee, B. Levy, J. Feliciano, C. L. Hann, D. S. Ettinger, C. Georgiades, F. Verde, P. Illei, Q. K. Li, A. S. Baras, E. Gabrielson, M. V. Brock, R. Karchin, D. M. Pardoll, S. B. Baylin, J. R. Brahmer, R. B. Scharpf, P. M. Forde, V. E. Velculescu, Multimodal genomic features predict outcome of immune checkpoint blockade in non-small-cell lung cancer. *Nat Cancer* **1**, 99-111 (2020).
2. A. Subramanian, P. Tamayo, V. K. Mootha, S. Mukherjee, B. L. Ebert, M. A. Gillette, A. Paulovich, S. L. Pomeroy, T. R. Golub, E. S. Lander, J. P. Mesirov, Gene set enrichment analysis: a knowledge-based approach for interpreting genome-wide expression profiles. *Proc Natl Acad Sci U S A* **102**, 15545-15550 (2005).
3. J. Reimand, R. Isserlin, V. Voisin, M. Kucera, C. Tannus-Lopes, A. Rostamianfar, L. Wadi, M. Meyer, J. Wong, C. Xu, D. Merico, G. D. Bader, Pathway enrichment analysis and visualization of omics data using g:Profiler, GSEA, Cytoscape and EnrichmentMap. *Nat Protoc* **14**, 482-517 (2019).

REVIEWERS' COMMENTS

Reviewer #1 (Remarks to the Author):

The manuscript is further approved and generally suitable for publication. There are only two remaining concerns that do not need to prompt another cycle of review, but rather should be verified by the editor.

1. The Github repository only contains code for figure generation. The actual analysis code is allegedly hosted on Zenodo. This would be much more accessible if it was hosted in the same Github repository. Additionally, the Zenodo link that is provided in the Github repository is still not publicly accessible (even though the authors state it is) – it still returns a “Permission required. You do not have sufficient permissions to view this page.” error when trying to view the Zenodo link. Therefore, the analysis code, and any supporting documentation and test data, cannot be assessed, and would not be accessible to readers.

2. Regarding the differential gene expression analysis and gene set enrichment analysis – I would recommend rewording lines 273-279 “To further determine the functional pathways associated with HAPS subgroups, we examined differentially expressed genes [...] . Further, gene set enrichment analysis revealed that [...]” to clarify that the result of the differential gene expression analysis is not used in the gene set enrichment analysis.

Reviewer #1 (Remarks to the Author):

1. The Github repository only contains code for figure generation. The actual analysis code is allegedly hosted on Zenodo. This would be much more accessible if it was hosted in the same Github repository. Additionally, the Zenodo link that is provided in the Github repository is still not publicly accessible (even though the authors state it is) – it still returns a “Permission required. You do not have sufficient permissions to view this page.” error when trying to view the Zenodo link. Therefore, the analysis code, and any supporting documentation and test data, cannot be assessed, and would not be accessible to readers.

Response: We apologize for any inconvenience caused. Due to the large size of the relevant files, specifically 700Mb and 6.2Gb, we were unable to upload them to GitHub, as it has a maximum upload limit of 25Mb. To address this issue, we have now organized and uploaded these code files to Figshare, which can be accessed via links from our GitHub repository (<https://doi.org/10.6084/m9.figshare.24763899.v1> and <https://doi.org/10.6084/m9.figshare.24763629.v1>).

Furthermore, we have set these files on Figshare to be openly downloadable. We have also compiled the download links for these two files and added them to the README file on our GitHub page for easier access and reference.

2. Regarding the differential gene expression analysis and gene set enrichment analysis – I would recommend rewording lines 273-279 “To further determine the functional pathways associated with HAPS subgroups, we examined differentially expressed genes [...] . Further, gene set enrichment analysis revealed that [...]” to clarify that the result of the differential gene expression analysis is not used in the gene set enrichment analysis.

Response: Thank you for your reminder. We have updated the manuscript to enhance clarity in the section you mentioned. The revised text now states, "Additionally, Gene Set Enrichment Analysis (GSEA) of all protein-coding genes revealed that antigen presentation-related pathways were significantly enriched in patients with a high HAPS, based on KEGG, REACTOME, and GO terms (Fig. 4G)." This revision can be found in lines 281-283 of the manuscript.